# Convex Distance Operator Transport:
# A Convex and Geometry-Preserving Formulation

**Junhyoung Chung** [1]  **Euijong Song** [2]  **Won Hwa Kim** [3 4]  **Gunwoong Park** [2 5 6]

## Abstract

We introduce Convex Distance Operator Transport (CDOT), the first convex optimal transport framework that aligns distributions across heterogeneous domains by jointly preserving feature correspondence and intrinsic geometric structure. Specifically, CDOT employs an operator-based regularization that aligns aggregated distance structures by introducing distance and conditional expectation operators. Consequently, the proposed regularization improves the robustness to local geometric variations. We further prove that the resulting CDOT discrepancy is a valid pseudometric on the space of attributed compact metric-measure spaces. In addition, we characterize the relationship between CDOT and Gromov–Wasserstein (GW) through a new notion of dispersion gap, formally elucidating the geometric source of non-convexity in GW compared to the convexity of CDOT. In the finite-sample regime, we derive a non-asymptotic risk bound decomposed into optimization and statistical errors, establishing risk consistency under a globally convergent Frank–Wolfe algorithm. Experiments on synthetic point clouds, brain connectomes, and graph classification benchmarks demonstrate better performance over existing methods, with stable and reliable behavior in practice.

## 1. Introduction

Optimal transport (OT) provides a fundamental framework for comparing probability measures by quantifying the minimal cost required to transport mass across spaces (Villani,

[1]KRAFTON [2]Department of Statistics, Seoul National University, Seoul, South Korea [3]Graduate School of AI, POSTECH, Pohang, South Korea [4]Computer Science and Engineering, POSTECH [5]IPAI, Seoul National University [6]IDIS, Seoul National University. Correspondence to: Gunwoong Park <gw-park23@snu.ac.kr>.

*Proceedings of the 43ʳᵈ International Conference on Machine Learning*, Seoul, South Korea. PMLR 306, 2026. Copyright 2026 by the author(s).

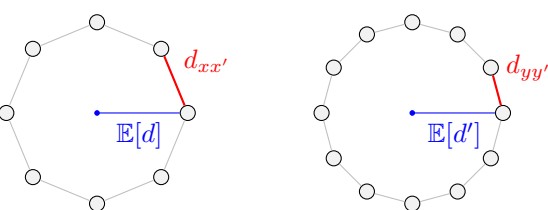

*(a)* Cyclic graph with 8 nodes.   *(b)* Cyclic graph with 12 nodes.

*Figure 1.* Cyclic graphs with different numbers of nodes. While FGW compares distances edge by edge ($|d - d'|^2$), CDOT aligns the aggregated distance profiles ($|\mathbb{E}[d] - \mathbb{E}[d']|^2$), comparing how each node relates to its geometric structure.

2008). As OT induces meaningful correspondences between samples, it has become a powerful tool in statistics and machine learning with various applications ranging from generative modeling to transfer learning (Rubner et al., 2000; Courty et al., 2014; Yuan et al., 2025).

However, the classical OT based on the Wasserstein distance necessitates that data distributions reside in the same space, which poses a restriction in many practical settings where observations come from heterogeneous domains. The Gromov–Wasserstein (GW) framework overcomes this limitation by comparing distributions through their associated metric-measure (mm) spaces, aligning intrinsic structure without requiring an explicit cross-domain cost (Mémoli, 2011). The resulting GW discrepancy induces a pseudometric on the space of compact mm spaces, vanishing if and only if the spaces are measure-preserving isometric. More recently, Vayer et al. (2020) propose the fused Gromov–Wasserstein (FGW) approach, which integrates the structural GW cost with a feature-based OT cost to utilize auxiliary information such as node attributes in graphs.

Despite its versatility, FGW inherits the non-convexity of GW arising from its formulation involving pairwise distance comparisons, thereby standard optimization algorithms typically converge only to local stationary points. This limitation calls for a convex objective that retains the metric property while producing a meaningful transport plan.

In this study, we propose *Convex Distance Operator Transport (CDOT)*, the first convex geometry-aware OT framework for aligning heterogeneous mm spaces via an operator-level formulation that encodes intrinsic geometric structure.

*Table 1.* Comparison of CDOT to existing methods. **Domain** groups methods by the underlying space of interest. **Plan** denotes whether the algorithm explicitly outputs a coupling/matching. **Theoretical Guarantees** summarize (pseudo-)metricity, convexity of the underlying optimization, and whether asymptotic consistency and finite-sample guarantees are available on the stated domain.

| Domain | Method | Plan | Feature Info. | Time | Theoretical Guarantees | | | |
| --- | --- | --- | --- | --- | --- | --- | --- | --- |
| | | | | | Metric | Convexity | Consistency | Finite |
| Metric-measure (mm) spaces | **CDOT (Ours)** | ✓ | ✓ | $\mathcal{O}(n^3)$ | ✓ | ✓ | ✓ | ✓ |
| | GW (Mémoli, 2011) | ✓ | ✗ | $\mathcal{O}(n^3)$ | ✓ | ✗ | △ | ✗ |
| | FGW (Vayer et al., 2020) | ✓ | ✓ | $\mathcal{O}(n^3)$ | △ | ✗ | △ | △ |
| | Entropic GW (Peyré et al., 2016) | ✓ | ✗ | $\mathcal{O}(n^2)$ | ✗ | △ | ✗ | ✗ |
| Euclidean | Sliced GW (Titouan et al., 2019) | ✗ | ✗ | $\mathcal{O}(n \log n)$ | ✓ | ✗ | ✗ | ✗ |
| Discrete mm spaces | Low-rank GW (Scetbon et al., 2022) | ✓ | ✗ | $\mathcal{O}(n)$ | ✗ | ✗ | ✗ | ✗ |
| | GW-SDP (Chen et al., 2024) | ✓ | ✗ | $\mathcal{O}(n^6)$ | ✗ | ✓ | ✗ | ✗ |
| | Sliced FGW (Piening & Beinert, 2025) | ✗ | ✓ | $\mathcal{O}(n^2 \log n)$ | ✓ | ✗ | ✗ | ✗ |
| Graphs | Spectral (Leordeanu & Hebert, 2005) | △ | ✗ | $\mathcal{O}(n^3)$ | ✗ | ✗ | ✗ | ✗ |
| | IsoRank (Singh et al., 2008) | ✓ | ✓ | $\mathcal{O}(n^3)$ | ✗ | ✗ | ✗ | ✗ |
| | COPT (Dong & Sawin, 2020) | ✓ | ✗ | $\mathcal{O}(n^{2.373})$ | ✓ | ✗ | ✗ | ✗ |

*Legend:* ✓: established guarantee on the stated domain. △: related statistical results exist, but they control the empirical risk/cost relative to its population counterpart rather than the population risk attained by the returned plan/estimator itself. ✗: no explicit guarantee in the cited reference. **Domain**: Metric-measure (mm) spaces: population-level mm spaces. Euclidean: standard Euclidean spaces $\mathbb{R}^d$. Discrete mm spaces: discrete/empirical mm spaces. Graphs: finite graphs. **Plan**: outputs an explicit coupling/matching. **Feature Info.**: uses feature information as data. **Time**: computational complexity. **Metric**: refers to (pseudo-)metricity on the stated domain. **Convexity**: convexity of the objective. **Consistency**: existence of an explicit asymptotic guarantee for the population risk attained by the returned plan/estimator on the stated domain; for CDOT, this is the risk-consistency result in Corollary 5.7. **Finite**: existence of an explicit finite-sample bound for the population risk attained by the returned plan/estimator on the stated domain.

Figure 1 contrasts two cyclic graphs, showing that (F)GW distinguishes them by enforcing rigid consistency between individual pairwise distances, whereas CDOT aligns aggregated distance profiles through operator-based regularization and thus identifies the two structures as equivalent despite differing node cardinalities. This relaxation attenuates structural noise and yields a convex optimization landscape.

The advantages of the proposed framework, as summarized in Table 1, can be highlighted as follows:

- **Transport plan.** Unlike slicing-based approaches that focus on scalable discrepancy estimation, CDOT explicitly recovers an optimal transport plan, enabling direct node correspondences between domains.

- **Convexity.** In contrast to existing approaches defined on population-level mm spaces, CDOT admits a convex optimization problem in its original, unregularized formulation, so global optimization does not depend on introducing an additional smoothing parameter such as entropic regularization.

- **Theoretical guarantees.** We establish that CDOT induces a pseudometric on the space of attributed compact mm spaces and provide a statistical theory, including non-asymptotic risk bounds and consistency, properties often absent in existing works.

- **Computational tractability.** CDOT admits an efficient implementation with complexity $\mathcal{O}(n^3)$, which is comparable to standard FGW and substantially lower than semidefinite relaxations $\mathcal{O}(n^6)$.

In addition, we rigorously compare CDOT with the GW objective through a *dispersion gap* analysis. Specifically, we decompose the non-convex GW cost into our convex CDOT objective and a concave dispersion penalty. This analysis reveals that the non-convexity of GW stems from this dispersion term which favors deterministic plans, whereas our approach achieves convexity by filtering out this effect.

We also substantiate our theoretical results through synthetic 2D point clouds and demonstrate the practicality of CDOT on real brain connectome and graph classification datasets.

The remainder of this paper is organized as follows. Section 2 fixes notation and reviews related work. Section 3 introduces the CDOT objective, establishes its convexity and pseudometric property, and quantifies its gap to the GW objective via a dispersion decomposition. Section 4 presents the empirical quadratic program and its optimization algorithm. Section 5 establishes the risk consistency of our estimator. Section 6 validates our algorithm empirically using both synthetic and real-world data. Finally, Section 7 concludes with a summary of our findings and discusses future directions.

**Conflict of Interest Disclosure.** The authors declare that they have no competing interests.

## 2. Preliminaries

### 2.1. Notations

Let $(\Omega, \mathcal{F}, \mathbb{P})$ be a probability space, and let $(\mathcal{X}, d_{\mathcal{X}})$, $(\mathcal{Y}, d_{\mathcal{Y}})$ be compact metric spaces. This guarantees that

both spaces are Polish so that regular conditional distributions exist (Lemma F.1). Measurable maps $X : \Omega \to \mathcal{X}$ and $Y : \Omega \to \mathcal{Y}$ are called random elements, with distributions $\mathbb{P}_X := \mathbb{P} \circ X^{-1}$ and $\mathbb{P}_Y := \mathbb{P} \circ Y^{-1}$, respectively. For simplicity, $\mathbb{P}_X$ and $\mathbb{P}_Y$ are fully supported. Then $(\mathcal{X}, d_\mathcal{X}, \mathbb{P}_X)$ and $(\mathcal{Y}, d_\mathcal{Y}, \mathbb{P}_Y)$ form compact metric-measure (mm) spaces. Denote by $L^2(\mathcal{X}, \mathbb{P}_X)$ the Hilbert space of real-valued, square-integrable functions on $\mathcal{X}$ with respect to $\mathbb{P}_X$. We define $L^2(\mathcal{Y}, \mathbb{P}_Y)$ analogously. We further introduce a compact feature space $\mathcal{M} \subset \mathbb{R}^k$ and define continuous mappings $f_\mathcal{X} : \mathcal{X} \to \mathcal{M}$ and $f_\mathcal{Y} : \mathcal{Y} \to \mathcal{M}$, referred to as feature functions. We define an attributed compact mm space as a tuple $\mathfrak{X} := (\mathcal{X}, d_\mathcal{X}, \mathbb{P}_X, f_\mathcal{X})$ and define $\mathfrak{Y} := (\mathcal{Y}, d_\mathcal{Y}, \mathbb{P}_Y, f_\mathcal{Y})$ similarly. For brevity, we assume that $\operatorname{diam}(\mathcal{X}) = \operatorname{diam}(\mathcal{Y}) = \operatorname{diam}(\mathcal{M}) = 1$, where $\operatorname{diam}(\mathcal{A}) := \sup_{a,a' \in \mathcal{A}} d_\mathcal{A}(a, a')$ denotes the diameter. For probability measures $\mu$ and $\nu$, denote by $\Pi(\mu, \nu)$ the set of couplings whose marginals are $\mu$ and $\nu$.

## 2.2. Related Work

The GW discrepancy (Mémoli, 2011) and its attributed variant, FGW (Vayer et al., 2020), serve as canonical frameworks for aligning heterogeneous structures:

$$\min_{\pi} (1 - \alpha)\mathbb{E}_\pi[c_f(X, Y)] + \alpha \mathcal{R}_{\mathrm{GW},2}(\pi),$$

where the minimum is taken over the set of valid couplings $\Pi(\mathbb{P}_X, \mathbb{P}_Y)$, $c_f(x, y) := \|f_\mathcal{X}(x) - f_\mathcal{Y}(y)\|_2^2$ is the squared Euclidean feature cost, and the quadratic GW cost is given by $\mathcal{R}_{\mathrm{GW},2}(\pi) := \mathbb{E}_{\pi \otimes \pi}[|d_\mathcal{X}(X, X') - d_\mathcal{Y}(Y, Y')|^2]$.

A fundamental limitation of (F)GW is the non-convexity arising from the tensor product $\pi \otimes \pi$ in $\mathcal{R}_{\mathrm{GW},2}$. As shown in Table 1, several works have attempted to tackle this issue, but few works have fully resolved the non-convexity and provided rigorous metric properties on mm spaces. A more comprehensive review is provided in Appendix B.

## 3. Proposed Method

### 3.1. CDOT: Convex Distance Operator Transport

The goal of CDOT is to align distributions on heterogeneous compact mm spaces by jointly matching features and intrinsic structures through a transport plan $\pi \in \Pi(\mathbb{P}_X, \mathbb{P}_Y)$. A fundamental challenge arises from the distinct metrics equipped on each space, which precludes direct comparison across domains. CDOT addresses this issue by lifting (i) the geometric structure of each space and (ii) the transport plan into linear operators acting on $L^2$ spaces, so that structural alignment can be measured through operator compositions.

**Definition 3.1** (Distance operator). Let $(\mathcal{X}, d_\mathcal{X}, \mathbb{P}_X)$ be a compact mm space. The *distance operator* $D_{\mathbb{P}_X}$ is an integral operator defined on $L^2(\mathcal{X}, \mathbb{P}_X)$ by

$$(D_{\mathbb{P}_X} f)(x) := \int_\mathcal{X} d_\mathcal{X}(x, x') f(x') \mathbb{P}_X(dx'),$$
$$\forall f \in L^2(\mathcal{X}, \mathbb{P}_X), \forall x \in \mathcal{X}.$$

The distance operator can be viewed as a continuous analogue of the pairwise distance matrix. For an empirical measure $\hat{\mathbb{P}}_X$ supported on $n$ distinct samples, the operator admits a matrix representation $[D_{\hat{\mathbb{P}}_X}]_{ij} = d_\mathcal{X}(X_i, X_j)/n$ (see Appendix G.1 for the derivation).

**Definition 3.2** (Conditional expectation operator). For a coupling $\pi \in \Pi(\mathbb{P}_X, \mathbb{P}_Y)$, the *conditional expectation operator* $T_\pi : L^2(\mathcal{Y}, \mathbb{P}_Y) \to L^2(\mathcal{X}, \mathbb{P}_X)$ is defined as

$$(T_\pi g)(x) := \int_\mathcal{Y} g(y) \pi(dy \mid x), \ \forall g \in L^2(\mathcal{Y}, \mathbb{P}_Y), \forall x \in \mathcal{X},$$

where $\pi(dy \mid x)$ denotes a Markov kernel.

Analogously, in finite-sample settings, $T_\pi$ reduces to the scaled transport matrix $[T_\pi]_{ij} = n\pi_{ij}$.

Leveraging these operator representations, the CDOT objective seeks a coupling that harmonizes both the feature representations and the geometric structures of the two spaces.

**Definition 3.3** (CDOT). Given two attributed compact mm spaces $\mathfrak{X}, \mathfrak{Y}$, and a fusion weight parameter $0 \leq \alpha \leq 1$, CDOT finds a transport plan $\pi \in \Pi(\mathbb{P}_X, \mathbb{P}_Y)$ by balancing the following two components:

**Feature:** $\mathcal{F}(\pi) := \mathbb{E}_\pi[c_f(X, Y)],$

**Structure:** $\mathcal{R}(\pi) := \|D_{\mathbb{P}_X} T_\pi - T_\pi D_{\mathbb{P}_Y}\|_{\mathrm{HS}}^2,$

**CDOT objective:** $\mathcal{L}_\alpha(\pi) := (1 - \alpha)\mathcal{F}(\pi) + \alpha/2\mathcal{R}(\pi),$

where $\|\cdot\|_{\mathrm{HS}}$ is the Hilbert–Schmidt (HS) norm. The *CDOT problem* is then

$$\inf_{\pi \in \Pi(\mathbb{P}_X, \mathbb{P}_Y)} \mathcal{L}_\alpha(\pi; \mathfrak{X}, \mathfrak{Y}).$$

The feature alignment term $\mathcal{F}(\pi)$ encourages element-wise similarity between features, whereas the structural regularization term $\mathcal{R}(\pi)$ penalizes structural discrepancy. Specifically, $\mathcal{R}(\pi)$ measures the extent to which the conditional expectation operator $T_\pi$ fails to intertwine the intrinsic distance operators of the two spaces (i.e., the failure of the commutativity relation $D_{\mathbb{P}_X} T_\pi \approx T_\pi D_{\mathbb{P}_Y}$).

The well-definedness of $\mathcal{R}(\pi)$ can be verified through its integral representation (see Appendix G.2 for more details):

$$\mathcal{R}(\pi; \mathfrak{X}, \mathfrak{Y}) = \int_\mathcal{Y} \int_\mathcal{X} \Gamma_\pi(x, y)^2 \, \mathbb{P}_X(dx)\mathbb{P}_Y(dy), \quad (1)$$

where $\Gamma_\pi^{(1)}(x,y) := \mathbb{E}_\pi[d_{\mathcal{X}}(x,X) \mid Y = y]$, $\Gamma_\pi^{(2)}(x,y) := \mathbb{E}_\pi[d_{\mathcal{Y}}(y,Y) \mid X = x]$, and $\Gamma_\pi = \Gamma_\pi^{(1)} - \Gamma_\pi^{(2)}$. Since the underlying spaces are compact, $|\Gamma_\pi(x,y)|$ is bounded, ensuring the finiteness of the integral.

**Theorem 3.4** (Convex optimization). *Given two attributed compact mm spaces $\mathfrak{X}$, $\mathfrak{Y}$, and a fusion weight parameter $0 \le \alpha \le 1$, the CDOT problem satisfies the following:*

**(Existence).** *There exists $\pi^* \in \Pi(\mathbb{P}_X, \mathbb{P}_Y)$ such that*

$$\mathcal{L}_\alpha(\pi^*; \mathfrak{X}, \mathfrak{Y}) = \inf_{\pi \in \Pi(\mathbb{P}_X, \mathbb{P}_Y)} \mathcal{L}_\alpha(\pi; \mathfrak{X}, \mathfrak{Y}).$$

**(Convexity).** *The CDOT objective is convex.*

*Proof.* See Appendix G.3. □

Theorem 3.4 confirms that the CDOT problem is a well-posed convex optimization problem. The existence of an optimal coupling is guaranteed by the weak compactness of the transport polytope $\Pi(\mathbb{P}_X, \mathbb{P}_Y)$, the lower semicontinuity of the objective, and the generalized Weierstrass theorem (Lemma F.11). Convexity arises from the linearity of $\mathcal{F}(\pi)$ and the convexity of the squared HS norm in $\mathcal{R}(\pi)$.

### 3.2. Geometric Interpretation: Pseudometricity

Beyond its optimization-theoretic properties, CDOT also serves as a discrepancy between attributed compact mm spaces. We define the CDOT discrepancy as the square root of the objective, which satisfies the pseudometric property on the space of attributed compact mm spaces.

**Theorem 3.5** (Pseudometric). *For each $0 \le \alpha \le 1$, define the CDOT discrepancy between two attributed compact mm spaces $\mathfrak{X}$ and $\mathfrak{Y}$ as*

$$d_{\mathrm{CT}}^{(\alpha)}(\mathfrak{X}, \mathfrak{Y}) := \left( \min_{\pi \in \Pi(\mathbb{P}_X, \mathbb{P}_Y)} \mathcal{L}_\alpha(\pi; \mathfrak{X}, \mathfrak{Y}) \right)^{1/2}.$$

*Then, $d_{\mathrm{CT}}^{(\alpha)}$ is a pseudometric on the space of attributed compact mm spaces.*

*Proof.* See Appendix G.4. □

Theorem 3.5 confirms that $d_{\mathrm{CT}}^{(\alpha)}$ is a valid pseudometric. The failure to be a genuine metric stems from the fact that $d_{\mathrm{CT}}^{(\alpha)}(\mathfrak{X}, \mathfrak{Y}) = 0$ does not necessarily imply $\mathfrak{X} = \mathfrak{Y}$; instead, it indicates the existence of a coupling that preserves both feature mappings and intrinsic distance operators. This operator-level equivalence is strictly weaker than the measure-preserving isometric identification underlying GW, and is conceptually closer to a fractional notion of structural equivalence (Grebík & Rocha, 2022). A complete

characterization of the zero-discrepancy class of CDOT in terms of such relaxed equivalences is left for future work.

The advantage of CDOT compared to GW-type methods is illustrated in Figure 1. While FGW enforces consistency at the level of individual pairwise distances and thus depends critically on pointwise correspondences between nodes, CDOT compares geometric structures through aggregated distance profiles, formalized as conditional expectations of distances. In this example, although the two spaces differ in the cardinality of nodes, their aggregated distance profiles coincide under a suitable coupling. As a result, CDOT assigns zero discrepancy, reflecting the alignment of local geometric structure, whereas GW-type methods incur a positive cost due to unavoidable pairwise mismatches.

### 3.3. CDOT vs. Gromov–Wasserstein

This section clarifies the structural difference between CDOT and FGW by isolating their geometry terms. Since the feature-alignment component is shared by CDOT and FGW, we focus exclusively on comparing our $\mathcal{R}(\pi)$ with the GW objective. To facilitate this comparison, we introduce a new functional, termed *dispersion*, which quantifies the spatial uncertainty induced by a coupling.

**Definition 3.6** (Dispersion). *The dispersion functional $\mathcal{V} : \Pi(\mathbb{P}_X, \mathbb{P}_Y) \to \mathbb{R}$ is defined by*

$$\mathcal{V}(\pi) := \iint \Big( \mathrm{Var}_\pi[d_{\mathcal{X}}(x,X) \mid Y = y] \\ + \mathrm{Var}_\pi[d_{\mathcal{Y}}(y,Y) \mid X = x] \Big) \mathbb{P}_X(dx) \mathbb{P}_Y(dy).$$

**Theorem 3.7** (Dispersion gap). *For any $\pi \in \Pi(\mathbb{P}_X, \mathbb{P}_Y)$,*

$$\mathcal{R}_{\mathrm{GW},2}(\pi) - \mathcal{R}(\pi) = \mathcal{V}(\pi).$$

*Proof.* See Appendix G.5. □

Theorem 3.7 reveals that the GW objective penalizes not only structural distortion, as captured by $\mathcal{R}(\pi)$, but also the spatial uncertainty of the coupling through the dispersion term $\mathcal{V}(\pi)$. In contrast, the CDOT formulation deliberately isolates the structural component by discarding this additional penalty, a design choice that underlies the differing geometric and optimization behavior of the two methods. As a side effect, the dispersion term in GW favors concentrated, nearly deterministic couplings, whereas the absence of this term in CDOT tends to admit more diffuse optimal plans, so applications that require a hard correspondence may proceed via the post-processing step in Section 4.

Figure 2 compares the level sets of $\mathcal{R}(\pi)$ and $\mathcal{R}_{\mathrm{GW},2}(\pi)$ on the joint range of the mapping $\pi \mapsto (\mathcal{R}(\pi), \mathcal{V}(\pi))$. The independent coupling $\pi_{\mathrm{ind}} = \mathbb{P}_X \otimes \mathbb{P}_Y$ exhibits maximal

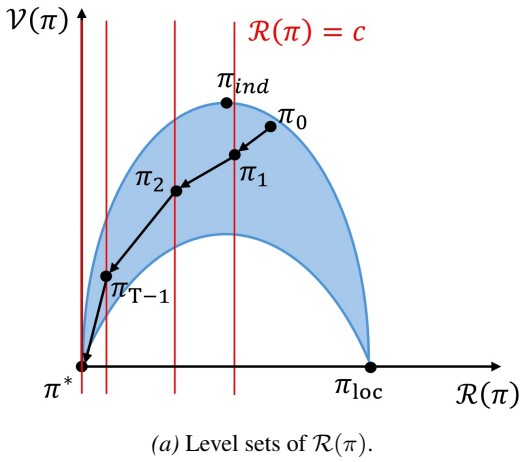

*(a) Level sets of $\mathcal{R}(\pi)$.*

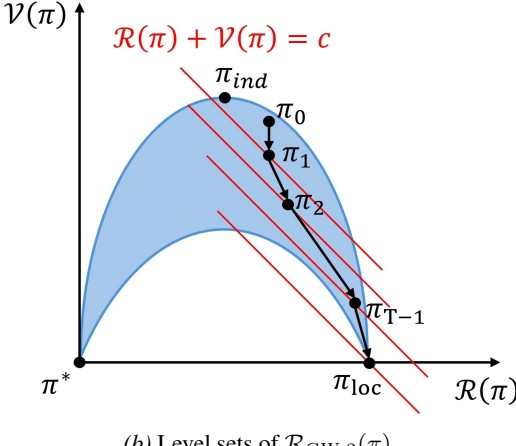

*(b) Level sets of $\mathcal{R}_{\mathrm{GW},2}(\pi)$.*

*Figure 2.* Comparison of level sets. The blue region represents the range of $\pi \mapsto (\mathcal{R}(\pi), \mathcal{V}(\pi))$, and red lines indicate the level sets of the respective objectives. $\pi_{\mathrm{ind}} := \mathbb{P}_X \otimes \mathbb{P}_Y$ denotes the independent coupling. (a) Our formulation yields vertical level sets, guiding the optimization trajectory (black arrows) from initialization $\pi_0$ directly toward the global optimum $\pi^*$. (b) In contrast, the GW objective induces diagonal level sets, causing the optimization trapped in a local optimum $\pi_{\mathrm{loc}}$.

dispersion, while bi-deterministic couplings lie on the zero-dispersion axis. The attainable range typically forms a non-convex, boomerang-like region, reflecting the trade-off between structural distortion and dispersion. Since our objective minimizes only $\mathcal{R}(\pi)$, its level sets are vertical ($x = c$), guiding descent-type methods toward the global optimum $\pi^*$ regardless of initialization. In contrast, the GW objective minimizes $\mathcal{R}(\pi) + \mathcal{V}(\pi)$, yielding diagonal level sets ($x + y = c$). These level sets can interact with the non-convex region and lead to suboptimal points. A numerical realization of the blue region is provided in Appendix C.

## 4. Finite-Sample Estimation

This section presents the empirical formulation of the CDOT objective, leading to a convex quadratic program over a polytope. We solve this problem using the Frank–Wolfe algorithm, with an optional post-processing step to extract a hard matching from the resulting soft transport plan.

Let $X_1, ..., X_{n_X} \in \mathcal{X}$ and $Y_1, ..., Y_{n_Y} \in \mathcal{Y}$ be distinct samples, with associated features $f_{\mathcal{X}}(X_i)$ and $f_{\mathcal{Y}}(Y_j)$. We define the empirical probability measures by

$$\hat{\mathbb{P}}_X := \frac{1}{n_X} \sum_{i=1}^{n_X} \delta_{X_i}, \quad \hat{\mathbb{P}}_Y := \frac{1}{n_Y} \sum_{j=1}^{n_Y} \delta_{Y_j},$$

where $\delta_x$ is the Dirac measure at $x$. With a slight abuse of notation, we identify the set of couplings $\Pi(\hat{\mathbb{P}}_X, \hat{\mathbb{P}}_Y)$ with the polytope of $n_X \times n_Y$ transport matrices. Under this identification, any coupling $\pi$ is treated as a matrix with entries $\pi_{ij} = \pi(\{X_i\} \times \{Y_j\})$.

The empirical counterparts of the components in Definition 3.3 can be naturally derived by plugging in the empirical probability measures. To this end, we first define the cost and distance matrices. Let $C_f \in \mathbb{R}^{n_X \times n_Y}$ be the

feature cost matrix with $[C_f]_{ij} = \|f_{\mathcal{X}}(X_i) - f_{\mathcal{Y}}(Y_j)\|_2^2$. Similarly, let $D_{\hat{\mathbb{P}}_X} \in \mathbb{R}^{n_X \times n_X}$ and $D_{\hat{\mathbb{P}}_Y} \in \mathbb{R}^{n_Y \times n_Y}$ be the normalized distance matrices defined by $[D_{\hat{\mathbb{P}}_X}]_{ij} = d_{\mathcal{X}}(X_i, X_j)/n_X$ and $[D_{\hat{\mathbb{P}}_Y}]_{ij} = d_{\mathcal{Y}}(Y_i, Y_j)/n_Y$. Substituting these into Definition 3.3, we obtain the following empirical counterparts:

**Empirical feature:** $\hat{\mathcal{F}}(\pi) := \langle C_f, \pi \rangle_F = \mathrm{Tr}(C_f^\top \pi),$

**Empirical structure:** $\hat{\mathcal{R}}(\pi) := n_X n_Y \| D_{\hat{\mathbb{P}}_X} \pi - \pi D_{\hat{\mathbb{P}}_Y} \|_F^2,$

**Empirical objective:** $\hat{\mathcal{L}}_\alpha(\pi) := (1-\alpha)\hat{\mathcal{F}}(\pi) + \alpha/2\hat{\mathcal{R}}(\pi).$

Consequently, the empirical CDOT problem is a convex quadratic program over the transport polytope:

$$\min_{\pi \in \Pi(\hat{\mathbb{P}}_X, \hat{\mathbb{P}}_Y)} \hat{\mathcal{L}}_\alpha(\pi; \mathfrak{X}, \mathfrak{Y}). \tag{2}$$

Since (2) is a convex program, a wide range of convex solvers can be applied (Boyd & Vandenberghe, 2004). In this paper, we adopt the Frank–Wolfe (FW) algorithm (Frank & Wolfe, 1956). The FW algorithm is well-suited to this setting due to its projection-free nature. At each iteration, it requires solving a linear minimization problem over $\Pi(\hat{\mathbb{P}}_X, \hat{\mathbb{P}}_Y)$, avoiding projections onto the transport polytope. This subproblem is often computationally simpler than the full projection, making the FW algorithm an attractive choice.

Appendix D presents two algorithms: the standard FW procedure (Algorithm 1) and an efficient variant termed lazy gradient FW (Algorithm 2). The latter leverages the quadratic structure of the objective to efficiently update gradients, reducing the cost from cubic to quadratic time. Although the overall per-iteration complexity remains $\mathcal{O}(n^3)$

dominated by the linear minimization problem, this strategy significantly lowers the practical computational overhead.

The estimated coupling $\hat{\pi}$ is generally a soft transport plan, i.e., a fractional coupling over $\Pi(\hat{\mathbb{P}}_X, \hat{\mathbb{P}}_Y)$. In applications that require a deterministic matching, we project the soft plan onto the set of permutation matrices via the linear assignment problem (LAP):

$$\hat{P} := \arg\max_{P \in \mathcal{P}_n} \mathrm{Tr}(P^\top \hat{\pi}), \tag{3}$$

where $\mathcal{P}_n$ is the set of permutation matrices when $n_X = n_Y = n$. This LAP can be efficiently solved by the Hungarian algorithm (Kuhn, 1955). When $n_X \neq n_Y$, an analogous rectangular assignment problem can be solved instead.

## 5. Risk Consistency

This section establishes the risk consistency of the estimator $\hat{\pi}$. Our analysis relies on Wasserstein-1 distances to control the discrepancy between empirical and population measures.

**Definition 5.1** (Wasserstein-$p$ distance). Let $(\mathcal{S}, d_{\mathcal{S}})$ be a compact metric space. For Borel probability measures $\mu$ and $\nu$ on $\mathcal{S}$, the Wasserstein-$p$ distance ($1 \leq p < \infty$) is

$$W_p^{d_{\mathcal{S}}}(\mu, \nu) := \left( \inf_{\pi \in \Pi(\mu, \nu)} \int_{\mathcal{S} \times \mathcal{S}} d_{\mathcal{S}}(x, y)^p \, \pi(dx, dy) \right)^{1/p}.$$

Since $d_{\mathcal{S}}^p$ is continuous and $\mathcal{S}$ is compact, the infimum is always attained (see Appendix F.2 for details).

Below are some technical assumptions required to derive the precise convergence rate for our proposed method.

**Assumption 5.2** (No atoms). $\mathbb{P}_X$ and $\mathbb{P}_Y$ are non-atomic; $\mathbb{P}_X(\{x\}) = \mathbb{P}_Y(\{y\}) = 0$ for all $x \in \mathcal{X}$ and $y \in \mathcal{Y}$.

**Assumption 5.3** (No ties). For any distinct $x_1, x_2 \in \mathcal{X}$, $y_1, y_2 \in \mathcal{Y}$, and for any $c \in \mathbb{R}$,

$$\mathbb{P}_X(\{x \in \mathcal{X} : d_{\mathcal{X}}(x, x_1) - d_{\mathcal{X}}(x, x_2) = c\}) = 0,$$
$$\mathbb{P}_Y(\{y \in \mathcal{Y} : d_{\mathcal{Y}}(y, y_1) - d_{\mathcal{Y}}(y, y_2) = c\}) = 0.$$

Assumptions 5.2 and 5.3 rule out degeneracies such as ties and mass concentration. In particular, these conditions prevent ambiguous assignments and discontinuities caused by boundary events, which allows empirical and population-level objectives to be compared in a stable manner.

**Assumption 5.4** (Lipschitzness). The feature functions $f_{\mathcal{X}}$ and $f_{\mathcal{Y}}$ are $L_f$-Lipschitz: for all $x_1, x_2 \in \mathcal{X}$ and $y_1, y_2 \in \mathcal{Y}$,

$$\|f_{\mathcal{X}}(x_1) - f_{\mathcal{X}}(x_2)\|_2 \leq L_f \, d_{\mathcal{X}}(x_1, x_2),$$
$$\|f_{\mathcal{Y}}(y_1) - f_{\mathcal{Y}}(y_2)\|_2 \leq L_f \, d_{\mathcal{Y}}(y_1, y_2).$$

Assumption 5.4 ensures that variations in the feature space $\mathcal{M}$ are controlled by the metric structures of $\mathcal{X}$ and $\mathcal{Y}$.

**Assumption 5.5** (Stability). For each $0 \leq \alpha \leq 1$, there exists an optimal coupling $\pi_\alpha^* \in \arg\min_{\pi \in \Pi(\mathbb{P}_X, \mathbb{P}_Y)} \mathcal{L}_\alpha(\pi)$ such that for some $L_W > 0$,

$$W_1^{d_{\mathcal{X}}}(\pi_\alpha^*(\cdot \mid y), \pi_\alpha^*(\cdot \mid y')) \leq L_W \, d_{\mathcal{Y}}(y, y'), \ \forall y, y' \in \mathcal{Y},$$
$$W_1^{d_{\mathcal{Y}}}(\pi_\alpha^*(\cdot \mid x), \pi_\alpha^*(\cdot \mid x')) \leq L_W \, d_{\mathcal{X}}(x, x'), \ \forall x, x' \in \mathcal{X}.$$

Assumption 5.5 is a kernel-regularity condition that prevents the conditional distributions of an optimal coupling from oscillating severely; analogous stability assumptions have been widely adopted in the OT literature (Deb et al., 2021; Hütter & Rigollet, 2021; Manole et al., 2024). A simple geometric example is the case where the optimal coupling is induced by a bi-Lipschitz Monge map. In addition, the assumption holds whenever the conditional laws of $\pi_\alpha^*$ are Lipschitz in total variation, via the standard bound $W_1 \leq \mathrm{diam}(\cdot) \| \cdot \|_{\mathrm{TV}}$; in the discrete setting, this condition is directly checkable from the rows and columns of the optimal coupling matrix.

We are now ready to present the main consistency results.

**Theorem 5.6** (Deterministic error bound). *Consider the fixed empirical measures $\hat{\mathbb{P}}_X$ and $\hat{\mathbb{P}}_Y$ supported on distinct points. Let $\hat{\pi}$ be the output of FW Algorithm 1 with $0 \leq \alpha \leq 1$, the iteration count $T$, and step size $\gamma_t = 2/(t + 2)$. In addition, suppose that Assumptions 5.2 to 5.5 hold. Let $Q_X \in \Pi(\mathbb{P}_X, \hat{\mathbb{P}}_X)$ and $Q_Y \in \Pi(\mathbb{P}_Y, \hat{\mathbb{P}}_Y)$ denote the optimal couplings minimizing $W_1^{d_{\mathcal{X}}}$ and $W_1^{d_{\mathcal{Y}}}$. We define the glued measure $\Phi_n : \Pi(\hat{\mathbb{P}}_X, \hat{\mathbb{P}}_Y) \to \Pi(\mathbb{P}_X, \mathbb{P}_Y)$ by*

$$\Phi_n(\pi)(dx, dy) := \int Q_X(dx \mid \hat{x}) \, Q_Y(dy \mid \hat{y}) \, \pi(d\hat{x}, d\hat{y}).$$

*Then, there exists a constant $C > 0$ such that*

$$\left| \mathcal{L}_\alpha(\Phi_n(\hat{\pi}); \mathfrak{X}, \mathfrak{Y}) - \min_{\pi \in \Pi(\mathbb{P}_X, \mathbb{P}_Y)} \mathcal{L}_\alpha(\pi; \mathfrak{X}, \mathfrak{Y}) \right|$$
$$\leq \underbrace{\frac{32\alpha \, n_{\min}}{T + 3}}_{\text{Optimization error}} + C \underbrace{\left( W_1^{d_{\mathcal{X}}}(\mathbb{P}_X, \hat{\mathbb{P}}_X) + W_1^{d_{\mathcal{Y}}}(\mathbb{P}_Y, \hat{\mathbb{P}}_Y) \right)}_{\text{Statistical error}},$$

*where $n_{\min} = \min\{n_X, n_Y\}$.*

*Proof.* See Appendix G.6. □

Theorem 5.6 demonstrates that the total error naturally decomposes into two distinct sources. The first is the *optimization error*, arising from the finite number of iterations in the FW algorithm; as expected, this term decays at the rate of $O(1/T)$. The second is the *statistical error*, which reflects the intrinsic uncertainty incurred by replacing the true distributions $\mathbb{P}_X$ and $\mathbb{P}_Y$ with their empirical counterparts. We note that the construction of the glued measure $\Phi_n$ relies on the disintegration of optimal couplings $Q_X$ and $Q_Y$; we

refer the reader to Lemma F.5 for details. Consequently, the statistical bound is governed by the convergence rates of the empirical measures, depending essentially on the geometry of the underlying sample spaces.

The glued measure $\Phi_n$ lifts the discrete solution $\hat{\pi}$ to the continuous population space with minimal transport cost. We can quantify the spatial displacement incurred by this lifting using Markov's inequality:

$$\mathbb{P}_{(X,\hat{X}) \sim Q_X}(d_{\mathcal{X}}(X, \hat{X}) > \varepsilon) \leq \varepsilon^{-1} W_1^{d_{\mathcal{X}}}(\mathbb{P}_X, \hat{\mathbb{P}}_X).$$

Therefore, provided that the empirical measures converge in $W_1$, the blurring effect introduced by lifting $\hat{\pi}$ to the continuous space $\Pi(\mathbb{P}_X, \mathbb{P}_Y)$ vanishes asymptotically.

**Corollary 5.7** (Risk consistency). *Suppose the assumptions in Theorem 5.6 hold. Assume further that the datasets consist of i.i.d. samples drawn from $\mathbb{P}_X$ and $\mathbb{P}_Y$, respectively. Consider sequences of sample sizes $n_X, n_Y \to \infty$ and iteration count $T_n$ such that $n_{\min} T_n^{-1} \to 0$. Then, $\Phi_n(\hat{\pi})$ asymptotically achieves a global optimum:*

$$\mathcal{L}_\alpha(\Phi_n(\hat{\pi}); \mathfrak{X}, \mathfrak{Y}) \xrightarrow{a.s.} \min_{\pi \in \Pi(\mathbb{P}_X, \mathbb{P}_Y)} \mathcal{L}_\alpha(\pi; \mathfrak{X}, \mathfrak{Y}).$$

The established error bound remains valid up to a constant factor when the population objective $\mathcal{L}_\alpha(\Phi_n(\hat{\pi}); \mathfrak{X}, \mathfrak{Y})$ is replaced by the empirical objective $\hat{\mathcal{L}}_\alpha(\hat{\pi}; \mathfrak{X}, \mathfrak{Y})$ in Theorem 5.6 and Corollary 5.7. Therefore, the finite-sample optimal transport cost itself is a consistent estimator of the population optimal cost, i.e., $\hat{\mathcal{L}}_\alpha(\hat{\pi}) \to \min_{\pi \in \Pi(\mathbb{P}_X, \mathbb{P}_Y)} \mathcal{L}_\alpha(\pi)$.

The deterministic bound in Theorem 5.6 holds on arbitrary compact mm spaces, but the resulting sample-size rate depends on the geometry of the underlying domain through the empirical Wasserstein terms. For i.i.d. samples on a compact subset of $\mathbb{R}^d$, the statistical error scales at the standard rate $O(n_{\min}^{-1/d})$ for $d \geq 3$ (Fournier & Guillin, 2015; Weed & Bach, 2019), with faster rates in low dimensions. In infinite-dimensional settings, however, such polynomial rates are not guaranteed without further structural assumptions on the underlying measures.

Finally, the choice of iteration count $T$ is briefly discussed. While the optimization error bound in Theorem 5.6 depends on $n_{\min}$ due to worst-case control of operator norms, a moderate number of iterations (e.g., $T = 50, 100, 200$) is sufficient to reduce the error to a negligible level in practice.

## 6. Empirical Validation

### 6.1. Synthetic Data Analysis

This section empirically validates the theoretical results using synthetic 2D clustered point clouds. Our primary focus is to assess optimal solution recovery and to verify the consistency of the estimated transport plan by increasing the

sample size. We first define the underlying space as $[0, 2]^2$, partitioned into four unit-square regions $\{\mathcal{R}_k\}_{k=1}^4$:

$$\mathcal{R}_1 = [0, 1)^2, \quad \mathcal{R}_2 = [1, 2] \times [0, 1),$$
$$\mathcal{R}_3 = [0, 1) \times [1, 2], \quad \mathcal{R}_4 = [1, 2]^2.$$

We equip this space with the Euclidean metric and a feature function $f : [0, 2]^2 \to \{1, ..., 4\}$ defined by $f(x) = k$ if $x \in \mathcal{R}_k$. This constructs the attributed mm spaces:

$$\mathfrak{X} = \mathfrak{Y} = \left([0, 2]^2, \|\cdot\|_2, U([0, 2]^2), f\right),$$

where $U([0, 2]^2)$ denotes the uniform distribution on $[0, 2]^2$.

We sample $n$ points uniformly from each region $\mathcal{R}_k$, resulting in a total sample size of $N = 4n$. The feature cost matrix $C_f$ is defined by the 0-1 cluster mismatch penalty:

$$[C_f]_{ij} = \min\{1, |f(x_i) - f(y_j)|\}.$$

We compare CDOT with five baseline algorithms: 1) FGW, 2) Entropic FGW (EFGW), 3) IsoRank, 4) Spectral Matching (Spectral), and 5) Coordinated Optimal Transport (COPT). Detailed configurations are provided in Appendix H. Since Spectral and COPT do not incorporate feature information, they are independent of the fusion weight $\alpha$. Distance matrices are normalized by their maximum values in all experiments. For EFGW, we report the best performance obtained at $\varepsilon = 10^{-3}$ over $\{10^{-4}, ..., 10^{-1}\}$. We vary the cluster size $n \in \{100, ..., 500\}$ with a fixed $\alpha = 0.5$ for CDOT, FGW, EFGW, and IsoRank. Results are averaged over 100 independent trials with random initialization, and the number of iterations is set to $T = 200$. Performance is evaluated using the mean squared error (MSE) between the source $X$ and the transported target $\hat{Y} = N\hat{\pi}Y$.

Table 2 summarizes the simulation results. CDOT consistently achieves the lowest MSE across all sample sizes, and the error decreases as $n$ increases. These trends empirically support the design of the CDOT objective, suggesting that the operator-based structural term provides a stable and effective geometric alignment signal when combined with feature matching. EFGW yields results comparable to CDOT, likely due to the entropic regularization; as the entropic regularization smooths the GW objective and partially mitigates issues arising from the dispersion term. IsoRank also shows comparable performance to CDOT, which is expected given the underlying isomorphism between the spaces. In contrast, Spectral and COPT exhibit higher MSEs, as they rely purely on structural information without leveraging the features.

### 6.2. Brain Connectome Analysis

The proposed CDOT is evaluated on the OASIS-3 brain connectome dataset (Kerepesi et al., 2017), which comprises structural brain networks from 696 subjects. Each network

*Table 2.* Comparison of MSE (mean ± std) over 100 trials. CDOT is compared against FGW, Entropic FGW (EFGW), IsoRank, Spectral Matching (Spectral), and COPT. The best MSE for each $n$ is highlighted in bold.

| $n = N/4$ | CDOT (Ours) | FGW | EFGW | IsoRank | Spectral | COPT |
|---|---|---|---|---|---|---|
| 100 | **0.0077** ± 0.00 | 0.0146 ± 0.00 | 0.0098 ± 0.00 | 0.0141 ± 0.00 | 1.3447 ± 0.07 | 0.6664 ± 0.02 |
| 200 | **0.0040** ± 0.00 | 0.0081 ± 0.00 | 0.0054 ± 0.00 | 0.0078 ± 0.00 | 1.3429 ± 0.06 | 0.6691 ± 0.01 |
| 300 | **0.0027** ± 0.00 | 0.0055 ± 0.00 | 0.0038 ± 0.00 | 0.0053 ± 0.00 | 1.3276 ± 0.04 | 0.6670 ± 0.01 |
| 400 | **0.0020** ± 0.00 | 0.0043 ± 0.00 | 0.0031 ± 0.00 | 0.0041 ± 0.00 | 1.3355 ± 0.04 | 0.6682 ± 0.01 |
| 500 | **0.0016** ± 0.00 | 0.0034 ± 0.00 | 0.0025 ± 0.00 | 0.0033 ± 0.00 | 1.3373 ± 0.03 | 0.6670 ± 0.01 |

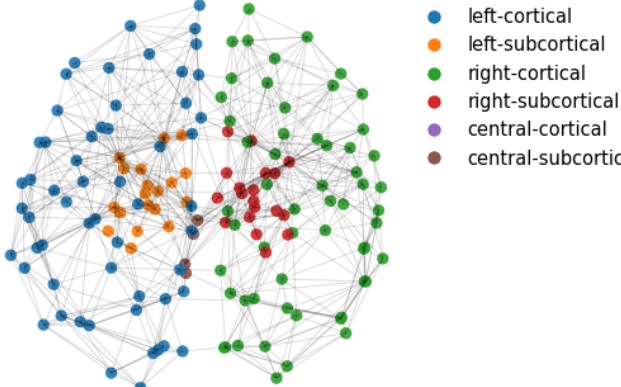

- left-cortical
- left-subcortical
- right-cortical
- right-subcortical
- central-cortical
- central-subcortical

*Figure 3.* Visualization of an OASIS-3 brain connectome (170 nodes). Nodes are colored according to six anatomical region labels, corresponding to left, right, and central regions, each subdivided into cortical and subcortical areas.

consists of 170 nodes annotated by six anatomical labels (see Figure 3), with weighted edges representing anatomical connectivity. The connectivity matrices are used only to define the graph geometry, while the six anatomical categories serve as auxiliary node features. We perform node correspondence across subjects: since all subjects share a common atlas, node $i$ in one subject and node $i$ in another subject represent the same anatomical region of interest, so the ground-truth matching is the identity permutation. Pairwise matching is performed between the first 100 subjects ($_{100}C_2$). Each method outputs a soft coupling or similarity matrix, which we convert to a hard one-to-one matching via the linear assignment problem; accuracy is reported as the fraction of nodes matched to their correct atlas index, averaged over subject pairs.

We use categorical labels representing hemispheric and cortical status as features, and we consider two distinct metrics: geodesic distance and diffusion distance. CDOT is compared against OT-based methods (FGW, EFGW) and topology-based baselines (Spectral, IsoRank, and COPT), where topology-based methods utilize graph adjacency matrices as inputs. Analogously to the synthetic setting, we use the fusion weight $\alpha = 0.5$ for CDOT, FGW, EFGW, and IsoRank. Further details are provided in Appendix H.3.

The results in Table 3 demonstrate CDOT's capacity to uti-

*Table 3.* Comparison of matching accuracy on OASIS-3. The best value for each structure representation is highlighted in bold.

| | Structure Representation | | |
|---|---|---|---|
| Method | Diffusion | Geodesic | Topology |
| CDOT (Ours) | **0.6136** | 0.4640 | – |
| FGW | 0.1853 | **0.5375** | – |
| EFGW | 0.4097 | 0.4583 | – |
| IsoRank | – | – | **0.4055** |
| Spectral | – | – | 0.0737 |
| COPT | – | – | 0.0253 |

lize robust geometric information. While FGW shows a marginal edge under the geodesic distance due to its direct minimization of pairwise distortions, CDOT achieves a lead when utilizing diffusion distance. This performance disparity stems from the fundamental difference in geometric perception. Unlike the brittle geodesic distance, which relies on single shortest paths vulnerable to noise, diffusion distance averages over all paths to capture robust global connectivity. CDOT effectively leverages this rich geometric information. Conversely, FGW struggles because diffusion smooths out local metric details; this reduced variance diminishes the distinct pairwise contrasts required by the GW objective for precise alignment.

Performance among topology-based baselines is mixed. Spectral and COPT yield poor accuracy, confirming that relying solely on adjacency structure is insufficient. Conversely, IsoRank achieves a competitive accuracy by incorporating node features. However, it still falls short of OT-based methods. This indicates that while integrating feature signals is crucial, IsoRank's reliance on local neighborhood similarity is less effective than OT frameworks.

### 6.3. Graph Classification

Beyond the matching application, we assess the capability of CDOT to serve as a metric for graph classification tasks. We conduct experiments on standard benchmark datasets from TUDataset (Morris et al., 2020), covering both bioinformatics (MUTAG, PROTEINS, NCI1, ENZYMES) and social network (IMDB-BINARY). Specifically, the bioinformatics tasks involve predicting molecular properties or biological

*Table 4.* Comparison of classification accuracy (mean ± std) across different graph datasets. We compare CDOT against FGW, GW, and COPT methods. The best accuracy for each dataset is highlighted in bold.

| Dataset | Avg. Nodes | Node Info. | CDOT (Ours) Acc. (mean ± std) | FGW Acc. (mean ± std) | GW Acc. (mean ± std) | COPT Acc. (mean ± std) |
|---|---|---|---|---|---|---|
| MUTAG | 17.93 | Label | **0.8617** ± 0.07 | 0.8249 ± 0.08 | 0.7175 ± 0.10 | 0.6330 ± 0.04 |
| IMDB-B | 19.77 | None | **0.6420** ± 0.04 | 0.6020 ± 0.07 | – | 0.6370 ± 0.03 |
| PROTEINS | 39.06 | Attr. | **0.7547** ± 0.03 | 0.7358 ± 0.03 | 0.6612 ± 0.03 | 0.6954 ± 0.04 |
| NCI1 | 29.87 | Label | **0.7477** ± 0.01 | 0.7302 ± 0.01 | 0.5708 ± 0.02 | 0.5993 ± 0.02 |
| ENZYMES | 32.63 | Attr. | **0.5133** ± 0.04 | 0.4450 ± 0.03 | 0.2383 ± 0.07 | 0.2350 ± 0.06 |

functions, whereas the social network task classifies actor collaboration graphs by movie genre.

We adopt a kernel-based classification framework using a support vector machine (SVM). For each method, we compute the pairwise distance matrix $D$ between all graphs in the dataset to construct a Gaussian RBF kernel $K = \exp(-\gamma D^2)$. We employ a nested cross-validation scheme to ensure a robust evaluation. The outer loop performs 10-fold stratified cross-validation to estimate accuracy. The inner loop (5-fold) performs a grid search to select optimal hyperparameters: the SVM regularization parameter $C \in \{10^{-1}, ..., 10^2\}$ and the kernel bandwidth $\gamma \in \{10^{-3}, ..., 10\}$. For fused approaches (CDOT, FGW), we also tune the fusion weight $\alpha \in 0.25 \times \{0, ..., 4\}$. Input graphs are represented by their geodesic distance matrices to capture global geometry. For node features, we utilize the provided labels or attributes. However, for IMDB-BINARY, lacking node features, we rely solely on structural geometry; in such cases, the fusion weight is fixed at $\alpha = 1$, and it is excluded from the hyperparameter tuning procedure.

Table 4 summarizes the classification performance. CDOT consistently outperforms the baseline methods across diverse datasets. The performance gap is particularly notable in datasets rich in feature information, such as MUTAG and ENZYMES, where CDOT achieves significantly higher accuracy than purely structural methods (GW and COPT). This highlights the efficacy of our method in fusing feature signals with geometric structure. Notably, even in IMDB-BINARY where no node features are available ($\alpha = 1$), CDOT maintains the best performance solely based on structural information. This suggests that the proposed operator geometry captures topological distinctions more effectively than the standard GW approaches. We emphasize that while specialized neural architectures may achieve state-of-the-art results on these benchmarks, CDOT demonstrates superior performance compared to existing OT baselines.

## 7. Discussion

This work introduces *Convex Distance Operator Transport (CDOT)*, a novel geometry-aware convex framework that aligns heterogeneous mm spaces via an operator-based formulation. By aggregating geometric information at the operator level, CDOT filters out the geometric dispersion term, the source of non-convexity in GW approaches, which ensures a convex optimization landscape while effectively capturing geometric information. Theoretically, we show that the resulting discrepancy serves as a valid pseudometric between attributed compact mm spaces and establish the risk consistency of our estimator through non-asymptotic risk bounds. Experiments on synthetic and real-world data empirically demonstrate the efficacy of our method.

We also note two limitations. First, CDOT is designed for operator-level alignment rather than nearly one-to-one matching, since zero discrepancy does not necessarily imply exact measure-preserving isometry; GW and FGW may therefore be more appropriate when rigid pointwise matching is the primary goal. Second, although CDOT admits a convex formulation, each Frank–Wolfe iteration is still dominated by a standard OT subproblem, so the current implementation is better suited to moderate problem sizes than to very large-scale settings.

Several avenues remain for future investigation. A particularly promising direction is the development of a statistical inference framework for mm spaces. Since the CDOT discrepancy constitutes a valid pseudometric, it naturally lends itself to statistical inference on mm spaces. Developing hypothesis testing frameworks, such as isomorphism tests based on the CDOT discrepancy, would be an interesting future work. Furthermore, while this study requires that the metric structure (e.g., geodesic or diffusion) be predefined, one could jointly learn the optimal ground metric or feature representation that minimizes the CDOT discrepancy by integrating the CDOT objective as a differentiable layer within deep neural networks. We expect that this direction bridges the gap between geometric optimal transport and representation learning.

## Impact Statement

This paper presents a methodological contribution to optimal transport and structured data analysis. Its potential

benefits include more reliable alignment and comparison of heterogeneous graphs or metric-measure spaces, with applications in scientific domains such as biology, neuroscience, and network analysis. Since CDOT is a general-purpose tool, its societal impact depends on the downstream setting; possible risks include privacy-sensitive graph matching or biased conclusions when applied to human-related network data. We do not foresee direct negative societal impacts beyond those common to machine learning methods, but deployment in sensitive domains should be accompanied by appropriate validation, privacy safeguards, and domain-specific oversight.

## Acknowledgments

This work was supported by the National Research Foundation of Korea (NRF) grant funded by the Korea government (MSIT) (No. RS-2026-25473737), the Institute of Information & Communications Technology Planning & Evaluation (IITP) grant funded by the Korea government (MSIT) (No. RS-2021-II211343), the Global-LAMP Program of the National Research Foundation of Korea (NRF) grant funded by the Ministry of Education (No. RS-2023-00301976), and the National Research Foundation of Korea (NRF) grant funded by the Korea government (MSIT) (No. RS-2026-25494850).

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

# A. Notations

We first introduce some notations used throughout the Appendix. Let $(\Omega, \mathcal{F}, \mathbb{P})$ be a probability space, and let $(\mathcal{X}, d_\mathcal{X})$ and $(\mathcal{Y}, d_\mathcal{Y})$ be compact metric spaces. We denote by $\mathcal{B}(\mathcal{X})$ the Borel $\sigma$-algebra on $\mathcal{X}$. The compactness guarantees that both spaces are Polish, ensuring that regular conditional distributions exist (Lemma F.1).

For a set $\mathcal{S}$, we denote by $\mathbb{I}_A$ the indicator function of a subset $A \subset \mathcal{S}$. For a metric space $(\mathcal{S}, d_\mathcal{S})$, let $C(\mathcal{S})$ be the space of continuous functions and $C_b(\mathcal{S})$ be the space of bounded continuous functions from $\mathcal{S}$ to $\mathbb{R}$. Note that since $\mathcal{X}$ and $\mathcal{Y}$ are compact, $C(\mathcal{X}) = C_b(\mathcal{X})$ and $C(\mathcal{Y}) = C_b(\mathcal{Y})$. For a measure $\mu$ on $\mathcal{S}$ and $p \in [1, \infty)$, let $L^p(\mathcal{S}, \mu)$ denote the Banach space of measurable functions $f : \mathcal{S} \to \mathbb{R}$ such that $\int_\mathcal{S} |f|^p d\mu < \infty$. When the domain is clear from the context, we abbreviate this as $L^p(\mu)$. Specifically, $L^2(\mu)$ forms a Hilbert space equipped with the inner product $\langle f, g \rangle_{L^2(\mu)} = \int f g d\mu$.

We denote by $\mathcal{P}(\mathcal{X})$ the set of all Borel probability measures on $\mathcal{X}$. For $x \in \mathcal{X}$, $\delta_x$ denotes the Dirac measure concentrated at $x$. Measurable maps $X : \Omega \to \mathcal{X}$ and $Y : \Omega \to \mathcal{Y}$ are called random elements taking values in $\mathcal{X}$ and $\mathcal{Y}$, with distributions $\mathbb{P}_X := \mathbb{P} \circ X^{-1}$ and $\mathbb{P}_Y := \mathbb{P} \circ Y^{-1}$, respectively. For simplicity, we assume $\mathbb{P}_X$ and $\mathbb{P}_Y$ are fully supported. Then $(\mathcal{X}, d_\mathcal{X}, \mathbb{P}_X)$ and $(\mathcal{Y}, d_\mathcal{Y}, \mathbb{P}_Y)$ form compact metric-measure (mm) spaces. We say a measurable map $T : \mathcal{X} \to \mathcal{Y}$ pushes forward $\mu$ to $\nu$ if $\mu(T^{-1}(A)) = \nu(A)$ for all $A \in \mathcal{B}(\mathcal{Y})$. We write $T_\# \mu = \nu$ if $T$ pushes forward $\mu$ to $\nu$. For a subset $A \in \mathcal{B}(\mathcal{X})$, the restriction of $\mu$ to $A$ is denoted by $\mu\lfloor_A$, defined as $\mu\lfloor_A(B) = \mu(A \cap B)$.

For probability measures $\mu_1, ..., \mu_m$, denote by

$$\Pi(\mu_1, ..., \mu_m) := \{\pi : \text{the marginals of } \pi \text{ are } \mu_1, ..., \mu_m\}$$

the set of all couplings among them. We further introduce a compact feature space $\mathcal{M} \subset \mathbb{R}^k$ and define continuous mappings $f_\mathcal{X} : \mathcal{X} \to \mathcal{M}$ and $f_\mathcal{Y} : \mathcal{Y} \to \mathcal{M}$, referred to as feature functions. With these components, we define an attributed compact mm space as a tuple $\mathfrak{X} := (\mathcal{X}, d_\mathcal{X}, \mathbb{P}_X, f_\mathcal{X})$. We define $\mathfrak{Y} := (\mathcal{Y}, d_\mathcal{Y}, \mathbb{P}_Y, f_\mathcal{Y})$ analogously. Throughout this paper, we assume that $\mathrm{diam}(\mathcal{X}) = \mathrm{diam}(\mathcal{Y}) = \mathrm{diam}(\mathcal{M}) = 1$, where $\mathrm{diam}(\mathcal{A}) := \sup_{a,a' \in \mathcal{A}} d_\mathcal{A}(a, a')$ denotes the diameter.

# B. Related Work

This section reviews representative approaches for aligning structured objects modeled as mm spaces and graphs. We begin with the definition of the Gromov–Wasserstein (GW) discrepancy, a fundamental framework for comparing metric structures. This is immediately followed by the fused Gromov–Wasserstein (FGW) formulation, which extends GW to incorporate auxiliary feature information alongside structural data. Given that both objectives involve a quadratic dependence on the coupling, we subsequently discuss strategies to address the inherent non-convexity of these problems, including entropic regularization schemes and convex relaxations. The review then proceeds to scalability-oriented approximations designed for large-scale applications, such as sliced and low-rank variants. Finally, we cover graph-specific optimal transport formulations, specifically Coordinated Optimal Transport (COPT), and summarize classical non-OT baselines such as spectral and random-walk-based network alignment methods.

For two compact mm spaces $(\mathcal{X}, d_\mathcal{X}, \mathbb{P}_X)$ and $(\mathcal{Y}, d_\mathcal{Y}, \mathbb{P}_Y)$, the $p$-th order GW (GW-$p$) objective $\mathcal{R}_{\mathrm{GW},p} : \Pi(\mathbb{P}_X, \mathbb{P}_Y) \to \mathbb{R}$ with $1 \le p < \infty$ is defined by

$$\mathcal{R}_{\mathrm{GW},p}(\pi) := \mathbb{E}_{\pi \otimes \pi}\left[\left|d_\mathcal{X}(X, X') - d_\mathcal{Y}(Y, Y')\right|^p\right] = \iint \left|d_\mathcal{X}(x, x') - d_\mathcal{Y}(y, y')\right|^p \pi(dx, dy)\pi(dx', dy').$$

The GW-$p$ discrepancy is obtained by minimizing $(\mathcal{R}_{\mathrm{GW},p}(\pi))^{1/p}$ over $\pi \in \Pi(\mathbb{P}_X, \mathbb{P}_Y)$, vanishing if and only if the two spaces are measure-preserving isometric. However, this formulation involves a tensor product of the coupling (i.e., $\pi \otimes \pi$), rendering the optimization landscape inherently non-convex due to its quadratic dependence on $\pi$. While GW-$p$ is defined for general $p$, most works focus on the quadratic case ($p = 2$) (Mémoli, 2011; Peyré et al., 2016; Rioux et al., 2024; Chen et al., 2024) due to its theoretical clarity. In the sequel, references to the GW objective imply $p = 2$ unless otherwise specified.

In many applications, points in mm spaces are equipped with auxiliary attributes alongside their structural information. The FGW framework (Vayer et al., 2020) addresses this by interpolating between a feature-based transport cost and the structural GW discrepancy. Given two attributed compact mm spaces $\mathfrak{X}$ and $\mathfrak{Y}$ coupled with a feature cost $c(\cdot, \cdot)$, the FGW objective $\mathcal{L}_{\mathrm{FGW},\alpha} : \Pi(\mathbb{P}_X, \mathbb{P}_Y) \to \mathbb{R}$ is defined by

$$\mathcal{L}_{\mathrm{FGW},\alpha}(\pi; \mathfrak{X}, \mathfrak{Y}) := (1 - \alpha)\, \mathbb{E}_\pi[c(X, Y)] + \alpha\, \mathcal{R}_{\mathrm{GW},2}(\pi),$$

where the parameter $0 \leq \alpha \leq 1$ regulates the trade-off between feature matching and structural alignment. Although Vayer et al. (2020) define FGW in a more general setting and establish its pseudometric properties in special cases, we focus here on the quadratic formulation. Consequently, the FGW objective inherits the non-convex nature of the GW objective $\mathcal{R}_{\text{GW},2}$.

Weak optimal transport provides another nonlinear variant of optimal transport, where the transportation cost may depend on the conditional law induced by a coupling (Backhoff-Veraguas & Pammer, 2022). This perspective is related to our formulation in that CDOT also depends on conditional transport mechanisms rather than only on a pointwise cost. However, standard weak OT typically considers costs depending on the forward conditional law $\pi(\cdot \mid x)$, whereas our structural term involves both directions of the coupling through conditional expectations with respect to $\pi(\cdot \mid x)$ and $\pi(\cdot \mid y)$. Thus, we regard weak OT as a conceptually related but distinct framework, rather than as a direct baseline for structured alignment.

A substantial literature aims to mitigate the non-convexity of (F)GW, either by proposing scalable solvers with convergence to stationary points or by constructing convex surrogates. For instance, Peyré et al. (2016) and Solomon et al. (2016) introduce entropic regularization together with projected-gradient or Sinkhorn-type schemes, which partially mitigates the non-convexity issue and improves numerical stability. Notably, Rioux et al. (2024) demonstrate that under sufficiently strong regularization, the dual formulation of the entropic GW objective becomes globally convex, admitting theoretical guarantees for global optimality. However, outside this specific high-regularization regime, the optimization landscape remains generally non-convex, with solvers typically converging only to stationary points. Furthermore, a limitation of these entropic formulations is that they generally fail to preserve the pseudometric properties on the space of attributed compact mm spaces. Finally, its performance is often highly sensitive to the choice of the regularization parameter, which necessitates careful tuning to balance numerical stability with approximation fidelity.

To mitigate the computational burden of GW-type objectives, several scalable approximations have been proposed. A prominent direction is a slicing-based approach, exemplified by Sliced GW (Titouan et al., 2019), which approximates the discrepancy by averaging costs computed over randomized one-dimensional projections. While this formulation offers significant scalability for large-scale Euclidean point clouds, it inherently requires the underlying spaces to possess a Euclidean structure and typically estimates the distance value rather than explicitly retrieving the transport coupling.

Recently, Piening & Beinert (2025) extend this paradigm to the fused setting (Sliced FGW), incorporating attribute information to handle structured domains efficiently. However, like its predecessors, it primarily targets fast discrepancy estimation via sliced projections and therefore does not directly output a globally consistent coupling or permutation, often necessitating auxiliary post-processing when explicit node correspondences are required. Moreover, its fidelity to the underlying FGW objective depends on approximation choices (e.g., the number and placement of quantile samples and random projections), introducing a trade-off between computational efficiency and the tightness of the approximation.

Alternatively, structural restrictions on the optimization variables offer another path to scalability. Scetbon et al. (2022) introduce a low-rank GW framework that restricts the admissible couplings and cost matrices to low-rank factorizations. This approach achieves linear time and memory complexity, making it applicable to massive datasets with millions of points. Nevertheless, its effectiveness is contingent upon the assumption that the intrinsic geometry of the data and their optimal alignment can be faithfully captured by a low-rank representation, which may not hold for all complex topological structures.

Chen et al. (2024) propose a semidefinite programming relaxation of the GW objective (GW-SDP), yielding a convex lower bound that can be optimized in a lifted space and providing a certificate to assess near-optimality. Despite these appealing guarantees, the relaxation can incur substantial computational overhead, limiting applicability to small-sized problems. Moreover, while the relaxation is provably tight in several structured settings, its broader theoretical behavior remains largely unexplored.

Beyond GW/FGW, COPT (Dong & Sawin, 2020) provides an OT-based approach to graph alignment by coupling vertex matching with the alignment of graph-induced random signal distributions. COPT equips each graph with a Gaussian measure over graph signals and seeks a vertex coupling that is consistent with a transport mechanism acting on these signal distributions. This perspective emphasizes global structural alignment through signal statistics, rather than relying on an explicit feature cost matrix. However, this framework requires the assumption of Gaussian signal distributions and lacks a direct mechanism to incorporate node attributes. A formal definition of the COPT objective is provided in Appendix H.

When the objects are finite graphs, matching is represented by a permutation matrix $P$ (or a soft assignment matrix $\pi$ after relaxation). In this discrete regime, matching by comparing adjacency matrices naturally leads to quadratic assignment–type objectives, linking GW-style structure matching to the classical graph matching literature (Zaslavskiy et al., 2008; Ravikumar & Lafferty, 2006; Aflalo et al., 2015; Schellewald et al., 2007; Schellewald & Schnörr, 2005; Swoboda et al.,

2019). For example, given adjacency matrices $A_1$ and $A_2$ of graphs $G_1$ and $G_2$, the classical graph matching problem seeks a permutation matrix $P$ that minimizes

$$\|A_1 - PA_2P^\top\|_F^2,$$

which is equivalent to $\|A_1P - PA_2\|_F^2$. A common convex relaxation replaces $P$ by a soft assignment matrix $\pi$ constrained to lie in the Birkhoff polytope (i.e., the set of doubly stochastic matrices), yielding the convex quadratic program

$$\min_{\pi \in \mathcal{B}} \|A_1\pi - \pi A_2\|_F^2,$$

where $\mathcal{B}$ denotes the Birkhoff polytope. Such relaxations are widely used to obtain tractable surrogates of combinatorial matching objectives and to provide principled initializations for non-convex solvers.

Classical baselines for matching and network alignment include spectral methods and random-walk-based similarity propagation. Spectral Matching (Spectral) (Leordeanu & Hebert, 2005) embeds nodes using eigenvectors of adjacency or Laplacian-derived matrices and then solve an assignment problem based on distances between spectral embeddings. This typically yields a continuous score first, and a hard matching is obtained only after an additional discretization or assignment step (e.g., solving an LAP). In network alignment, IsoRank (Singh et al., 2008) iteratively propagates similarity scores over the Kronecker product of two graphs, based on the principle that two nodes are similar if their neighbors are similar. These approaches are typically computationally efficient and conceptually simple, but their performance can be sensitive to symmetries, spectral ambiguities, or local neighborhood distortions, depending on the underlying graph geometry.

## C. Numerical Realization of $\pi \mapsto (\mathcal{R}(\pi), \mathcal{V}(\pi))$

We empirically project the coupling space $\Pi(\mathbb{P}_X, \mathbb{P}_Y)$ onto the two-dimensional plane spanned by the structural regularization $\mathcal{R}(\pi)$ and the dispersion penalty $\mathcal{V}(\pi)$. We consider a synthetic matching problem between two Euclidean spaces with $n = 4$ points, constructed to be isomorphic (measure-preserving isometric) up to a rigid transformation.

The resulting landscape is illustrated in Figure 4. The horizontal and vertical axes correspond to the structural regularization $\mathcal{R}(\pi)$ and the dispersion penalty $\mathcal{V}(\pi)$, respectively. The gray scatter points represent transport plans $\pi$ randomly sampled from the interior of the transport polytope ($\Pi(\mathbb{P}_X, \mathbb{P}_Y)$) using Dirichlet mixtures. The black circular points denote the permutation matrices (extreme points), while the red triangle marks the independent coupling ($\mathbb{P}_X \otimes \mathbb{P}_Y$). Notably, the projected points form a distinctive non-convex, boomerang-like region, empirically validating the schematic illustration discussed in Section 3.3. This shape highlights the geometry of the coupling space: the independent coupling sits at the peak with maximal dispersion, while the feasible region bifurcates and descends towards the zero-dispersion axis where the bi-deterministic permutation matrices reside.

## D. Algorithms

In this appendix, we provide detailed pseudocodes for the algorithms discussed in Section 4 and present a comparative analysis of their computational complexity. Importantly, we note that these algorithms are mathematically equivalent.

### D.1. Standard FW Algorithm

Algorithm 1 outlines the standard Frank–Wolfe (FW) procedure. In each iteration, it computes the full gradient of the objective function and solves a linear minimization subproblem over the transport polytope.

The procedure begins by constructing the feature cost matrix $C_f$ and the normalized distance matrices $D_{\hat{\mathbb{P}}_X}$ and $D_{\hat{\mathbb{P}}_Y}$ based on the input data (lines 3–5). The main optimization loop (lines 6–10) iteratively refines the transport plan $\pi$. In each iteration $t$, we first compute the full gradient of the objective function, $\nabla\hat{\mathcal{L}}_\alpha(\pi^{(t)})$, at the current iterate $\pi^{(t)}$ (line 7). Next, we solve the linear minimization oracle (LMO) in line 8; this step corresponds to a standard linear optimal transport problem where the cost matrix is given by the computed gradient. This yields the update direction $\mu^{(t)}$, which is a vertex of the transport polytope. The new iterate $\pi^{(t+1)}$ is then obtained by taking a convex combination of the current plan $\pi^{(t)}$ and the direction $\mu^{(t)}$ using a step size $\gamma_t$ (line 9). After $T$ iterations, the algorithm outputs the final soft coupling $\hat{\pi}$ (line 11). Finally, line 12 describes the optional post-processing step, which solves the linear assignment problem (LAP) to project the soft coupling $\hat{\pi}$ onto the set of deterministic assignments $\mathcal{P}$, yielding the hard assignment $\hat{P}$.

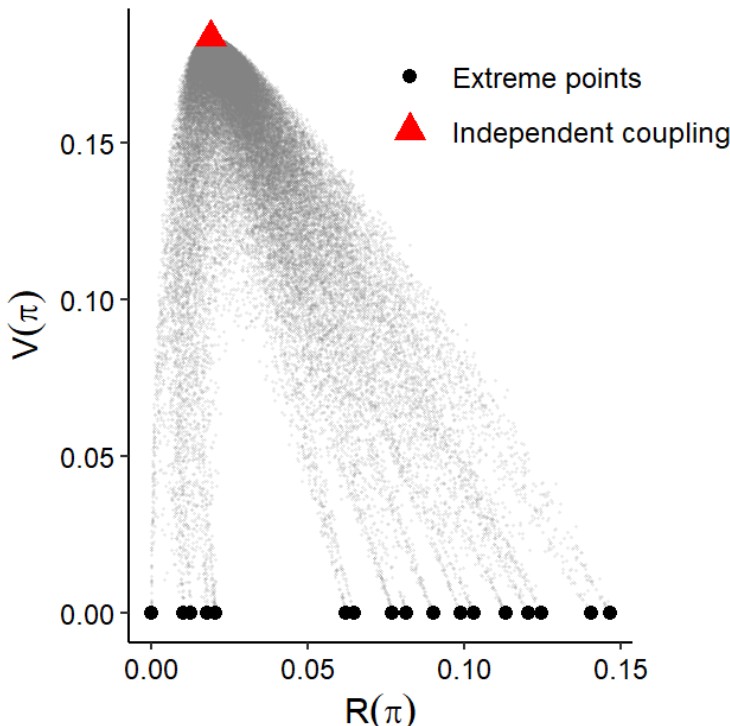

*Figure 4.* Empirical visualization of the optimization landscape. The scatter plot illustrates the image of the transport polytope projected onto the plane spanned by the structural regularization $\mathcal{R}(\pi)$ ($x$-axis) and the dispersion penalty $\mathcal{V}(\pi)$ ($y$-axis) for an isomorphic matching problem with $n = 4$. Black circles represent permutation matrices (extreme points), the red triangle marks the independent coupling $(\mathbb{P}_X \otimes \mathbb{P}_Y)$, and gray points denote random transport plans sampled from the interior of the polytope.

### D.2. Lazy Gradient FW Algorithm

The standard FW algorithm typically requires recomputing the full gradient $\nabla \hat{\mathcal{L}}_\alpha(\pi^{(t)})$ at every iteration $t$:

$$\nabla \hat{\mathcal{L}}_\alpha(\pi^{(t)}) = (1 - \alpha)C_f + \alpha n_X n_Y \left( D_{\hat{\mathbb{P}}_X}^\top (D_{\hat{\mathbb{P}}_X} \pi^{(t)} - \pi^{(t)} D_{\hat{\mathbb{P}}_Y}) - (D_{\hat{\mathbb{P}}_X} \pi^{(t)} - \pi^{(t)} D_{\hat{\mathbb{P}}_Y}) D_{\hat{\mathbb{P}}_Y}^\top \right).$$

For the proposed objective function, the structural term involves matrix multiplications (e.g., $D_{\hat{\mathbb{P}}_X}^\top (D_{\hat{\mathbb{P}}_X} \pi^{(t)} - \pi^{(t)} D_{\hat{\mathbb{P}}_Y})$), which can be computationally intensive to recalculate from scratch.

However, we observe that the proposed objective function $\hat{\mathcal{L}}_\alpha(\pi)$ is quadratic with respect to the transport plan $\pi$. Consequently, the gradient mapping $\pi \mapsto \nabla \hat{\mathcal{L}}_\alpha(\pi)$ is affine. Another key observation is that the FW algorithm updates the transport plan via a convex combination, i.e.,

$$\pi^{(t+1)} = (1 - \gamma_t)\pi^{(t)} + \gamma_t \mu^{(t)}, \quad \text{where} \quad \mu^{(t)} = \underset{\mu \in \Pi(\hat{\mathbb{P}}_X, \hat{\mathbb{P}}_Y)}{\arg\min} \ \mathrm{Tr}\left( \nabla \hat{\mathcal{L}}_\alpha(\pi^{(t)})^\top \mu \right).$$

Thus, the gradient at the next iteration satisfies the linearity property:

$$\underbrace{\nabla \hat{\mathcal{L}}_\alpha(\pi^{(t+1)})}_{=G^{(t+1)}} = \nabla \hat{\mathcal{L}}_\alpha \big( (1 - \gamma_t)\pi^{(t)} + \gamma_t \mu^{(t)} \big) = (1 - \gamma_t) \underbrace{\nabla \hat{\mathcal{L}}_\alpha(\pi^{(t)})}_{=G^{(t)}} + \gamma_t \nabla \hat{\mathcal{L}}_\alpha(\mu^{(t)}).$$

Algorithm 2 exploits this property by maintaining the gradient matrix $G^{(t)} \coloneqq \nabla \hat{\mathcal{L}}_\alpha(\pi^{(t)})$ and performing a lazy update using the gradient at the LMO solution $\nabla \hat{\mathcal{L}}_\alpha(\mu^{(t)})$. This approach allows for efficient in-place updates of the gradient matrix without redundant matrix operations involving the dense coupling $\pi^{(t+1)}$.

---

**Algorithm 1** Convex Distance Operator Transport Plan via FW and Optional LAP

---

1: **Input:** Source data $\{(X_i, f_{\mathcal{X}}(X_i))\}_{i=1}^{n_X}$, target data $\{(Y_j, f_{\mathcal{Y}}(Y_j))\}_{j=1}^{n_Y}$, fusion weight parameter $0 \leq \alpha \leq 1$, iteration count $T$, initial coupling $\pi^{(0)}$

2: Construct matrices:

3: $[C_f]_{ij} \leftarrow \|f_{\mathcal{X}}(X_i) - f_{\mathcal{Y}}(Y_j)\|_2^2$

4: $[D_{\hat{\mathbb{P}}_X}]_{ii'} \leftarrow d_{\mathcal{X}}(X_i, X_{i'})/n_X$

5: $[D_{\hat{\mathbb{P}}_Y}]_{jj'} \leftarrow d_{\mathcal{Y}}(Y_j, Y_{j'})/n_Y$

6: **for** $t = 0$ **to** $T - 1$ **do**

7:     Calculate the gradient $\nabla \hat{\mathcal{L}}_\alpha(\pi^{(t)})$

8:     $\mu^{(t)} \leftarrow \arg\min_{\mu \in \Pi(\hat{\mathbb{P}}_X, \hat{\mathbb{P}}_Y)} \text{Tr}\big(\nabla \hat{\mathcal{L}}_\alpha(\pi^{(t)})^\top \mu\big)$

9:     $\pi^{(t+1)} \leftarrow (1 - \gamma_t)\pi^{(t)} + \gamma_t \mu^{(t)}$ for some $0 < \gamma_t \leq 1$

10: **end for**

11: $\hat{\pi} \leftarrow \pi^{(T)}$

12: **Optional (LAP):** $\hat{P} \leftarrow \arg\max_{P \in \mathcal{P}} \text{Tr}(P^\top \hat{\pi})$

13: **Return:** $\hat{\pi}$ (and optionally $\hat{P}$)

---

## D.3. Computational Complexity

We compare the computational complexity of the two algorithms. Assuming each distance and feature evaluation is $\mathcal{O}(1)$, both algorithms share the same initialization cost of $\mathcal{O}(n_X^2 + n_Y^2 + n_X n_Y)$ to construct $C_f$, $D_{\hat{\mathbb{P}}_X}$, and $D_{\hat{\mathbb{P}}_Y}$. By defining $n_{\max} := \max\{n_X, n_Y\}$, the initialization cost scales as $\mathcal{O}(n_{\max}^2)$.

Both Algorithm 1 (line 8) and Algorithm 2 (line 9) require solving the same linear minimization problem over the transport polytope at each iteration. This subproblem is an optimal transport problem and its complexity depends on the chosen solver. For example, using an exact network-simplex solver (e.g., `emd` from (Flamary et al., 2021)) incurs a worst-case complexity $\mathcal{O}(n_{\max}^3)$. Since this step is identical for both methods, it dictates the worst-case bound for the entire procedure $\mathcal{O}(T n_{\max}^3)$.

The advantage of Algorithm 2 becomes evident in the gradient computation step.

- **Standard FW (Algorithm 1):** Computing the gradient in line 7 requires dense matrix multiplications involving the current coupling $\pi^{(t)}$ (e.g., $D_{\hat{\mathbb{P}}_X} \pi^{(t)}$). Since $\pi^{(t)}$ is generally dense, these operations incur a complexity of $\mathcal{O}(n_{\max}^3)$.

- **Lazy FW (Algorithm 2):** Instead of recomputing the gradient from the dense $\pi^{(t)}$, the algorithm computes lazy terms involving the LMO solution $\mu^{(t)}$ in line 12 (e.g., $D_{\hat{\mathbb{P}}_X} \mu^{(t)}$). Since $\mu^{(t)}$ is a vertex of the transport polytope, it is extremely sparse with at most $n_X + n_Y - 1$ non-zero entries. Consequently, these matrix products involve only sparse operations, reducing the cost to $\mathcal{O}(n_X(n_X + n_Y) + n_Y(n_X + n_Y)) = \mathcal{O}((n_X + n_Y)^2)$, which scales as $\mathcal{O}(n_{\max}^2)$. The gradient matrix $G^{(t)}$ is then updated via a simple linear combination (line 13), which is also $\mathcal{O}(n_{\max}^2)$.

In summary, while the solver complexity makes the asymptotic bound equal, the lazy algorithm reduces the gradient update cost from cubic $O(n_{\max}^3)$ to quadratic $O(n_{\max}^2)$. In practice, where dense matrix multiplication has a large constant factor, this reduction leads to significant improvements in total runtime.

## D.4. Empirical Convergence and Wall-Clock Runtime

We complement the optimization analysis with two empirical studies. The first compares the lazy-gradient Frank–Wolfe (FW) implementation with the standard FW implementation on the CDOT objective, and the second reports wall-clock runtimes of CDOT against FGW and EFGW under a common benchmark setting. Both studies use three families of attributed compact mm spaces: quadrant-labeled point clouds, anisotropic clustered point clouds, and attributed stochastic block model (SBM) graphs. The reported numbers below correspond to the median of three runs at $N = 400$; consistent behavior is observed at $N = 200$.

---

**Algorithm 2** Convex Distance Operator Transport Plan via FW and Optional LAP with Lazy Gradient Updates

---

1: **Input:** Source data $\{(X_i, f_{\mathcal{X}}(X_i))\}_{i=1}^{n_X}$, target data $\{(Y_j, f_{\mathcal{Y}}(Y_j))\}_{j=1}^{n_Y}$, fusion weight parameter $0 \leq \alpha \leq 1$, iteration count $T$, initial coupling $\pi^{(0)}$
2: Construct matrices:
3: $[C_f]_{ij} \leftarrow \|f_{\mathcal{X}}(X_i) - f_{\mathcal{Y}}(Y_j)\|_2^2$
4: $[D_{\hat{\mathbb{P}}_X}]_{ii'} \leftarrow d_{\mathcal{X}}(X_i, X_{i'})/n_X$
5: $[D_{\hat{\mathbb{P}}_Y}]_{jj'} \leftarrow d_{\mathcal{Y}}(Y_j, Y_{j'})/n_Y$
6: Calculate the initial gradient $\nabla\hat{\mathcal{L}}_\alpha(\pi^{(0)})$
7: $G^{(0)} \leftarrow \nabla\hat{\mathcal{L}}_\alpha(\pi^{(0)})$
8: **for** $t = 0$ **to** $T - 1$ **do**
9: $\quad \mu^{(t)} \leftarrow \arg\min_{\mu \in \Pi(\hat{\mathbb{P}}_X, \hat{\mathbb{P}}_Y)} \operatorname{Tr}\left((G^{(t)})^\top \mu\right)$
10: $\quad \pi^{(t+1)} \leftarrow (1 - \gamma_t)\pi^{(t)} + \gamma_t\mu^{(t)}$ for some $0 < \gamma_t \leq 1$
11: $\quad$ **if** $t \leq T - 2$ **then**
12: $\quad\quad$ Calculate the directional component (lazy term) $\nabla\hat{\mathcal{L}}_\alpha(\mu^{(t)})$
13: $\quad\quad G^{(t+1)} \leftarrow (1 - \gamma_t)G^{(t)} + \gamma_t\nabla\hat{\mathcal{L}}_\alpha(\mu^{(t)})$
14: $\quad$ **end if**
15: **end for**
16: $\hat{\pi} \leftarrow \pi^{(T)}$
17: **Optional (LAP):** $\hat{P} \leftarrow \arg\max_{P \in \mathcal{P}} \operatorname{Tr}(P^\top\hat{\pi})$
18: **Return:** $\hat{\pi}$ (and optionally $\hat{P}$)

---

*Table 5.* Lazy versus standard FW on the CDOT objective. **Lazy (s)** and **Standard (s)** are total wall-clock runtimes; **Max obj. diff.** and **Max gap diff.** denote the maximum absolute differences in objective value and FW duality gap over the run.

| Space | $N$ | Lazy (s) | Standard (s) | Max obj. diff. | Max gap diff. |
|---|---|---|---|---|---|
| Quadrant point cloud | 400 | 2.692 | 2.699 | $1.84 \times 10^{-7}$ | $2.47 \times 10^{-6}$ |
| Anisotropic point cloud | 400 | 2.764 | 2.865 | $1.33 \times 10^{-7}$ | $8.36 \times 10^{-7}$ |
| SBM graph metric | 400 | 2.890 | 2.954 | $7.80 \times 10^{-8}$ | $1.51 \times 10^{-6}$ |

Table 5 reports the convergence comparison. In all three families, the lazy and standard FW variants produce nearly indistinguishable trajectories: the maximum discrepancy is below $1.9 \times 10^{-7}$ in objective value and below $2.5 \times 10^{-6}$ in duality gap. This shows that the lazy update does not alter the numerical optimization path of FW, so it preserves the convergence behavior of the standard method while saving per-iteration gradient cost.

*Table 6.* Wall-clock runtime of CDOT, FGW, and EFGW on the three benchmark families ($N = 400$), reporting median over three runs with outer-iteration counts in parentheses. All methods use a cap of 200 outer iterations. CDOT uses `tol`$= 10^{-7}$; FGW and EFGW use the POT defaults `tol`$= 10^{-9}$.

| Family | $N$ | CDOT | FGW | EFGW | CDOT/iter (s) | FGW/iter (s) | EFGW/iter (s) |
|---|---|---|---|---|---|---|---|
| Quadrant point cloud | 400 | 2.768 s (200) | 0.065 s (5) | 0.047 s (11) | $1.38 \times 10^{-2}$ | $1.30 \times 10^{-2}$ | $4.27 \times 10^{-3}$ |
| Anisotropic point cloud | 400 | 2.697 s (200) | 0.061 s (5) | 0.048 s (11) | $1.35 \times 10^{-2}$ | $1.21 \times 10^{-2}$ | $4.39 \times 10^{-3}$ |
| SBM graph metric | 400 | 3.063 s (200) | 0.220 s (15) | 0.040 s (11) | $1.53 \times 10^{-2}$ | $1.47 \times 10^{-2}$ | $3.65 \times 10^{-3}$ |

Table 6 summarizes the wall-clock comparison. Under the present benchmark setting, CDOT requires more total wall-clock time than FGW and EFGW, while the per-iteration cost of CDOT is on the order of $10^{-2}$ s. The runtime gap is therefore driven primarily by the stopping rule and the resulting iteration count rather than by a substantially larger per-iteration computational burden. A looser practical stopping rule could reduce the number of CDOT iterations in these benchmark families.

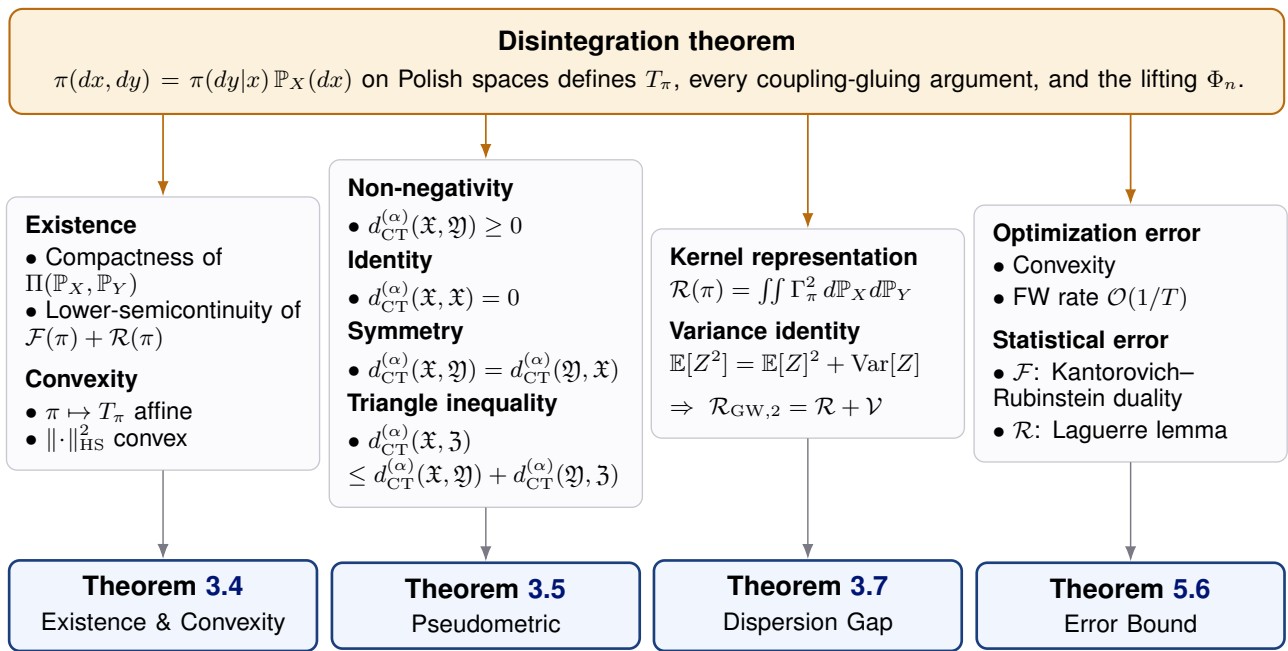

*Figure 5.* Proof roadmap for the four main theoretical results of CDOT. The orange banner names the foundational tool (the disintegration theorem), and the four panels list the technical ingredients feeding into each of the navy theorem boxes.

# E. Proof Roadmap

This appendix collects the technical machinery that supports our four main theoretical results. Before diving into the lemmas, Figure 5 provides a bird's-eye view of how the pieces fit together. The disintegration theorem (Lemma F.1) at the top underlies the conditional expectation operator $T_\pi$ and every coupling-gluing argument used throughout the appendix. The four panels below collect the technical ingredients feeding into each of Theorems 3.4, 3.5, 3.7, and 5.6, with the supporting lemmas and assumptions cited inline.

The four theorems group naturally into two layers. Theorems 3.4, 3.5, and 3.7 are population-level structural results: well-posedness and convexity of the CDOT problem, the pseudometric property of $d_{\mathrm{CT}}^{(\alpha)}$, and the precise gap to the Gromov–Wasserstein objective. Each of them rests directly on the disintegration theorem through the conditional expectation operator $T_\pi$, and uses either lower semicontinuity and convexity of $\|\cdot\|_{\mathrm{HS}}^2$ (Theorem 3.4), the contraction and composition properties of $T_\pi$ that underpin the triangle inequality (Theorem 3.5), or the Hilbert–Schmidt kernel representation of $\mathcal{R}$ together with the elementary variance identity (Theorem 3.7). Theorem 5.6 is the empirical-statistical result and splits cleanly into an optimization-error term, controlled by the Frank–Wolfe $\mathcal{O}(1/T)$ rate (which relies on the convexity guaranteed by Theorem 3.4), and a statistical-error term, whose feature component is handled by Kantorovich–Rubinstein duality (Lemma F.7) under the Lipschitz Assumption 5.4, and whose structural component is controlled by the Laguerre representation (Lemma F.5) under Assumptions 5.2–5.3 together with the stability Assumption 5.5.

# F. Useful Lemmas

This section presents useful lemmas to prove our theoretical results. We begin by reviewing the disintegration theorem, which provides the theoretical basis for decomposing joint couplings into conditional probability measures (Markov kernels). Subsequently, we summarize fundamental results in optimal transport theory, covering the existence of optimal couplings, Kantorovich duality, and specific characterizations for semi-discrete transport problems and the Wasserstein-1 distance. Finally, we introduce essential concepts regarding the weak topology of measures and the weak operator topology (WOT), which are requisite for establishing the topological properties of our proposed framework.

## F.1. Disintegration Theorem

**Lemma F.1** (Disintegration theorem). *Let $(\mathcal{X}, \mu)$ and $(\mathcal{Y}, \nu)$ be Polish probability spaces, where $\mu$ and $\nu$ are Borel probability measures. Then, for each joint coupling $\pi \in \Pi(\mu, \nu)$, there exists a $\mu$-a.e. uniquely determined family of probability measures $\{\pi_x\}_{x \in \mathcal{X}}$ on $\mathcal{Y}$ such that*

- *for every Borel-measurable set $B \subset \mathcal{Y}$, the function $x \mapsto \pi_x(B)$ is Borel-measurable;*

- *for every Borel-measurable function $f : \mathcal{X} \times \mathcal{Y} \to [0, \infty]$,*

$$\int_{\mathcal{X} \times \mathcal{Y}} f(x, y) \pi(dx, dy) = \int_{\mathcal{X}} \int_{\mathcal{Y}} f(x, y) \pi_x(dy) \mu(dx).$$

The disintegration theorem rigorously establishes the existence of a *regular conditional distribution* under the topological assumption that the underlying spaces are Polish. The family $\{\pi_x\}_{x \in \mathcal{X}}$ satisfies the definition of a *Markov kernel* from $\mathcal{X}$ to $\mathcal{Y}$. We will interpret $\pi_x$ as the conditional distribution of $y$ given $x$, and adopt the notation $\pi(dy \mid x) := \pi_x(dy)$.

## F.2. Optimal Transport

### F.2.1. EXISTENCE AND KANTOROVICH DUALITY

**Lemma F.2** (Existence of an optimal coupling). *Let $(\mathcal{X}, \mu)$ and $(\mathcal{Y}, \nu)$ be two Polish probability spaces with Borel probability measures $\mu$ and $\nu$. Let $a : \mathcal{X} \to \mathbb{R} \cup \{-\infty\}$ and $b : \mathcal{Y} \to \mathbb{R} \cup \{-\infty\}$ be two upper semicontinuous functions such that $a \in L^1(\mathcal{X}, \mu)$ and $b \in L^1(\mathcal{Y}, \nu)$. Let $c : \mathcal{X} \times \mathcal{Y} \to \mathbb{R} \cup \{+\infty\}$ be a lower semicontinuous cost function, such that $c(x, y) \geq a(x) + b(y)$ for all $x, y$. Then there exists an optimal coupling $\pi^* \in \Pi(\mu, \nu)$ such that*

$$\int_{\mathcal{X} \times \mathcal{Y}} c(x, y) \pi^*(dx, dy) = \inf_{\pi \in \Pi(\mu, \nu)} \left( \int_{\mathcal{X} \times \mathcal{Y}} c(x, y) \pi(dx, dy) \right) =: C(\mu, \nu).$$

*Proof.* The details can be found in Theorem 4.1 of Villani (2008). □

**Definition F.3** (*c*-convexity in Definition 5.2 of Villani (2008)). Let $\mathcal{X}, \mathcal{Y}$ be sets, and $c : \mathcal{X} \times \mathcal{Y} \to (-\infty, +\infty]$. A function $\psi : \mathcal{X} \to \mathbb{R} \cup \{+\infty\}$ is said to be *c-convex* if it is not identically $+\infty$ and there exists $\zeta : \mathcal{Y} \to \mathbb{R} \cup \{\pm\infty\}$ such that

$$\forall x \in \mathcal{X}, \quad \psi(x) = \sup_{y \in \mathcal{Y}} \left( \zeta(y) - c(x, y) \right).$$

Then, its *c-transform* is the function $\psi^c$ defined by

$$\forall y \in \mathcal{Y}, \quad \psi^c(y) = \inf_{x \in \mathcal{X}} \left( \psi(x) + c(x, y) \right).$$

**Lemma F.4** (Kantorovich duality). *Let $(\mathcal{X}, \mu)$ and $(\mathcal{Y}, \nu)$ be two Polish probability spaces with Borel probability measures $\mu$ and $\nu$. Let $c : \mathcal{X} \times \mathcal{Y} \to \mathbb{R}$ be a real-valued lower semicontinuous cost function, such that*

$$\forall (x, y) \in \mathcal{X} \times \mathcal{Y}, \quad c(x, y) \geq a(x) + b(y)$$

*for some real-valued upper semicontinuous functions $a \in L^1(\mathcal{X}, \mu)$ and $b \in L^1(\mathcal{Y}, \nu)$. Then, the following duality holds:*

$$C(\mu, \nu) = \sup_{\psi \in L^1(\mu)} \left\{ \int_{\mathcal{Y}} \psi^c(y) \, \nu(dy) - \int_{\mathcal{X}} \psi(x) \, \mu(dx) \right\}.$$

*If it is further assumed that the optimal cost $C(\mu, \nu)$ is finite and the cost $c$ has a pointwise upper bound, i.e.,*

$$c(x, y) \leq c_{\mathcal{X}}(x) + c_{\mathcal{Y}}(y), \quad (c_{\mathcal{X}}, c_{\mathcal{Y}}) \in L^1(\mu) \times L^1(\nu),$$

*the dual problem has solutions. Moreover, $\pi \in \Pi(\mu, \nu)$ is optimal if and only if there is a c-convex $\psi$ such that*

$$\psi^c(y) - \psi(x) = c(x, y), \quad \text{for } \pi\text{-almost every } (x, y).$$

*Proof.* The details can be found in Theorem 5.10 of Villani (2008). □

The Kantorovich duality transforms the primal minimization problem over the space of probability measures into a dual maximization problem. The optimality condition $\psi^c(y) - \psi(x) = c(x, y)$ enables the characterization of the closed form of optimal transport plans in the semi-discrete problem.

## F.3. Semi-Discrete Optimal Transport

**Lemma F.5** (Laguerre lemma). *Let $(\mathcal{X}, d_{\mathcal{X}})$ be a compact metric space, $\mathbb{P}_X$ be a Borel probability measure on $\mathcal{X}$ that satisfies Assumptions 5.2 and 5.3, and $\hat{\mathbb{P}}_X$ be the empirical probability measure of $\mathbb{P}_X$ supported on $n$ fixed distinct data points, i.e.,*

$$\hat{\mathbb{P}}_X = \frac{1}{n} \sum_{i=1}^{n} \delta_{X_i}.$$

*Let $\hat{\mathcal{X}} := \{X_1, ..., X_n\} \subset \mathcal{X}$. Then $(\hat{\mathcal{X}}, \hat{\mathbb{P}}_X)$ is clearly a Polish probability space. Let $(\pi^*, \psi^*)$ be primal and dual optimal solutions for the semi-discrete problem:*

$$\min_{\pi \in \Pi(\mathbb{P}_X, \hat{\mathbb{P}}_X)} \left( \int_{\mathcal{X} \times \hat{\mathcal{X}}} d_{\mathcal{X}}(x, \hat{x}) \pi(dx, d\hat{x}) \right) = \max_{\psi \in L^1(\hat{\mathcal{X}}, \hat{\mathbb{P}}_X)} \left( \int_{\mathcal{X}} \psi^{d_{\mathcal{X}}}(x) \mathbb{P}_X(dx) - \frac{1}{n} \sum_{i=1}^{n} \psi(X_i) \right),$$

*where the $d_{\mathcal{X}}$-transform is taken as $\psi^{d_{\mathcal{X}}}(x) = \min_{1 \le j \le n} (d_{\mathcal{X}}(x, X_j) + \psi(X_j))$. Note that all elements in $L^1(\hat{\mathcal{X}}, \hat{\mathbb{P}}_X)$ can be identified with $n$-dimensional real-valued vectors.*

*For each $i = 1, ..., n$, define the Laguerre-like cell $V_i$ by*

$$V_i := \{x \in \mathcal{X} : d_{\mathcal{X}}(x, X_i) + \psi^*(X_i) \le d_{\mathcal{X}}(x, X_j) + \psi^*(X_j), \forall j\} \cup \{X_i\}.$$

*Then $\{V_i\}_{i=1}^{n}$ forms a $\mathbb{P}_X$-a.e. partition of $\mathcal{X}$, satisfies $\mathbb{P}_X(V_i) = 1/n$ for all $i$, and*

$$\pi^* = \sum_{i=1}^{n} \mathbb{P}_X \lfloor_{V_i} \otimes \delta_{X_i}, \quad \mathbb{P}_X \lfloor_{V_i}(dx) = \mathbb{I}_{V_i}(x) \mathbb{P}_X(dx),$$

*where $\mathbb{I}_{V_i}$ is the indicator function of $V_i$.*

*Proof.* Denote by $\psi_j^* := \psi^*(X_j)$. Then, we have the following:

$$(\psi^*)^{d_{\mathcal{X}}}(x) = \min_{1 \le j \le n} \left( d_{\mathcal{X}}(x, X_j) + \psi_j^* \right).$$

Now define, for each $i = 1, ..., n$,

$$\tilde{V}_i := \left\{ x \in \mathcal{X} : d_{\mathcal{X}}(x, X_i) + \psi_i^* \le d_{\mathcal{X}}(x, X_j) + \psi_j^*, \forall j \right\}.$$

By the no-ties assumption (Assumption 5.3), the sets $\{\tilde{V}_i\}_{i=1}^{n}$ form a $\mathbb{P}_X$-a.e. partition of $\mathcal{X}$, and moreover for $\mathbb{P}_X$-a.e. $x$ there exists a unique index $i(x)$ such that $x \in \tilde{V}_{i(x)}$.

Let $\pi^* \in \Pi(\mathbb{P}_X, \hat{\mathbb{P}}_X)$ be a primal optimal plan. Since the second marginal of $\pi^*$ is supported on the finite set $\{X_1, ..., X_n\}$, there exist Borel-measurable functions $p_i : \mathcal{X} \to [0, 1]$ such that

$$\sum_{i=1}^{n} p_i(x) = 1 \quad \text{for } \mathbb{P}_X\text{-a.e. } x,$$

and the disintegration of $\pi^*$ with respect to the first marginal $\mathbb{P}_X$ takes the form

$$\pi^*(dx, d\hat{x}) = \sum_{i=1}^{n} \underbrace{p_i(x) \, \delta_{X_i}(d\hat{x})}_{= \pi^*(d\hat{x}|x)} \mathbb{P}_X(dx).$$

We next use complementary slackness. By the Kantorovich duality lemma, the following holds:

$$(\psi^*)^{d_{\mathcal{X}}}(x) - \psi^*(\hat{x}) = d_{\mathcal{X}}(x, \hat{x}), \quad \text{for } \pi^*\text{-a.e. } (x, \hat{x}).$$

In particular, for each $i$ and for $\mathbb{P}_X$-a.e. $x$ such that $p_i(x) > 0$, we must have

$$(\psi^*)^{d_{\mathcal{X}}}(x) - \psi^*(X_i) = d_{\mathcal{X}}(x, X_i).$$

Equivalently,

$$(\psi^*)^{d_{\mathcal{X}}}(x) = d_{\mathcal{X}}(x, X_i) + \psi_i^*.$$

But we already have

$$(\psi^*)^{d_{\mathcal{X}}}(x) = \min_{1 \leq j \leq n} \big(d_{\mathcal{X}}(x, X_j) + \psi_j^*\big),$$

so the equality implies that $i$ attains the minimum, i.e.,

$$p_i(x) > 0 \quad \implies \quad x \in \tilde{V}_i \quad \text{for } \mathbb{P}_X\text{-a.e. } x.$$

By the no-ties assumption (Assumption 5.3), the minimizer is unique for $\mathbb{P}_X$-a.e. $x$, hence for $\mathbb{P}_X$-a.e. $x$ there exists a unique index $i(x)$ such that

$$p_{i(x)}(x) = 1, \quad p_j(x) = 0, \quad (j \neq i(x)),$$

and necessarily $x \in \tilde{V}_{i(x)}$. Therefore,

$$\pi^*(dx, d\hat{x}) = \sum_{i=1}^{n} \mathbb{I}_{\tilde{V}_i}(x) \, \mathbb{P}_X(dx) \, \delta_{X_i}(d\hat{x}) = \sum_{i=1}^{n} \mathbb{P}_X \lfloor_{\tilde{V}_i}(dx) \, \delta_{X_i}(d\hat{x}).$$

Now we prove the mass constraint $\mathbb{P}_X(\tilde{V}_i) = 1/n$. Since the second marginal of $\pi^*$ is $\hat{\mathbb{P}}_X$, we have for each $i$,

$$\frac{1}{n} = \hat{\mathbb{P}}_X(\{X_i\}) = \pi^*(\mathcal{X} \times \{X_i\}) = \pi^*(\tilde{V}_i \times \{X_i\}) = \mathbb{P}_X(\tilde{V}_i).$$

Finally, define

$$V_i := \tilde{V}_i \cup \{X_i\}.$$

Since $\mathbb{P}_X$ is non-atomic, $\mathbb{P}_X(\{X_i\}) = 0$, hence $\mathbb{P}_X(V_i) = \mathbb{P}_X(\tilde{V}_i) = 1/n$. Moreover, the representation of $\pi^*$ remains valid when $\tilde{V}_i$ is replaced by $V_i$, because $\mathbb{P}_X \lfloor_{V_i}$ and $\mathbb{P}_X \lfloor_{\tilde{V}_i}$ are equivalent as they differ only on a $\mathbb{P}_X$-null set. This completes the proof. $\qquad \square$

Lemma F.5 characterizes the optimal plan via a partition of $\mathcal{X}$ into $n$ Laguerre cells $V_i$. Unlike standard Voronoi tessellations, $V_i$ incorporates the optimal dual potential $\psi^*(X_i)$ as an additive weight. Optimality ensures these weights adjust the cell boundaries to be perfectly balanced, capturing exactly $1/n$ of the mass $\mathbb{P}_X$ per cell.

Figure 6 illustrates the geometric intuition of this result. Here, the generator points and their associated polygonal regions correspond to the samples $X_i$ and the Laguerre cells $V_i$, respectively. Note that the figure depicts standard Voronoi cells for brevity, which corresponds to the simplified assumption of constant dual potentials (i.e., $\psi^*(X_i) = c$). As $n$ increases from 100 to 1000, the spatial extent of each cell shrinks significantly (mass $10^{-2} \to 10^{-3}$). Since the optimal coupling $Q_X$ maps the entire region $V_i$ to a single point $X_i$, this shrinkage forces the conditional measure $Q_X(\cdot|\hat{x})$ to concentrate tightly around the identity mapping $x = \hat{x}$. Consequently, the probability mass accumulates along the diagonal, asymptotically eliminating the blurring effect.

Formalizing this via Markov's inequality on the $W_1^{d_{\mathcal{X}}}$-optimal coupling $Q_X \in \Pi(\mathbb{P}_X, \hat{\mathbb{P}}_X)$, we have

$$\mathbb{P}_{(X, \hat{X}) \sim Q_X}\Big(d_{\mathcal{X}}(X, \hat{X}) > \varepsilon\Big) \leq \frac{W_1^{d_{\mathcal{X}}}(\mathbb{P}_X, \hat{\mathbb{P}}_X)}{\varepsilon}.$$

Therefore, provided that the empirical measures converge in the Wasserstein-1 metric, the blurring effect introduced by lifting $\hat{\pi}$ to the continuous space $\Pi(\mathbb{P}_X, \mathbb{P}_Y)$ vanishes asymptotically as the sample size $n$ increases.

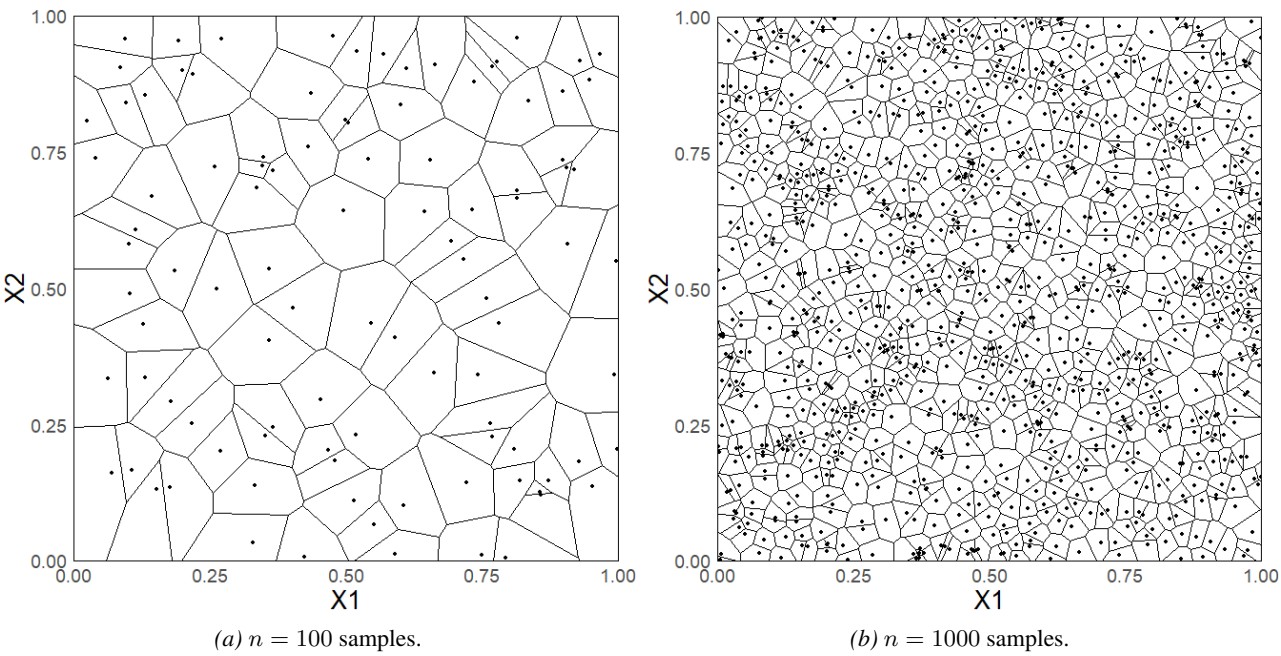

*(a)* $n = 100$ samples.  *(b)* $n = 1000$ samples.

*Figure 6.* Voronoi tessellations generated from samples drawn from the uniform distribution $U([0,1]^2)$.

### F.3.1. DUAL FORM OF THE WASSERSTEIN-1 DISTANCE

**Definition F.6** (Wasserstein-$p$ distance). Let $(\mathcal{S}, d_\mathcal{S})$ be a compact metric space. For Borel probability measures $\mu$ and $\nu$ on $\mathcal{S}$, the Wasserstein-$p$ distance $(1 \leq p < \infty)$ is

$$W_p^{d_\mathcal{S}}(\mu, \nu) := \left( \min_{\pi \in \Pi(\mu, \nu)} \int_{\mathcal{S} \times \mathcal{S}} d_\mathcal{S}(x, y)^p \, \pi(dx, dy) \right)^{1/p}.$$

We recall the definition of the Wasserstein-$p$ distance from the main text. Since $d_\mathcal{S}^p$ is continuous and $\mathcal{S}$ is compact, the infimum is always attained by Lemma F.2.

**Lemma F.7** (Kantorovich–Rubinstein formula). *Let $(\mathcal{X}, d_\mathcal{X})$ be a compact metric space. Then, for any two Borel probability measures $\mu, \nu$ on $\mathcal{X}$,*

$$W_1^{d_\mathcal{X}}(\mu, \nu) = \sup_{\|\psi\|_{\mathrm{Lip}} \leq 1} \left( \int_\mathcal{X} \psi d\mu - \int_\mathcal{X} \psi d\nu \right),$$

*where $\| \cdot \|_{\mathrm{Lip}}$ is defined by*

$$\|\psi\|_{\mathrm{Lip}} = \sup_{x \neq y} \frac{|\psi(x) - \psi(y)|}{d_\mathcal{X}(x, y)}.$$

Lemma F.7 provides a crucial simplification of Lemma F.4 when the cost function is a metric $d_\mathcal{X}$. In this case, the complex constraint involving the $c$-transform is relaxed into a global 1-Lipschitz constraint. This duality plays a pivotal role in the proof of Theorem 5.6. By establishing that the feature cost function $c_f$ is Lipschitz continuous (Assumption 5.4), we invoke this lemma to directly upper-bound the difference in expected costs by the Wasserstein-1 distance between the empirical and population couplings.

### F.4. Weak Topology

In this subsection, we introduce the fundamental concepts of weak topology and the weak operator topology (WOT). These topological structures are essential for establishing the lower semicontinuity of the proposed CDOT objective and the pseudometric property of the CDOT discrepancy.

### F.4.1. WEAK TOPOLOGY

**Definition F.8** (Weak topology). Let $(\mathcal{S}, \mathcal{B}(\mathcal{S}))$ be a Polish space and $\mathcal{P}(\mathcal{S})$ be the set of all Borel probability measures on $(\mathcal{S}, \mathcal{B}(\mathcal{S}))$. For any $f \in C_b(\mathcal{S})$, consider the functional

$$\phi_f : \mathcal{P}(\mathcal{S}) \to \mathbb{R}, \quad \phi_f(\mu) = \int_{\mathcal{S}} f(s)\mu(ds).$$

Let $\mathcal{F} = \{\phi_f : f \in C_b(\mathcal{S})\}$. The weak topology on $\mathcal{P}(\mathcal{S})$ is defined as the coarsest topology making every functional in $\mathcal{F}$ continuous. Consequently, a net $(\mu_\alpha)$ converges weakly to $\mu$ if and only if

$$\int_{\mathcal{S}} f(s)\mu_\alpha(ds) \to \int_{\mathcal{S}} f(s)\mu(ds), \quad \forall f \in C_b(\mathcal{S}).$$

We write $\mu_\alpha \xrightarrow{w} \mu$ if $(\mu_\alpha)$ converges weakly to $\mu$.

Since the underlying space $\mathcal{S}$ is Polish, the space $\mathcal{P}(\mathcal{S})$ equipped with the weak topology is also a Polish space (i.e., separable and completely metrizable). This topological structure allows us to utilize powerful convergence results, most notably Prokhorov's theorem, which states that a subset of probability measures is relatively compact if and only if it is tight.

This property is particularly important when analyzing couplings. Let $\mathcal{X}$ and $\mathcal{Y}$ be Polish spaces and consider $\Pi(\mu, \nu)$ for two fixed Borel probability measures $\mu \in \mathcal{P}(\mathcal{X})$ and $\nu \in \mathcal{P}(\mathcal{Y})$. Since $\mathcal{X}$ and $\mathcal{Y}$ are Polish, the measures $\mu$ and $\nu$ are tight by Ulam's theorem. Consequently, the collection $\Pi(\mu, \nu)$ is tight in $\mathcal{P}(\mathcal{X} \times \mathcal{Y})$ because a set of measures on a product space is tight if and only if its marginals are tight. Furthermore, since the constraint of having fixed marginals is closed under weak convergence, $\Pi(\mu, \nu)$ is a closed subset of a compact set. Therefore, $\Pi(\mu, \nu)$ is compact in the weak topology.

### F.4.2. WEAK OPERATOR TOPOLOGY

Now we introduce the weak operator topology. Unless stated otherwise, we assume all Hilbert spaces to be real and separable.

**Definition F.9** (Weak operator topology). Let $\mathcal{H}_X$ and $\mathcal{H}_Y$ be Hilbert spaces, and let $B(\mathcal{H}_X, \mathcal{H}_Y)$ denote the set of bounded linear operators from $\mathcal{H}_X$ to $\mathcal{H}_Y$. For $x \in \mathcal{H}_X$ and $y \in \mathcal{H}_Y$, define the linear functional

$$\phi_{x,y} : B(\mathcal{H}_X, \mathcal{H}_Y) \to \mathbb{R}, \quad \phi_{x,y}(T) = \langle Tx, y \rangle_{\mathcal{H}_Y}.$$

Let $\mathcal{F} = \{\phi_{x,y} : x \in \mathcal{H}_X, y \in \mathcal{H}_Y\}$. The *weak operator topology (WOT)* on $B(\mathcal{H}_X, \mathcal{H}_Y)$ is the coarsest topology making every functional in $\mathcal{F}$ continuous. Equivalently, a net $(T_\alpha)$ converges to $T$ in WOT if and only if

$$\langle T_\alpha x, y \rangle_{\mathcal{H}_Y} \to \langle Tx, y \rangle_{\mathcal{H}_Y}, \quad \forall x \in \mathcal{H}_X, y \in \mathcal{H}_Y.$$

The WOT plays a role analogous to the weak topology of measures but in the context of linear operators. In infinite-dimensional Hilbert spaces, the closed unit ball is not compact in the operator norm topology (a result known as Riesz's lemma). However, the unit ball is compact in the WOT (a consequence of the Banach–Alaoglu theorem). This compactness is pivotal for our analysis: it ensures that any uniformly bounded sequence of operators—such as the conditional expectations $\{T_{\pi_n}\}$ induced by a sequence of couplings—possesses a convergent subsequence in WOT. This property allows us to define limits for operators even when they do not converge in the stronger norm topology.

### F.5. Lower Semicontinuity

This subsection establishes the lower semicontinuity properties required to guarantee the existence of an optimal solution to the CDOT problem. We recall the definition of lower semicontinuity and the generalized Weierstrass theorem, and show that the squared Hilbert–Schmidt (HS) norm is lower semicontinuous with respect to the weak operator topology.

**Definition F.10** (Lower semicontinuity). Let $(\mathcal{X}, \tau)$ be a topological space. A function $f : \mathcal{X} \to \mathbb{R} \cup \{\pm\infty\}$ is said to be *lower semicontinuous at $x_0 \in \mathcal{X}$* if for every net $(x_\alpha)_{\alpha \in A}$ in $\mathcal{X}$ such that $x_\alpha \to x_0$ w.r.t. $\tau$,

$$f(x_0) \leq \liminf_\alpha f(x_\alpha), \quad \text{where} \quad \liminf_\alpha f(x_\alpha) := \sup_{\alpha_0 \in A} \inf_{\alpha \succeq \alpha_0} f(x_\alpha).$$

We say that $f$ is *lower semicontinuous* if it is lower semicontinuous at every $x_0 \in \mathcal{X}$.

**Lemma F.11** (Generalized Weierstrass theorem). *Let $\mathcal{X}$ be a topological space and $\mathcal{K} \subset \mathcal{X}$ be a nonempty compact set. If $f : \mathcal{K} \to \mathbb{R}$ is a lower semicontinuous function, then $f$ is bounded below and there exists $x^* \in \mathcal{K}$ such that $f(x^*) = \inf_{x \in \mathcal{K}} f(x)$.*

*Proof.* The proof can be found in Theorem 2.43 of Aliprantis & Border (2006). □

This theorem generalizes the classical Weierstrass extreme value theorem (which requires continuity) to lower semicontinuous functions. It serves as the primary tool for establishing the existence of optimal solutions in variational problems. Since we have already established that the set of couplings $\Pi(\mu, \nu)$ is compact in the weak topology, proving the lower semicontinuity of our loss function is sufficient to guarantee the existence of an optimal transport plan.

**Lemma F.12.** *Let $\mathcal{H}_X$ and $\mathcal{H}_Y$ be Hilbert spaces. Denote by $\mathrm{HS}(\mathcal{H}_X, \mathcal{H}_Y) \subset B(\mathcal{H}_X, \mathcal{H}_Y)$ the set of Hilbert–Schmidt operators from $\mathcal{H}_X$ to $\mathcal{H}_Y$ equipped with the weak operator topology. Then, $\| \cdot \|_{\mathrm{HS}}^2$ is lower semicontinuous in WOT: for any sequence $\{T_n\}_{n=1}^{\infty} \subset \mathrm{HS}(\mathcal{H}_X, \mathcal{H}_Y)$ that converges to $T \in \mathrm{HS}(\mathcal{H}_X, \mathcal{H}_Y)$ in WOT, it holds that*

$$\|T\|_{\mathrm{HS}}^2 \leq \liminf_{n \to \infty} \|T_n\|_{\mathrm{HS}}^2.$$

*Proof.* Let $\{e_i\}_{i=1}^{\infty}$ and $\{\varphi_j\}_{j=1}^{\infty}$ be orthonormal bases for $\mathcal{H}_X$ and $\mathcal{H}_Y$, respectively. By the definition of the Hilbert–Schmidt norm, we express the norm of $T$ as

$$\|T\|_{\mathrm{HS}}^2 = \sum_{i=1}^{\infty} \|Te_i\|_{\mathcal{H}_Y}^2 = \sum_{i=1}^{\infty} \sum_{j=1}^{\infty} |\langle Te_i, \varphi_j \rangle_{\mathcal{H}_Y}|^2.$$

By the assumption that $T_n \to T$ in WOT, for any fixed basis vectors $e_i$ and $\varphi_j$, we have

$$\lim_{n \to \infty} \langle T_n e_i, \varphi_j \rangle_{\mathcal{H}_Y} = \langle Te_i, \varphi_j \rangle_{\mathcal{H}_Y} \implies \lim_{n \to \infty} |\langle T_n e_i, \varphi_j \rangle_{\mathcal{H}_Y}| = |\langle Te_i, \varphi_j \rangle_{\mathcal{H}_Y}|.$$

Therefore, by Fatou's lemma,

$$\|T\|_{\mathrm{HS}}^2 = \sum_{i,j} |\langle Te_i, \varphi_j \rangle_{\mathcal{H}_Y}|^2 \leq \liminf_{n \to \infty} \sum_{i,j} |\langle T_n e_i, \varphi_j \rangle_{\mathcal{H}_Y}|^2 = \liminf_{n \to \infty} \|T_n\|_{\mathrm{HS}}^2.$$

This concludes the proof. □

Lemma F.12 is useful to prove the lower semicontinuity of our structural regularization term, which in turn guarantees the existence of an optimal solution for our CDOT problem.

# G. Proofs

This section provides the proofs for our main theoretical results.

## G.1. Derivation of Matrix Representations

We show that under the empirical measures $\hat{\mathbb{P}}_X = \sum_{i=1}^{n} \delta_{X_i}/n$ and $\hat{\mathbb{P}}_Y = \sum_{j=1}^{n} \delta_{Y_j}/n$, the distance and conditional expectation operators (Definitions 3.1 and 3.2) admit the following matrix representations:

$$[D_{\hat{\mathbb{P}}_X}]_{ij} = \frac{d_{\mathcal{X}}(X_i, X_j)}{n}, \quad [D_{\hat{\mathbb{P}}_Y}]_{ij} = \frac{d_{\mathcal{Y}}(Y_i, Y_j)}{n}, \quad [T_\pi]_{ij} = n\pi_{ij}.$$

We rigorously derive the representation for the distance operator; the other results follow analogously.

Let $\hat{\mathcal{X}} = \{X_1, ..., X_n\}$ be the discrete support equipped with a metric $d_{\mathcal{X}}$ and the uniform probability measure $\hat{\mathbb{P}}_X$. The function space $L^2(\hat{\mathcal{X}}, \hat{\mathbb{P}}_X)$ is an $n$-dimensional Hilbert space isomorphic to the Euclidean space $\mathbb{R}^n$. Specifically, we consider the canonical bijection $\varphi : L^2(\hat{\mathcal{X}}, \hat{\mathbb{P}}_X) \to \mathbb{R}^n$ defined by evaluation at the support points:

$$f \longmapsto \mathbf{f} := \begin{pmatrix} f(X_1) \\ \vdots \\ f(X_n) \end{pmatrix} \in \mathbb{R}^n.$$

Since $D_{\hat{\mathbb{P}}_X}$ is a linear operator on a finite-dimensional space, it admits a unique matrix representation $\mathbf{D}_X \in \mathbb{R}^{n \times n}$ satisfying $\varphi(D_{\hat{\mathbb{P}}_X} f) = \mathbf{D}_X \mathbf{f}$ for all $f$. For any $i \in \{1, ..., n\}$, the $i$-th component of the transformed vector is given by the definition of the operator:

$$(D_{\hat{\mathbb{P}}_X} f)(X_i) = \int_{\hat{x} \in \hat{\mathcal{X}}} d_{\mathcal{X}}(X_i, \hat{x}) f(\hat{x}) \hat{\mathbb{P}}_X(d\hat{x})$$
$$= \sum_{j=1}^{n} d_{\mathcal{X}}(X_i, X_j) f(X_j) \hat{\mathbb{P}}_X(\{X_j\})$$
$$= \sum_{j=1}^{n} \frac{d_{\mathcal{X}}(X_i, X_j)}{n} f(X_j).$$

On the other hand, from the definition of matrix-vector multiplication, the $i$-th component is:

$$(\mathbf{D}_X \mathbf{f})_i = \sum_{j=1}^{n} [\mathbf{D}_X]_{ij} f(X_j).$$

Since this equality holds for any arbitrary function $f$ (and consequently any vector $\mathbf{f}$), comparing the coefficients of $f(X_j)$ yields the desired entry-wise equality:

$$[\mathbf{D}_X]_{ij} = \frac{d_{\mathcal{X}}(X_i, X_j)}{n}.$$

### G.2. Hilbert–Schmidt Kernel Representation of $\mathcal{R}(\pi)$

In this subsection, we rigorously derive the integral representation of $\mathcal{R}(\pi)$ in (1). For $g \in L^2(\mathcal{Y}, \mathbb{P}_Y)$, observe that

$$(D_{\mathbb{P}_X} T_\pi g)(x) = \mathbb{E}\left[ d_{\mathcal{X}}(x, X) \mathbb{E}\left[ g(Y) \mid X \right] \right] = \int_{\mathcal{Y}} \Gamma_\pi^{(1)}(x, y) g(y) \mathbb{P}_Y(dy),$$
$$(T_\pi D_{\mathbb{P}_Y} g)(x) = \mathbb{E}\left[ (D_{\mathbb{P}_Y} g)(Y) \mid X = x \right] = \int_{\mathcal{Y}} \Gamma_\pi^{(2)}(x, y) g(y) \mathbb{P}_Y(dy).$$

Since $d_{\mathcal{X}}$ and $d_{\mathcal{Y}}$ are bounded by 1, $D_{\mathbb{P}_X} T_\pi - T_\pi D_{\mathbb{P}_Y}$ is a well-defined Hilbert–Schmidt operator with a kernel $\Gamma_\pi(x, y)$:

$$\|D_{\mathbb{P}_X} T_\pi - T_\pi D_{\mathbb{P}_Y}\|_{\text{HS}}^2 = \int_{\mathcal{Y}} \int_{\mathcal{X}} \Gamma_\pi(x, y)^2 \mathbb{P}_X(dx) \mathbb{P}_Y(dy) \leq 1.$$

### G.3. Proof for Theorem 3.4

*Proof.* The existence follows from the generalized Weierstrass theorem (Lemma F.11), which requires two ingredients: (i) the transport polytope $\Pi(\mathbb{P}_X, \mathbb{P}_Y)$ is nonempty and compact in the weak topology, as discussed in Appendix F.4; and (ii) the CDOT objective $\mathcal{L}_\alpha(\cdot; \mathfrak{X}, \mathfrak{Y})$ is lower semicontinuous on $\Pi(\mathbb{P}_X, \mathbb{P}_Y)$, as established in the following lemma.

**Lemma G.1.** *Let $\mathcal{L}_\alpha(\pi; \mathfrak{X}, \mathfrak{Y})$ be the CDOT objective as defined in Definition 3.3:*

$$\mathcal{L}_\alpha(\pi; \mathfrak{X}, \mathfrak{Y}) = (1 - \alpha) \underbrace{\mathbb{E}_\pi\left[ \|f_{\mathcal{X}}(X) - f_{\mathcal{Y}}(Y)\|_2^2 \right]}_{=:\mathcal{F}(\pi; \mathfrak{X}, \mathfrak{Y})} + \frac{\alpha}{2} \underbrace{\|D_{\mathbb{P}_X} T_\pi - T_\pi D_{\mathbb{P}_Y}\|_{\text{HS}}^2}_{=:\mathcal{R}(\pi; \mathfrak{X}, \mathfrak{Y})}.$$

*Then, $\mathcal{L}_\alpha(\pi; \mathfrak{X}, \mathfrak{Y})$ is lower semicontinuous.*

*Proof.* See Appendix G.7. $\square$

Combining (i) and (ii), the generalized Weierstrass theorem yields the existence of an optimal coupling $\pi^* \in \Pi(\mathbb{P}_X, \mathbb{P}_Y)$.

To establish convexity, it suffices to show that

$$\pi \mapsto \|D_{\mathbb{P}_X} T_\pi - T_\pi D_{\mathbb{P}_Y}\|_{\text{HS}}^2$$

is convex on $\Pi(\mathbb{P}_X, \mathbb{P}_Y)$, since the convexity of the first term is obvious.

Let $\pi_1, \pi_2 \in \Pi(\mathbb{P}_X, \mathbb{P}_Y)$ and $t \in [0, 1]$, and define $\pi_t = t\pi_1 + (1-t)\pi_2$. Since both $\pi_1$ and $\pi_2$ share the same marginals $\mathbb{P}_X$ and $\mathbb{P}_Y$, the disintegration theorem ensures that we may choose versions of the disintegrations such that

$$\pi_t(\cdot \mid x) = t\,\pi_1(\cdot \mid x) + (1-t)\,\pi_2(\cdot \mid x) \quad \text{for } \mathbb{P}_X\text{-a.e. } x \in \mathcal{X}.$$

That is, the conditional distribution of $Y$ given $X = x$ under $\pi_t$ is the convex combination of the conditional distributions under $\pi_1$ and $\pi_2$. Consequently, for any $g \in L^2(\mathcal{Y}, \mathbb{P}_Y)$,

$$(T_{\pi_t}g)(x) = \int g(y)\,\pi_t(dy \mid x) = t \int g(y)\,\pi_1(dy \mid x) + (1-t) \int g(y)\,\pi_2(dy \mid x)$$
$$= t\,(T_{\pi_1}g)(x) + (1-t)\,(T_{\pi_2}g)(x),$$

which shows that $T_{\pi_t}$ depends affinely on $\pi$, i.e.,

$$T_{\pi_t} = t\,T_{\pi_1} + (1-t)\,T_{\pi_2}.$$

Because $D_{\mathbb{P}_X}$ and $D_{\mathbb{P}_Y}$ are linear operators, it follows that

$$D_{\mathbb{P}_X}T_{\pi_t} - T_{\pi_t}D_{\mathbb{P}_Y} = t\,(D_{\mathbb{P}_X}T_{\pi_1} - T_{\pi_1}D_{\mathbb{P}_Y}) + (1-t)\,(D_{\mathbb{P}_X}T_{\pi_2} - T_{\pi_2}D_{\mathbb{P}_Y}).$$

Denoting $A_i := D_{\mathbb{P}_X}T_{\pi_i} - T_{\pi_i}D_{\mathbb{P}_Y}$ $(i = 1, 2)$ and $A_t := D_{\mathbb{P}_X}T_{\pi_t} - T_{\pi_t}D_{\mathbb{P}_Y}$, we have $A_t = tA_1 + (1-t)A_2$. Since $\|\cdot\|_{\mathrm{HS}}^2$ is clearly convex, we obtain

$$\|A_t\|_{\mathrm{HS}}^2 = \|tA_1 + (1-t)A_2\|_{\mathrm{HS}}^2 \le t\,\|A_1\|_{\mathrm{HS}}^2 + (1-t)\,\|A_2\|_{\mathrm{HS}}^2.$$

Therefore, $\pi \mapsto \|D_{\mathbb{P}_X}T_\pi - T_\pi D_{\mathbb{P}_Y}\|_{\mathrm{HS}}^2$ is convex on $\Pi(\mathbb{P}_X, \mathbb{P}_Y)$. $\qquad\square$

### G.4. Proof for Theorem 3.5

*Proof.* We need to prove the following properties: for any attributed compact mm spaces $\mathfrak{X}, \mathfrak{Y}$, and $\mathfrak{Z}$,

**(Non-negativity).** $d_{\mathrm{CT}}^{(\alpha)}(\mathfrak{X}, \mathfrak{Y}) \ge 0$.

**(Identity).** $d_{\mathrm{CT}}^{(\alpha)}(\mathfrak{X}, \mathfrak{X}) = 0$.

**(Symmetry).** $d_{\mathrm{CT}}^{(\alpha)}(\mathfrak{X}, \mathfrak{Y}) = d_{\mathrm{CT}}^{(\alpha)}(\mathfrak{Y}, \mathfrak{X})$.

**(Triangle inequality).** $d_{\mathrm{CT}}^{(\alpha)}(\mathfrak{X}, \mathfrak{Z}) \le d_{\mathrm{CT}}^{(\alpha)}(\mathfrak{X}, \mathfrak{Y}) + d_{\mathrm{CT}}^{(\alpha)}(\mathfrak{Y}, \mathfrak{Z})$.

**(Non-negativity) & (Identity).** The non-negativity and the identity are immediate from the definition.

**(Symmetry).** Let $\pi^* \in \Pi(\mathbb{P}_X, \mathbb{P}_Y)$ be an optimal solution that minimizes $\mathcal{L}_\alpha(\pi; \mathfrak{X}, \mathfrak{Y})$, i.e.,

$$d_{\mathrm{CT}}^{(\alpha)}(\mathfrak{X}, \mathfrak{Y}) = \mathcal{L}_\alpha(\pi^*; \mathfrak{X}, \mathfrak{Y})^{1/2}.$$

Define a bijection $\varphi : \Pi(\mathbb{P}_X, \mathbb{P}_Y) \to \Pi(\mathbb{P}_Y, \mathbb{P}_X)$ by

$$\varphi(\pi)(A \times B) = \pi(B \times A), \quad \forall A \in \mathcal{B}(\mathcal{Y}), \ \forall B \in \mathcal{B}(\mathcal{X}).$$

It suffices to show that for any $\pi \in \Pi(\mathbb{P}_X, \mathbb{P}_Y)$, $\mathcal{L}_\alpha(\pi; \mathfrak{X}, \mathfrak{Y}) = \mathcal{L}_\alpha(\varphi(\pi); \mathfrak{Y}, \mathfrak{X})$. To show this, we prove

$$\mathcal{F}(\pi; \mathfrak{X}, \mathfrak{Y}) = \mathcal{F}(\varphi(\pi); \mathfrak{Y}, \mathfrak{X}), \quad \mathcal{R}(\pi; \mathfrak{X}, \mathfrak{Y}) = \mathcal{R}(\varphi(\pi); \mathfrak{Y}, \mathfrak{X}).$$

The first claim is clear by the symmetry of the Euclidean norm. For the second claim, we note that the adjoint operator $T_\pi^* : L^2(\mathcal{X}, \mathbb{P}_X) \to L^2(\mathcal{Y}, \mathbb{P}_Y)$ is defined by $(T_\pi^* f)(y) = \mathbb{E}_\pi[f(X) \mid Y = y]$ for each $f \in L^2(\mathcal{X}, \mathbb{P}_X)$. In particular, $T_\pi^* = T_{\varphi(\pi)}$ holds. Thus,

$$\mathcal{R}(\pi; \mathfrak{X}, \mathfrak{Y}) = \|D_{\mathbb{P}_X}T_\pi - T_\pi D_{\mathbb{P}_Y}\|_{\mathrm{HS}}^2 = \|(D_{\mathbb{P}_X}T_\pi - T_\pi D_{\mathbb{P}_Y})^*\|_{\mathrm{HS}}^2 = \|D_{\mathbb{P}_Y}T_{\varphi(\pi)} - T_{\varphi(\pi)}D_{\mathbb{P}_X}\|_{\mathrm{HS}}^2 = \mathcal{R}(\varphi(\pi); \mathfrak{Y}, \mathfrak{X}).$$

Since $\varphi$ is a bijection between $\Pi(\mathbb{P}_X, \mathbb{P}_Y)$ and $\Pi(\mathbb{P}_Y, \mathbb{P}_X)$, the identity $\mathcal{L}_\alpha(\pi; \mathfrak{X}, \mathfrak{Y}) = \mathcal{L}_\alpha(\varphi(\pi); \mathfrak{Y}, \mathfrak{X})$ implies that the two minima coincide. More precisely, one can derive a contradiction by assuming that there exists $\mu \in \Pi(\mathbb{P}_Y, \mathbb{P}_X)$ such that $\mathcal{L}_\alpha(\mu; \mathfrak{Y}, \mathfrak{X}) < \mathcal{L}_\alpha(\varphi(\pi^*); \mathfrak{Y}, \mathfrak{X})$. Therefore, taking the square roots yields $d_{\mathrm{CT}}^{(\alpha)}(\mathfrak{X}, \mathfrak{Y}) = d_{\mathrm{CT}}^{(\alpha)}(\mathfrak{Y}, \mathfrak{X})$.

**(Triangle inequality).** Let $\pi_{XY} \in \Pi(\mathbb{P}_X, \mathbb{P}_Y)$ and $\pi_{YZ} \in \Pi(\mathbb{P}_Y, \mathbb{P}_Z)$ be optimal couplings for $d_{\mathrm{CT}}^{(\alpha)}(\mathfrak{X}, \mathfrak{Y})$ and $d_{\mathrm{CT}}^{(\alpha)}(\mathfrak{Y}, \mathfrak{Z})$, respectively. Let $\pi_{XY}(dy \mid x)$ and $\pi_{YZ}(dz \mid y)$ be the Markov kernels of $\pi_{XY}$ and $\pi_{YZ}$, respectively. By the disintegration theorem, we can construct a joint coupling $\gamma \in \Pi(\mathbb{P}_X, \mathbb{P}_Y, \mathbb{P}_Z)$ such that

$$\gamma(dx, dy, dz) = \mathbb{P}_X(dx)\, \pi_{XY}(dy \mid x)\, \pi_{YZ}(dz \mid y).$$

Now, denote by $\pi_{XZ}$ the $(X, Z)$-marginal of $\gamma$, i.e.,

$$\pi_{XZ}(dx, dz) = \int_{y \in \mathcal{Y}} \gamma(dx, dy, dz) = \mathbb{P}_X(dx) \underbrace{\int_{y \in \mathcal{Y}} \pi_{YZ}(dz \mid y) \pi_{XY}(dy \mid x)}_{= \pi_{XZ}(dz \mid x)}.$$

According to Definition 3.3, $d_{\mathrm{CT}}^{(\alpha)}(\mathfrak{X}, \mathfrak{Y})$ is bounded above by the $L^2$-norm of a vector in $\mathbb{R}^2$:

$$d_{\mathrm{CT}}^{(\alpha)}(\mathfrak{X}, \mathfrak{Z}) \le \left\| \begin{pmatrix} \sqrt{1-\alpha}\, \mathcal{F}(\pi_{XZ}; \mathfrak{X}, \mathfrak{Z})^{1/2} \\ \sqrt{\alpha/2}\, \mathcal{R}(\pi_{XZ}; \mathfrak{X}, \mathfrak{Z})^{1/2} \end{pmatrix} \right\|_2.$$

For the constructed coupling $\pi_{XZ}$, we apply a triangle-type bound for each component. Under the glued coupling $\gamma \in \Pi(\mathbb{P}_X, \mathbb{P}_Y, \mathbb{P}_Z)$ constructed above, the marginals of $(X, Y)$, $(Y, Z)$, and $(X, Z)$ are $\pi_{XY}$, $\pi_{YZ}$, and $\pi_{XZ}$, respectively. Using $f_{\mathcal{X}}(X) - f_{\mathcal{Z}}(Z) = (f_{\mathcal{X}}(X) - f_{\mathcal{Y}}(Y)) + (f_{\mathcal{Y}}(Y) - f_{\mathcal{Z}}(Z))$ and Minkowski's inequality in $L^2(\gamma)$, we obtain

$$\mathcal{F}(\pi_{XZ}; \mathfrak{X}, \mathfrak{Z})^{1/2} \le \mathcal{F}(\pi_{XY}; \mathfrak{X}, \mathfrak{Y})^{1/2} + \mathcal{F}(\pi_{YZ}; \mathfrak{Y}, \mathfrak{Z})^{1/2}.$$

We show the similar inequality for $\mathcal{R}(\pi_{XZ}; \mathfrak{X}, \mathfrak{Z})$ by leveraging two properties of the conditional expectation operator: (i) the contraction property $\|T_\pi\|_{\mathrm{op}} \le 1$, and (ii) the composition property $T_{\pi_{XZ}} = T_{\pi_{XY}} \circ T_{\pi_{YZ}}$. In fact, the contraction property can be readily shown by using the Jensen inequality: for an arbitrary $g \in L^2(\mathcal{Y}, \mathbb{P}_Y)$,

$$\|T_\pi g\|_{L^2(\mathbb{P}_X)}^2 = \int_{x \in \mathcal{X}} (\mathbb{E}_\pi[g(Y) \mid X = x])^2\, \mathbb{P}_X(dx) \le \int_{x \in \mathcal{X}} \mathbb{E}_\pi[g(Y)^2 \mid X = x] \mathbb{P}_X(dx) = \mathbb{E}_{\mathbb{P}_Y}[g(Y)^2] = \|g\|_{L^2(\mathbb{P}_Y)}^2.$$

In addition, to show the composition property, let $h \in L^2(\mathcal{Z}, \mathbb{P}_Z)$ be arbitrary. Then observe the following:

$$\begin{aligned}
(T_{\pi_{XY}} T_{\pi_{YZ}} h)(x) &= \int_{y \in \mathcal{Y}} \left( \int_{z \in \mathcal{Z}} h(z) \pi_{YZ}(dz \mid y) \right) \pi_{XY}(dy \mid x) \\
&= \int_{\mathcal{Z}} h(z) \pi_{XZ}(dz \mid x) \\
&= (T_{\pi_{XZ}} h)(x), \quad \forall x \in \mathcal{X}.
\end{aligned}$$

Using these properties, we obtain

$$\begin{aligned}
\mathcal{R}(\pi_{XZ}; \mathfrak{X}, \mathfrak{Z})^{1/2} &= \|D_{\mathbb{P}_X} T_{\pi_{XZ}} - T_{\pi_{XZ}} D_{\mathbb{P}_Z}\|_{\mathrm{HS}} \\
&= \|D_{\mathbb{P}_X} T_{\pi_{XY}} T_{\pi_{YZ}} - T_{\pi_{XY}} T_{\pi_{YZ}} D_{\mathbb{P}_Z}\|_{\mathrm{HS}} \quad (\because \text{composition property}) \\
&= \|(D_{\mathbb{P}_X} T_{\pi_{XY}} - T_{\pi_{XY}} D_{\mathbb{P}_Y}) T_{\pi_{YZ}} + T_{\pi_{XY}}(D_{\mathbb{P}_Y} T_{\pi_{YZ}} - T_{\pi_{YZ}} D_{\mathbb{P}_Z})\|_{\mathrm{HS}} \\
&\le \|D_{\mathbb{P}_X} T_{\pi_{XY}} - T_{\pi_{XY}} D_{\mathbb{P}_Y}\|_{\mathrm{HS}} \|T_{\pi_{YZ}}\|_{\mathrm{op}} + \|T_{\pi_{XY}}\|_{\mathrm{op}} \|D_{\mathbb{P}_Y} T_{\pi_{YZ}} - T_{\pi_{YZ}} D_{\mathbb{P}_Z}\|_{\mathrm{HS}} \\
&\le \mathcal{R}(\pi_{XY}; \mathfrak{X}, \mathfrak{Y})^{1/2} + \mathcal{R}(\pi_{YZ}; \mathfrak{Y}, \mathfrak{Z})^{1/2} \quad (\because \text{contraction property}).
\end{aligned}$$

Finally, applying Minkowski's inequality to the vector representation yields:

$$d_{\mathrm{CT}}^{(\alpha)}(\mathfrak{X}, \mathfrak{Z}) \leq \left((1-\alpha)\mathcal{F}(\pi_{XZ}; \mathfrak{X}, \mathfrak{Z}) + \frac{\alpha}{2}\mathcal{R}(\pi_{XZ}; \mathfrak{X}, \mathfrak{Z})\right)^{1/2}$$

$$\leq \left((1-\alpha)\left(\mathcal{F}(\pi_{XY}; \mathfrak{X}, \mathfrak{Y})^{1/2} + \mathcal{F}(\pi_{YZ}; \mathfrak{Y}, \mathfrak{Z})^{1/2}\right)^2 + \frac{\alpha}{2}\left(\mathcal{R}(\pi_{XY}; \mathfrak{X}, \mathfrak{Y})^{1/2} + \mathcal{R}(\pi_{YZ}; \mathfrak{Y}, \mathfrak{Z})^{1/2}\right)^2\right)^{1/2}$$

$$\leq \left((1-\alpha)\mathcal{F}(\pi_{XY}; \mathfrak{X}, \mathfrak{Y}) + \frac{\alpha}{2}\mathcal{R}(\pi_{XY}; \mathfrak{X}, \mathfrak{Y})\right)^{1/2} + \left((1-\alpha)\mathcal{F}(\pi_{YZ}; \mathfrak{Y}, \mathfrak{Z}) + \frac{\alpha}{2}\mathcal{R}(\pi_{YZ}; \mathfrak{Y}, \mathfrak{Z})\right)^{1/2}$$

$$= d_{\mathrm{CT}}^{(\alpha)}(\mathfrak{X}, \mathfrak{Y}) + d_{\mathrm{CT}}^{(\alpha)}(\mathfrak{Y}, \mathfrak{Z}).$$

The first inequality holds from the definition of minimum; the second one is obtained by applying the triangle-type bounds for $\mathcal{F}^{1/2}$ and $\mathcal{R}^{1/2}$; the third is the direct result of Minkowski's inequality (or triangle inequality, equivalently):

$$\|\mathbf{a} + \mathbf{b}\|_2 = ((a_1 + b_1)^2 + (a_2 + b_2)^2)^{1/2} \leq (a_1^2 + a_2^2)^{1/2} + (b_1^2 + b_2^2)^{1/2} = \|\mathbf{a}\|_2 + \|\mathbf{b}\|_2.$$

This concludes the proof. □

### G.5. Proof for Theorem 3.7

*Proof.* First, recall the definitions of $\mathcal{R}(\pi)$ and $\mathcal{R}_{\mathrm{GW},2}(\pi)$: for an arbitrary $\pi \in \Pi(\mathbb{P}_X, \mathbb{P}_Y)$,

$$\mathcal{R}(\pi) = \int_{(x,y) \in \mathcal{X} \times \mathcal{Y}} \left(\int_{x' \in \mathcal{X}} d_{\mathcal{X}}(x, x')\pi_y(dx') - \int_{y' \in \mathcal{Y}} d_{\mathcal{Y}}(y, y')\pi_x(dy')\right)^2 (\mathbb{P}_X \otimes \mathbb{P}_Y)(dx, dy),$$

$$\mathcal{R}_{\mathrm{GW},2}(\pi) = \int_{(x,y) \in \mathcal{X} \times \mathcal{Y}} \int_{(x',y') \in \mathcal{X} \times \mathcal{Y}} |d_{\mathcal{X}}(x, x') - d_{\mathcal{Y}}(y, y')|^2 \pi(dx', dy')\pi(dx, dy).$$

By the symmetry of metrics $d_{\mathcal{X}}$ and $d_{\mathcal{Y}}$, $\mathcal{R}_{\mathrm{GW},2}$ can be equivalently represented as

$$\mathcal{R}_{\mathrm{GW},2}(\pi) = \int_{(x,y') \in \mathcal{X} \times \mathcal{Y}} \int_{(x',y) \in \mathcal{X} \times \mathcal{Y}} |d_{\mathcal{X}}(x, x') - d_{\mathcal{Y}}(y', y)|^2 \pi(dx', dy)\pi(dx, dy')$$

$$= \int_{y \in \mathcal{Y}} \int_{x \in \mathcal{X}} \int_{y' \in \mathcal{Y}} \int_{x' \in \mathcal{X}} |d_{\mathcal{X}}(x, x') - d_{\mathcal{Y}}(y', y)|^2 \pi(dx' \mid y)\pi(dy' \mid x)\mathbb{P}_X(dx)\mathbb{P}_Y(dy)$$

$$= \int_{(x,y) \in \mathcal{X} \times \mathcal{Y}} \left(\int_{(x',y') \in \mathcal{X} \times \mathcal{Y}} |d_{\mathcal{X}}(x, x') - d_{\mathcal{Y}}(y', y)|^2 (\pi_y \otimes \pi_x)(dx', dy')\right) (\mathbb{P}_X \otimes \mathbb{P}_Y)(dx, dy),$$

where the second and third equalities follow from the disintegration theorem and Fubini's theorem.

Now, let $\mu_{x,y} := \pi_y \otimes \pi_x$. Then, by the definition of expectation,

$$\mathbb{E}_{\mu_{x,y}}\left[|d_{\mathcal{X}}(x, X) - d_{\mathcal{Y}}(y, Y)|^2\right] = \int_{(x',y') \in \mathcal{X} \times \mathcal{Y}} |d_{\mathcal{X}}(x, x') - d_{\mathcal{Y}}(y, y')|^2 \mu_{x,y}(dx', dy'),$$

$$\left(\mathbb{E}_{\mu_{x,y}}\left[d_{\mathcal{X}}(x, X) - d_{\mathcal{Y}}(y, Y)\right]\right)^2 = \left(\int_{x' \in \mathcal{X}} d_{\mathcal{X}}(x, x')\pi_y(dx') - \int_{y' \in \mathcal{Y}} d_{\mathcal{Y}}(y, y')\pi_x(dy')\right)^2.$$

Thus, considering the basic fact $\mathbb{E}[Z^2] - (\mathbb{E}[Z])^2 = \mathrm{Var}[Z]$,

$$\mathbb{E}_{\mu_{x,y}}\left[|d_{\mathcal{X}}(x, X) - d_{\mathcal{Y}}(y, Y)|^2\right] - \left(\mathbb{E}_{\mu_{x,y}}\left[d_{\mathcal{X}}(x, X) - d_{\mathcal{Y}}(y, Y)\right]\right)^2 = \mathrm{Var}_{\mu_{x,y}}[d_{\mathcal{X}}(x, X) - d_{\mathcal{Y}}(y, Y)].$$

Moreover, since $X$ and $Y$ are independent in $\mu_{x,y}$,

$$\mathrm{Var}_{\mu_{x,y}}[d_{\mathcal{X}}(x, X) - d_{\mathcal{Y}}(y, Y)] = \mathrm{Var}_{\pi_y}[d_{\mathcal{X}}(x, X)] + \mathrm{Var}_{\pi_x}[d_{\mathcal{Y}}(y, Y)].$$

This completes the proof. □

### G.6. Proof for Theorem 5.6

*Proof.* First, let $\pi_n^* \in \arg\min_{\pi \in \Pi(\hat{\mathbb{P}}_X, \hat{\mathbb{P}}_Y)} \hat{\mathcal{L}}_\alpha(\pi; \mathfrak{X}, \mathfrak{Y})$, and let $\pi^* \in \arg\min_{\pi \in \Pi(\mathbb{P}_X, \mathbb{P}_Y)} \mathcal{L}_\alpha(\pi; \mathfrak{X}, \mathfrak{Y})$ be an optimal coupling that satisfies Assumption 5.5. Then, it follows that

$$
\left| \mathcal{L}_\alpha(\Phi_n(\hat{\pi})) - \mathcal{L}_\alpha(\pi^*) \right|
$$

$$
\leq \left| \mathcal{L}_\alpha(\Phi_n(\hat{\pi})) - \inf_{\pi \in \Pi(\hat{\mathbb{P}}_X, \hat{\mathbb{P}}_Y)} \mathcal{L}_\alpha(\Phi_n(\pi)) \right| + \left| \inf_{\pi \in \Pi(\hat{\mathbb{P}}_X, \hat{\mathbb{P}}_Y)} \mathcal{L}_\alpha(\Phi_n(\pi)) - \mathcal{L}_\alpha(\pi^*) \right|
$$

$$
\leq \left| \mathcal{L}_\alpha(\Phi_n(\hat{\pi})) - \hat{\mathcal{L}}_\alpha(\hat{\pi}) \right| + \left| \hat{\mathcal{L}}_\alpha(\hat{\pi}) - \hat{\mathcal{L}}_\alpha(\pi_n^*) \right|
$$

$$
+ \left| \hat{\mathcal{L}}_\alpha(\pi_n^*) - \inf_{\pi \in \Pi(\hat{\mathbb{P}}_X, \hat{\mathbb{P}}_Y)} \mathcal{L}_\alpha(\Phi_n(\pi)) \right| + \left| \inf_{\pi \in \Pi(\hat{\mathbb{P}}_X, \hat{\mathbb{P}}_Y)} \mathcal{L}_\alpha(\Phi_n(\pi)) - \mathcal{L}_\alpha(\pi^*) \right|
$$

$$
\leq \left| \hat{\mathcal{L}}_\alpha(\hat{\pi}) - \hat{\mathcal{L}}_\alpha(\pi_n^*) \right| + 2 \sup_{\pi \in \Pi(\hat{\mathbb{P}}_X, \hat{\mathbb{P}}_Y)} \left| \hat{\mathcal{L}}_\alpha(\pi) - \mathcal{L}_\alpha(\Phi_n(\pi)) \right| + \left| \inf_{\pi \in \Pi(\hat{\mathbb{P}}_X, \hat{\mathbb{P}}_Y)} \mathcal{L}_\alpha(\Phi_n(\pi)) - \mathcal{L}_\alpha(\pi^*) \right|
$$

$$
=: E_1 + E_2 + E_3.
$$

### Step 1: Bound for $E_1$.

We first provide the bound for $E_1$, which is the optimization error.

**Lemma G.2.** *Let $\{\pi^{(t)} : t \geq 0\}$ be the sequence of iterates generated by Algorithm 1 with*

$$
\gamma_t = \frac{2}{t+2},
$$

*for each $t$. Then, for any $t \geq 1$,*

$$
\hat{\mathcal{L}}_\alpha(\pi^{(t)}) - \hat{\mathcal{L}}_\alpha(\pi_n^*) \leq \frac{8\alpha\, n_{\min}}{t+3} \left( \|D_{\hat{\mathbb{P}}_X}\|_{\mathrm{op}} + \|D_{\hat{\mathbb{P}}_Y}\|_{\mathrm{op}} \right)^2.
$$

*Proof.* See Appendix G.8. $\qquad\square$

Lemma G.2 is a standard result for the convergence of the FW algorithm. Considering that $\|\cdot\|_{\mathrm{op}} \leq \|\cdot\|_\infty$ and that the distance operators are symmetric,

$$
\|D_{\hat{\mathbb{P}}_X}\|_{\mathrm{op}} \leq \max_i \sum_{j=1}^{n_X} [D_{\hat{\mathbb{P}}_X}]_{ij} \leq 1, \quad \|D_{\hat{\mathbb{P}}_Y}\|_{\mathrm{op}} \leq 1.
$$

This gives that

$$
\left( \|D_{\hat{\mathbb{P}}_X}\|_{\mathrm{op}} + \|D_{\hat{\mathbb{P}}_Y}\|_{\mathrm{op}} \right)^2 \leq 4.
$$

Thus,

$$
E_1 \leq \frac{32\alpha\, n_{\min}}{T+3}. \tag{4}
$$

### Step 2: Bound for $E_2$.

We now look into $E_2$. Let $\pi \in \Pi(\hat{\mathbb{P}}_X, \hat{\mathbb{P}}_Y)$ be arbitrary. Note that $\Phi_n(\pi)$ can be represented as the following form:

$$
\Phi_n(\pi)(dx, dy) = \int_{(\hat{x}, \hat{y}) \in \mathcal{X} \times \mathcal{Y}} Q_X(dx \mid \hat{x})\, Q_Y(dy \mid \hat{y})\, \pi(d\hat{x}, d\hat{y}) = \sum_{i=1}^{n_X} \sum_{j=1}^{n_Y} \pi_{ij} Q_X(dx \mid X_i) Q_Y(dy \mid Y_j). \tag{5}
$$

Precisely, we may define the Markov kernels $Q_Y(dy \mid \hat{y})$ and $Q_X(dx \mid \hat{x})$ by using the Laguerre lemma F.5:

$$Q_X(dx \mid \hat{x}) = n_X \sum_{i=1}^{n_X} \mathbb{P}_X \lfloor_{V_i}(dx) \mathbb{I}_{\{X_i\}}(\hat{x}), \quad Q_Y(dy \mid \hat{y}) = n_Y \sum_{j=1}^{n_Y} \mathbb{P}_Y \lfloor_{W_j}(dy) \mathbb{I}_{\{Y_j\}}(\hat{y}), \tag{6}$$

where $V_i$ and $W_j$ are the $i$-th and the $j$-th Laguerre cells defined for $Q_X$ and $Q_Y$, respectively. In fact, $\Phi_n$ is the marginal of a coupling $\xi \in \Pi(\mathbb{P}_X, \hat{\mathbb{P}}_X, \hat{\mathbb{P}}_Y, \mathbb{P}_Y)$ defined by

$$\xi(dx, d\hat{x}, d\hat{y}, dy) := \underbrace{Q_Y(dy \mid \hat{y})}_{\hat{\mathbb{P}}_Y \to \mathbb{P}_Y} \underbrace{Q_X(dx \mid \hat{x})}_{\hat{\mathbb{P}}_X \to \mathbb{P}_X} \pi(d\hat{x}, d\hat{y}).$$

In addition, plugging (6) into (5) yields that

$$\Phi_n(\pi)(dx \mid y) = n_Y \sum_{i=1}^{n_X} \sum_{j=1}^{n_Y} \pi_{ij} Q_X(dx \mid X_i) \mathbb{I}_{W_j}(y) = n_Y \sum_{i=1}^{n_X} \sum_{j=1}^{n_Y} \pi_{ij} \mathbb{I}_{W_j}(y) \left( n_X \mathbb{P}_X \lfloor_{V_i}(dx) \right) \tag{7}$$

$$\Phi_n(\pi)(dy \mid x) = n_X \sum_{i=1}^{n_X} \sum_{j=1}^{n_Y} \pi_{ij} Q_Y(dy \mid Y_j) \mathbb{I}_{V_i}(x) = n_X \sum_{i=1}^{n_X} \sum_{j=1}^{n_Y} \pi_{ij} \mathbb{I}_{V_i}(x) \left( n_Y \mathbb{P}_Y \lfloor_{W_j}(dy) \right). \tag{8}$$

For brevity, let $c_f(x, y) := \|f_{\mathcal{X}}(x) - f_{\mathcal{Y}}(y)\|_2^2$. Then,

$$\begin{aligned}
&|c_f(x, y) - c_f(x', y')| \\
&\leq \left( \|f_{\mathcal{X}}(x) - f_{\mathcal{Y}}(y)\|_2 + \|f_{\mathcal{X}}(x') - f_{\mathcal{Y}}(y')\|_2 \right) \left| \|f_{\mathcal{X}}(x) - f_{\mathcal{Y}}(y)\|_2 - \|f_{\mathcal{X}}(x') - f_{\mathcal{Y}}(y')\|_2 \right| \\
&\leq 2 \left( \|f_{\mathcal{X}}(x) - f_{\mathcal{X}}(x')\|_2 + \|f_{\mathcal{Y}}(y) - f_{\mathcal{Y}}(y')\|_2 \right) \\
&\leq 2L_f \left( d_{\mathcal{X}}(x, x') + d_{\mathcal{Y}}(y, y') \right).
\end{aligned}$$

The last inequality uses Assumption 5.4 and the fact that $\mathrm{diam}(\mathcal{M}) = 1$. Note that this confirms that $c_f$ is a bounded and $2L_f$-Lipschitz function with respect to the metric

$$d_{\oplus}((x, y), (x', y')) := d_{\mathcal{X}}(x, x') + d_{\mathcal{Y}}(y, y').$$

Thus, by the duality formula of the Wasserstein-1 distance (Lemma F.7),

$$\left| \mathbb{E}_{(\hat{X}, \hat{Y}) \sim \pi}[c_f(\hat{X}, \hat{Y})] - \mathbb{E}_{(X, Y) \sim \Phi_n(\pi)}[c_f(X, Y)] \right| \leq 2L_f W_1^{d_{\oplus}}(\pi, \Phi_n(\pi)).$$

Here $W_1^{d_{\oplus}}(\pi, \Phi_n(\pi))$ can be decomposed as follows:

$$\begin{aligned}
W_1^{d_{\oplus}}(\pi, \Phi_n(\pi)) &= \inf_{\mu \in \Pi(\pi, \Phi_n(\pi))} \mathbb{E}_{((\hat{X}, \hat{Y}), (X, Y)) \sim \mu} \left[ d_{\mathcal{X}}(X, \hat{X}) + d_{\mathcal{Y}}(Y, \hat{Y}) \right] \\
&\leq \mathbb{E}_{(X, \hat{X}, \hat{Y}, Y) \sim \xi} \left[ d_{\mathcal{X}}(X, \hat{X}) + d_{\mathcal{Y}}(Y, \hat{Y}) \right] \\
&= \mathbb{E}_{(X, \hat{X}) \sim Q_X} \left[ d_{\mathcal{X}}(X, \hat{X}) \right] + \mathbb{E}_{(\hat{Y}, Y) \sim Q_Y} \left[ d_{\mathcal{Y}}(Y, \hat{Y}) \right] \\
&= W_1^{d_{\mathcal{X}}}(\mathbb{P}_X, \hat{\mathbb{P}}_X) + W_1^{d_{\mathcal{Y}}}(\mathbb{P}_Y, \hat{\mathbb{P}}_Y). \tag{9}
\end{aligned}$$

The second inequality follows from the definition of infimum; the last equality holds since $Q_X$ and $Q_Y$ are the optimal couplings that minimize $W_1^{d_{\mathcal{X}}}(\mathbb{P}_X, \hat{\mathbb{P}}_X)$ and $W_1^{d_{\mathcal{Y}}}(\mathbb{P}_Y, \hat{\mathbb{P}}_Y)$, respectively.

We now analyze the structural regularization term:

$$
\left| \| D_{\hat{\mathbb{P}}_X} T_\pi - T_\pi D_{\hat{\mathbb{P}}_Y} \|_{\mathrm{HS}}^2 - \| D_{\mathbb{P}_X} T_{\Phi_n(\pi)} - T_{\Phi_n(\pi)} D_{\mathbb{P}_Y} \|_{\mathrm{HS}}^2 \right|
$$

$$
= \left| \int_{\mathcal{Y}} \int_{\mathcal{X}} \Gamma_\pi(x,y)^2 \hat{\mathbb{P}}_X(dx)\hat{\mathbb{P}}_Y(dy) - \int_{\mathcal{Y}} \int_{\mathcal{X}} \Gamma_{\Phi_n(\pi)}(x,y)^2 \mathbb{P}_X(dx)\mathbb{P}_Y(dy) \right|
$$

$$
\leq \underbrace{\left| \int_{\mathcal{Y}} \int_{\mathcal{X}} (\Gamma_\pi(x,y)^2 - \Gamma_{\Phi_n(\pi)}(x,y)^2) \hat{\mathbb{P}}_X(dx)\hat{\mathbb{P}}_Y(dy) \right|}_{=:(A)}
$$

$$
+ \underbrace{\left| \int_{\mathcal{Y}} \int_{\mathcal{X}} \Gamma_{\Phi_n(\pi)}(x,y)^2 (\hat{\mathbb{P}}_X(dx)\hat{\mathbb{P}}_Y(dy) - \mathbb{P}_X(dx)\mathbb{P}_Y(dy)) \right|}_{=:(B)}.
$$

Considering that $|\Gamma_\pi| \leq 1$, $(A)$ is bounded above by

$$
(A) \leq 2 \int_{\mathcal{Y}} \int_{\mathcal{X}} \left| \Gamma_\pi(x,y) - \Gamma_{\Phi_n(\pi)}(x,y) \right| \hat{\mathbb{P}}_X(dx)\hat{\mathbb{P}}_Y(dy)
$$

$$
= \frac{2}{n_X n_Y} \sum_{i,j} \left| \Gamma_\pi(X_i, Y_j) - \Gamma_{\Phi_n(\pi)}(X_i, Y_j) \right|
$$

$$
\leq \frac{2}{n_X n_Y} \sum_{i,j} \left| \Gamma_\pi^{(1)}(X_i, Y_j) - \Gamma_{\Phi_n(\pi)}^{(1)}(X_i, Y_j) \right| + \frac{2}{n_X n_Y} \sum_{i,j} \left| \Gamma_\pi^{(2)}(X_i, Y_j) - \Gamma_{\Phi_n(\pi)}^{(2)}(X_i, Y_j) \right|
$$

Here $\Gamma_\pi^{(1)}(X_i, Y_j)$ and $\Gamma_{\Phi_n(\pi)}^{(1)}(X_i, Y_j)$ are well-defined as

$$
\Gamma_\pi^{(1)}(X_i, Y_j) = n_Y \sum_{i'=1}^{n_X} \pi_{i'j} d_{\mathcal{X}}(X_i, X_{i'}),
$$

$$
\Gamma_{\Phi_n(\pi)}^{(1)}(X_i, Y_j) = \int_{x \in \mathcal{X}} d_{\mathcal{X}}(X_i, x) \Phi_n(dx \mid Y_j) = n_Y \sum_{i'=1}^{n_X} \pi_{i'j} \int_{x \in V_{i'}} d_{\mathcal{X}}(X_i, x) \Big( n_X \mathbb{P}_X(dx) \Big),
$$

where $\Phi_n(dx \mid Y_j)$ is as defined in (7). A similar technique also ensures the well-definedness of $\Gamma_\pi^{(2)}(X_i, Y_j)$ and $\Gamma_{\Phi_n(\pi)}^{(2)}(X_i, Y_j)$. Then,

$$
\left| \Gamma_\pi(X_i, Y_j) - \Gamma_{\Phi_n(\pi)}(X_i, Y_j) \right|
$$

$$
\leq n_X n_Y \sum_{i'=1}^{n_X} \pi_{i'j} \int_{x \in V_{i'}} |d_{\mathcal{X}}(X_i, x) - d_{\mathcal{X}}(X_i, X_{i'})| \, \mathbb{P}_X(dx)
$$

$$
+ n_X n_Y \sum_{j'=1}^{n_Y} \pi_{ij'} \int_{y \in W_{j'}} |d_{\mathcal{Y}}(Y_j, y) - d_{\mathcal{Y}}(Y_j, Y_{j'})| \, \mathbb{P}_Y(dy).
$$

Finally, we get

$$
(A) \leq 2 \sum_{i,j,i'} \pi_{i'j} \int_{V_{i'}} \left| d_{\mathcal{X}}(X_i, x) - d_{\mathcal{X}}(X_i, X_{i'}) \right| \mathbb{P}_X(dx) + 2 \sum_{i,j,j'} \pi_{ij'} \int_{W_{j'}} \left| d_{\mathcal{Y}}(Y_j, y) - d_{\mathcal{Y}}(Y_j, Y_{j'}) \right| \mathbb{P}_Y(dy)
$$

$$
\leq \frac{2}{n_X} \sum_{i,i'} \int_{V_{i'}} \left| d_{\mathcal{X}}(X_i, x) - d_{\mathcal{X}}(X_i, X_{i'}) \right| \mathbb{P}_X(dx) + \frac{2}{n_Y} \sum_{j,j'} \int_{W_{j'}} \left| d_{\mathcal{Y}}(Y_j, y) - d_{\mathcal{Y}}(Y_j, Y_{j'}) \right| \mathbb{P}_Y(dy)
$$

$$
\leq 2 \sum_{i'=1}^{n_X} \int_{V_{i'}} d_{\mathcal{X}}(x, X_{i'}) \, \mathbb{P}_X(dx) + 2 \sum_{j'=1}^{n_Y} \int_{W_{j'}} d_{\mathcal{Y}}(y, Y_{j'}) \, \mathbb{P}_Y(dy)
$$

$$
= 2 \big( W_1^{d_{\mathcal{X}}}(\mathbb{P}_X, \hat{\mathbb{P}}_X) + W_1^{d_{\mathcal{Y}}}(\mathbb{P}_Y, \hat{\mathbb{P}}_Y) \big).
$$

We now move onto $(B)$:

$$(B) \leq \left| \int_{\mathcal{X}} \left( \int_{\mathcal{Y}} \Gamma_{\Phi_n(\pi)}(x,y)^2 \hat{\mathbb{P}}_Y(dy) \right) (\hat{\mathbb{P}}_X - \mathbb{P}_X)(dx) \right|$$
$$+ \left| \int_{\mathcal{Y}} \left( \int_{\mathcal{X}} \Gamma_{\Phi_n(\pi)}(x,y)^2 \mathbb{P}_X(dx) \right) (\hat{\mathbb{P}}_Y - \mathbb{P}_Y)(dy) \right|.$$

Similarly, it suffices to bound the first term by symmetry. Let

$$\gamma(x) := \int_{\mathcal{Y}} \Gamma_{\Phi_n(\pi)}(x,y)^2 \hat{\mathbb{P}}_Y(dy).$$

Then, it follows that

$$\left| \int_{\mathcal{X}} \left( \int_{\mathcal{Y}} \Gamma_{\Phi_n(\pi)}(x,y)^2 \hat{\mathbb{P}}_Y(dy) \right) (\hat{\mathbb{P}}_X - \mathbb{P}_X)(dx) \right| = \left| \mathbb{E}_{\hat{X} \sim \hat{\mathbb{P}}_X}[\gamma(\hat{X})] - \mathbb{E}_{X \sim \mathbb{P}_X}[\gamma(X)] \right|$$
$$= \left| \int_{\mathcal{X} \times \mathcal{X}} (\gamma(\hat{x}) - \gamma(x)) Q_X(dx, d\hat{x}) \right|$$
$$\leq \int_{\mathcal{X} \times \mathcal{X}} |\gamma(\hat{x}) - \gamma(x)| \, Q_X(dx, d\hat{x}).$$

By the fact that $|\Gamma_{\Phi_n(\pi)}| \leq 1$, we have

$$|\gamma(\hat{x}) - \gamma(x)| \leq 2 \int_{\mathcal{Y}} \left| \Gamma_{\Phi_n(\pi)}(\hat{x},y) - \Gamma_{\Phi_n(\pi)}(x,y) \right| \hat{\mathbb{P}}_Y(dy).$$

Moreover,

$$\int_{\mathcal{Y}} \left| \Gamma_{\Phi_n(\pi)}(\hat{x},y) - \Gamma_{\Phi_n(\pi)}(x,y) \right| \hat{\mathbb{P}}_Y(dy)$$
$$\leq \frac{1}{n_Y} \sum_{j=1}^{n_Y} \int_{x' \in \mathcal{X}} |d_{\mathcal{X}}(\hat{x},x') - d_{\mathcal{X}}(x,x')| \Phi_n(\pi)(dx' \mid Y_j)$$
$$+ \frac{1}{n_Y} \sum_{j=1}^{n_Y} \left| \int_{y' \in \mathcal{Y}} d_{\mathcal{Y}}(Y_j,y') \Phi_n(\pi)(dy' \mid \hat{x}) - \int_{y' \in \mathcal{Y}} d_{\mathcal{Y}}(Y_j,y') \Phi_n(dy' \mid x) \right|.$$

For $(x, \hat{x}) \sim Q_X$, we have $\hat{x} = X_i$ and $x \in V_i$ for some $i$ with probability 1, meaning that both $x$ and $\hat{x}$ belong to the same cell $V_i$. We recall the form of $\Phi_n(\pi)(dy \mid x)$ in (8):

$$\Phi_n(\pi)(dy \mid x) = n_X \sum_{i=1}^{n_X} \sum_{j=1}^{n_Y} \pi_{ij} \mathbb{I}_{V_i}(x) \left( n_Y \mathbb{P}_Y \lfloor_{W_j}(dy) \right).$$

As a result, $\Phi_n(\cdot \mid x)$ depends on $x$ only through the indicator $\mathbb{I}_{V_i}(x)$. Since $\mathbb{I}_{V_i}(x) = \mathbb{I}_{V_i}(\hat{x}) = 1$, the second term in the above upper bound vanishes. In addition, the first term is bounded by $d_{\mathcal{X}}(\hat{x},x)$ by the triangle inequality. This yields that

$$\int_{\mathcal{X} \times \mathcal{X}} |\gamma(\hat{x}) - \gamma(x)| \, Q_X(dx, d\hat{x}) \leq 2 \int_{\mathcal{X} \times \mathcal{X}} d_{\mathcal{X}}(\hat{x},x) Q_X(dx, d\hat{x}) = 2 W_1^{d_{\mathcal{X}}}(\mathbb{P}_X, \hat{\mathbb{P}}_X).$$

Thus,

$$(B) \leq 2 \left( W_1^{d_{\mathcal{X}}}(\mathbb{P}_X, \hat{\mathbb{P}}_X) + W_1^{d_{\mathcal{Y}}}(\mathbb{P}_Y, \hat{\mathbb{P}}_Y) \right),$$

which concludes that

$$E_2 \leq 4(L_f + 2)(W_1^{d_{\mathcal{X}}}(\mathbb{P}_X, \hat{\mathbb{P}}_X) + W_1^{d_{\mathcal{Y}}}(\mathbb{P}_Y, \hat{\mathbb{P}}_Y)). \tag{10}$$

**Step 3:** Bound for $E_3$.

To bound $E_3$, let $Q_X \in \Pi(\mathbb{P}_X, \hat{\mathbb{P}}_X)$ and $Q_Y \in \Pi(\mathbb{P}_Y, \hat{\mathbb{P}}_Y)$ be optimal couplings attaining $W_1^{d_{\mathcal{X}}}(\mathbb{P}_X, \hat{\mathbb{P}}_X)$ and $W_1^{d_{\mathcal{Y}}}(\mathbb{P}_Y, \hat{\mathbb{P}}_Y)$, respectively. Define $\Psi_n : \Pi(\mathbb{P}_X, \mathbb{P}_Y) \to \Pi(\hat{\mathbb{P}}_X, \hat{\mathbb{P}}_Y)$ by

$$\Psi_n(\sigma)(d\hat{x}, d\hat{y}) := \int_{\mathcal{X} \times \mathcal{Y}} Q_X(d\hat{x} \mid x) \, Q_Y(d\hat{y} \mid y) \, \sigma(dx, dy), \qquad \sigma \in \Pi(\mathbb{P}_X, \mathbb{P}_Y).$$

Since $\Phi_n(\Pi(\hat{\mathbb{P}}_X, \hat{\mathbb{P}}_Y)) \subset \Pi(\mathbb{P}_X, \mathbb{P}_Y)$, we may define

$$\tau := \Phi_n\big(\Psi_n(\pi^*)\big) \in \Pi(\mathbb{P}_X, \mathbb{P}_Y),$$

and thus

$$0 \le E_3 \le \mathcal{L}_\alpha(\tau) - \mathcal{L}_\alpha(\pi^*) = \big|\mathcal{L}_\alpha(\tau) - \mathcal{L}_\alpha(\pi^*)\big|.$$

Analogously to (9), for any $\pi \in \Pi(\mathbb{P}_X, \mathbb{P}_Y)$ we have

$$\Big|\mathbb{E}_\pi[c_f(X, Y)] - \mathbb{E}_{\Phi_n(\Psi_n(\pi))}[c_f(X', Y')]\Big| \le 4L_f\Big(W_1^{d_{\mathcal{X}}}(\mathbb{P}_X, \hat{\mathbb{P}}_X) + W_1^{d_{\mathcal{Y}}}(\mathbb{P}_Y, \hat{\mathbb{P}}_Y)\Big),$$

and in particular this holds for $\pi = \pi^*$. For the structural regularization term, using (1) and $|\Gamma_\pi| \le 1$, we obtain

$$\Big| \|D_{\mathbb{P}_X} T_{\pi^*} - T_{\pi^*} D_{\mathbb{P}_Y}\|_{\mathrm{HS}}^2 - \|D_{\mathbb{P}_X} T_\tau - T_\tau D_{\mathbb{P}_Y}\|_{\mathrm{HS}}^2 \Big|$$

$$\le \int_{\mathcal{Y}} \int_{\mathcal{X}} \big|\Gamma_{\pi^*}(x, y)^2 - \Gamma_\tau(x, y)^2\big| \, \mathbb{P}_X(dx) \, \mathbb{P}_Y(dy)$$

$$\le 2 \int_{\mathcal{Y}} \int_{\mathcal{X}} \big|\Gamma_{\pi^*}(x, y) - \Gamma_\tau(x, y)\big| \, \mathbb{P}_X(dx) \, \mathbb{P}_Y(dy). \tag{11}$$

We bound the integrand by splitting $\Gamma = \Gamma^{(1)} - \Gamma^{(2)}$: for $\mathbb{P}_X \otimes \mathbb{P}_Y$-a.e. $(x, y)$,

$$\big|\Gamma_{\pi^*}(x, y) - \Gamma_\tau(x, y)\big| \le \big|\Gamma_{\pi^*}^{(1)}(x, y) - \Gamma_\tau^{(1)}(x, y)\big| + \big|\Gamma_{\pi^*}^{(2)}(x, y) - \Gamma_\tau^{(2)}(x, y)\big|.$$

We only bound the $\Gamma^{(1)}$-part; the $\Gamma^{(2)}$-part follows by symmetry.

To bound $\big|\Gamma_{\pi^*}^{(1)}(x, y) - \Gamma_\tau^{(1)}(x, y)\big|$, we introduce an intermediate term by *cell-averaging* $\Gamma_{\pi^*}^{(1)}(x, \cdot)$ along the Laguerre partition induced by the $W_1$-optimal coupling $Q_Y$. By the Laguerre lemma, we may choose a partition $\{W_j\}_{j=1}^{n_Y}$ such that

$$Q_Y(d\hat{y} \mid y) = \sum_{j=1}^{n_Y} \mathbf{1}_{W_j}(y) \, \delta_{Y_j}(d\hat{y}) \quad (\mathbb{P}_Y\text{-a.e. } y), \quad \mathbb{P}_Y(W_j) = \frac{1}{n_Y}.$$

Define

$$\bar{\Gamma}_{\pi^*}^{(1)}(x, y) := \sum_{j=1}^{n_Y} \mathbf{1}_{W_j}(y) \left( \frac{1}{\mathbb{P}_Y(W_j)} \int_{W_j} \Gamma_{\pi^*}^{(1)}(x, u) \, \mathbb{P}_Y(du) \right) = \sum_{j=1}^{n_Y} \mathbf{1}_{W_j}(y) \left( n_Y \int_{W_j} \Gamma_{\pi^*}^{(1)}(x, u) \, \mathbb{P}_Y(du) \right).$$

**(I) Oscillation induced by $Q_Y$.** Recall that $Q_Y \in \Pi(\mathbb{P}_Y, \hat{\mathbb{P}}_Y)$ is an optimal coupling for $W_1^{d_{\mathcal{Y}}}$.

Then, for $\mathbb{P}_Y$-a.e. $y \in W_j$, using Assumption 5.5, we have

$$\big|\Gamma_{\pi^*}^{(1)}(x, y) - \bar{\Gamma}_{\pi^*}^{(1)}(x, y)\big| = \left|\Gamma_{\pi^*}^{(1)}(x, y) - n_Y \int_{W_j} \Gamma_{\pi^*}^{(1)}(x, u) \, \mathbb{P}_Y(du)\right|$$

$$\le n_Y \int_{W_j} \big|\Gamma_{\pi^*}^{(1)}(x, y) - \Gamma_{\pi^*}^{(1)}(x, u)\big| \, \mathbb{P}_Y(du)$$

$$\le L_W \, n_Y \int_{W_j} d_{\mathcal{Y}}(y, u) \, \mathbb{P}_Y(du).$$

Integrating over $(x, y) \sim \mathbb{P}_X \otimes \mathbb{P}_Y$ and summing over the cells yields

$$\int_{\mathcal{X} \times \mathcal{Y}} |\Gamma_{\pi^*}^{(1)}(x, y) - \bar{\Gamma}_{\pi^*}^{(1)}(x, y)| (\mathbb{P}_X \otimes \mathbb{P}_Y)(dx, dy) \leq L_W \sum_{j=1}^{n_Y} \int_{\mathcal{X}} \int_{W_j} n_Y \int_{W_j} d_{\mathcal{Y}}(y, u) \, \mathbb{P}_Y(du) \, \mathbb{P}_Y(dy) \, \mathbb{P}_X(dx)$$

$$= L_W \, n_Y \sum_{j=1}^{n_Y} \int_{W_j \times W_j} d_{\mathcal{Y}}(y, u) \, \mathbb{P}_Y(dy) \mathbb{P}_Y(du)$$

$$= 2 L_W \, \mathbb{E}_{(Y, \hat{Y}) \sim Q_Y} [d_{\mathcal{Y}}(Y, \hat{Y})]$$

$$= 2 L_W \, W_1^{d_{\mathcal{Y}}}(\mathbb{P}_Y, \hat{\mathbb{P}}_Y). \tag{12}$$

The third equality follows by rewriting the within-cell double integral as an expectation under the coupling $Q_Y$, using the representation

$$Q_Y(dy, d\hat{y}) = \sum_{j=1}^{n_Y} \mathbb{P}_Y \lfloor_{W_j}(dy) \, \delta_{Y_j}(d\hat{y}),$$

together with the inequality

$$d_{\mathcal{Y}}(y, u) \leq d_{\mathcal{Y}}(y, Y_j) + d_{\mathcal{Y}}(Y_j, u), \quad y, u \in W_j.$$

**(II) Bias induced by $Q_X$ inside each $W_j$.** Recall that $Q_X \in \Pi(\mathbb{P}_X, \hat{\mathbb{P}}_X)$ is an optimal coupling for $W_1^{d_{\mathcal{X}}}$. Similarly, a Markov kernel $Q_X(d\hat{x} \mid x)$ can be chosen as:

$$Q_X(d\hat{x} \mid x) = \sum_{i=1}^{n_X} \mathbb{I}_{V_i}(x) \delta_{X_i}(d\hat{x}).$$

Define the composed Markov kernel

$$K_X(dx' \mid x) := \int_{\hat{x} \in \mathcal{X}} Q_X(dx' \mid \hat{x}) \, Q_X(d\hat{x} \mid x).$$

Let $\tau := \Phi_n(\Psi_n(\pi^*))$. For each $j$, define the normalized $X$-marginals on the strip $\mathcal{X} \times W_j$:

$$\pi_j^{*, X}(dx) := n_Y \, \pi^*(dx, W_j), \qquad \tau_j^X(dx) := n_Y \, \tau(dx, W_j).$$

Then for any Borel set $A \subset \mathcal{X}$,

$$\tau_j^X(A) = n_Y \, \tau(A \times W_j)$$

$$= n_Y \int_{(\hat{x}, \hat{y}) \in \mathcal{X} \times \mathcal{Y}} Q_X(A \mid \hat{x}) Q_Y(W_j \mid \hat{y}) \Psi_n(\pi^*)(d\hat{x}, d\hat{y})$$

$$= n_Y \sum_{i=1}^{n_X} \sum_{k=1}^{n_Y} \Psi_n(\pi^*)(\{X_i\} \times \{Y_k\}) Q_X(A \mid X_i) Q_Y(W_j \mid Y_k)$$

$$= n_Y \sum_{i=1}^{n_X} \Psi_n(\pi^*)(\{X_i\} \times \{Y_j\}) \, Q_X(A \mid X_i)$$

$$= n_Y \sum_{i=1}^{n_X} \pi^*(V_i \times W_j) \, Q_X(A \mid X_i) = \int_{\mathcal{X}} K_X(A \mid x) \, \pi_j^{*, X}(dx),$$

where the last identity uses that $K_X(\cdot \mid x) = Q_X(\cdot \mid X_i)$ for $x \in V_i$ under the Laguerre representation of $Q_X$. Hence $\tau_j^X = \pi_j^{*, X} K_X$.

Consequently, for $\mathbb{P}_Y$-a.e. $y \in W_j$,

$$\bar{\Gamma}_{\pi^*}^{(1)}(x, y) = \int_{\mathcal{X}} d_{\mathcal{X}}(x, u) \, \pi_j^{*, X}(du), \qquad \Gamma_{\tau}^{(1)}(x, y) = \int_{\mathcal{X}} d_{\mathcal{X}}(x, u) \, \tau_j^X(du).$$

Using $|d_{\mathcal{X}}(x, u) - d_{\mathcal{X}}(x, v)| \leq d_{\mathcal{X}}(u, v)$ and the coupling $\eta_j(du, dv) := \pi_j^{*,X}(du) K_X(dv \mid u) \in \Pi(\pi_j^{*,X}, \tau_j^X)$, we obtain

$$\left| \bar{\Gamma}_{\pi^*}^{(1)}(x, y) - \Gamma_\tau^{(1)}(x, y) \right| \leq \int_{\mathcal{X} \times \mathcal{X}} d_{\mathcal{X}}(u, v) \, \eta_j(du, dv).$$

Integrating over $(x, y) \sim \mathbb{P}_X \otimes \mathbb{P}_Y$ and averaging over $j$ yields

$$\int_{\mathcal{X} \times \mathcal{Y}} \left| \bar{\Gamma}_{\pi^*}^{(1)}(x, y) - \Gamma_\tau^{(1)}(x, y) \right| (\mathbb{P}_X \otimes \mathbb{P}_Y)(dx, dy) \leq \mathbb{E}_{K_X \mathbb{P}_X} \left[ d_{\mathcal{X}}(X, X') \right],$$

where $K_X \mathbb{P}_X$ denotes the joint distribution $K_X(dx' \mid x) \mathbb{P}_X(dx)$. Construct a joint coupling $\nu \in \Pi(\mathbb{P}_X, \hat{\mathbb{P}}_X, \mathbb{P}_X)$ by

$$\nu(dx, d\hat{x}, dx') := Q_X(dx, d\hat{x}) Q_X(dx' \mid \hat{x}) = Q_X(d\hat{x} \mid x) Q_X(dx' \mid \hat{x}) \mathbb{P}_X(dx).$$

Then, for $(X, \hat{X}, X') \sim \nu$, the marginals of $(X, \hat{X})$, $(X, X')$, and $(X', \hat{X})$ are $Q_X(dx, d\hat{x})$, $K_X(dx' \mid x)\mathbb{P}_X(dx)$, and $Q_X(dx', d\hat{x})$, respectively. By applying the triangle inequality,

$$\mathbb{E}_{K_X \mathbb{P}_X} \left[ d_{\mathcal{X}}(X, X') \right] \leq \mathbb{E}_\nu \left[ d_{\mathcal{X}}(X, \hat{X}) \right] + \mathbb{E}_\nu \left[ d_{\mathcal{X}}(\hat{X}, X') \right] = 2 W_1^{d_{\mathcal{X}}}(\mathbb{P}_X, \hat{\mathbb{P}}_X). \tag{13}$$

Combining (12) and (13), we conclude

$$\int_{\mathcal{X} \times \mathcal{Y}} \left| \Gamma_{\pi^*}^{(1)}(x, y) - \Gamma_\tau^{(1)}(x, y) \right| (\mathbb{P}_X \otimes \mathbb{P}_Y)(dx, dy) \leq 2 W_1^{d_{\mathcal{X}}}(\mathbb{P}_X, \hat{\mathbb{P}}_X) + L_W \, W_1^{d_{\mathcal{Y}}}(\mathbb{P}_Y, \hat{\mathbb{P}}_Y).$$

Consequently, we arrive at

$$E_3 \leq 4(L_f + 2L_W + 2) \left( W_1^{d_{\mathcal{X}}}(\mathbb{P}_X, \hat{\mathbb{P}}_X) + W_1^{d_{\mathcal{Y}}}(\mathbb{P}_Y, \hat{\mathbb{P}}_Y) \right). \tag{14}$$

Combining (4), (10), and (14), we obtain

$$\left| \mathcal{L}_\alpha(\Phi_n(\hat{\pi})) - \mathcal{L}_\alpha(\pi^*) \right| \leq \frac{32\alpha \, n_{\min}}{T + 3} + 4(2L_f + 2L_W + 4)(W_1^{d_{\mathcal{X}}}(\mathbb{P}_X, \hat{\mathbb{P}}_X) + W_1^{d_{\mathcal{Y}}}(\mathbb{P}_Y, \hat{\mathbb{P}}_Y)).$$

This completes the proof. $\square$

### G.7. Proof for Lemma G.1

*Proof.* The first term is clearly continuous w.r.t. $\pi$: since $\|f_{\mathcal{X}}(x) - f_{\mathcal{Y}}(y)\|_2^2 \in C_b(\mathcal{X} \times \mathcal{Y})$, $\pi_n \xrightarrow{w} \pi$ implies

$$\mathbb{E}_{\pi_n} \left[ \|f_{\mathcal{X}}(X) - f_{\mathcal{Y}}(Y)\|_2^2 \right] = \int_{\mathcal{X} \times \mathcal{Y}} \|f_{\mathcal{X}}(x) - f_{\mathcal{Y}}(y)\|_2^2 \pi_n(dx, dy) \to \mathbb{E}_\pi \left[ \|f_{\mathcal{X}}(X) - f_{\mathcal{Y}}(Y)\|_2^2 \right].$$

To show that $\mathcal{R}(\pi)$ is lower semicontinuous, we first show that $\pi \mapsto T_\pi$ is continuous. Let $\{\pi_n\}_{n=1}^\infty$ be a sequence that weakly converges to some $\pi$.

Note that $C(\mathcal{X}) \subset L^2(\mathcal{X}, \mathbb{P}_X)$ and $C(\mathcal{Y}) \subset L^2(\mathcal{Y}, \mathbb{P}_Y)$ are both dense due to the compactness of $\mathcal{X}$ and $\mathcal{Y}$. In particular, $C(\mathcal{X}) = C_b(\mathcal{X})$ and $C(\mathcal{Y}) = C_b(\mathcal{Y})$. Thus, for $f \in C(\mathcal{X})$ and $g \in C(\mathcal{Y})$,

$$\langle T_{\pi_n} g, f \rangle_{L^2(\mathbb{P}_X)} = \int f(x) g(y) \pi_n(dx, dy) \to \int f(x) g(y) \pi(dx, dy) = \langle T_\pi g, f \rangle_{L^2(\mathbb{P}_X)}, \quad n \to \infty, \tag{15}$$

which follows from the definition of weak convergence. Since the conditional expectation operators are contractions (i.e., $\|T_{\pi_n}\|_{\mathrm{op}} \leq 1$ for all $n$), the sequence is uniformly bounded. By the density of continuous functions in $L^2$ and the uniform boundedness of $\{T_{\pi_n}\}$, (15) extends to all $f \in L^2(\mathcal{X}, \mathbb{P}_X)$ and $g \in L^2(\mathcal{Y}, \mathbb{P}_Y)$. Thus, $\pi_n \xrightarrow{w} \pi$ implies $T_{\pi_n} \to T_\pi$ in WOT, which also entails

$$D_{\mathbb{P}_X} T_{\pi_n} - T_{\pi_n} D_{\mathbb{P}_Y} \to D_{\mathbb{P}_X} T_\pi - T_\pi D_{\mathbb{P}_Y}, \quad \text{in WOT.}$$

Therefore, by Lemma F.12, we conclude that $\mathcal{R}(\pi)$ is lower semicontinuous. $\square$

### G.8. Proof for Lemma G.2

*Proof.* First, we obtain the smoothness of $\hat{\mathcal{L}}_\alpha$. Observe that $D_{\hat{\mathbb{P}}_X}$ and $D_{\hat{\mathbb{P}}_Y}$ are symmetric. Hence, the gradient of $\nabla \hat{\mathcal{L}}_\alpha(\pi)$ can be calculated as

$$(\alpha n_X n_Y)^{-1} \nabla \hat{\mathcal{L}}_\alpha(\pi) = D_{\hat{\mathbb{P}}_X}^2 \pi + \pi D_{\hat{\mathbb{P}}_Y}^2 - 2 D_{\hat{\mathbb{P}}_X} \pi D_{\hat{\mathbb{P}}_Y} + \text{const.},$$

where the constant does not depend on $\pi$. Thus, the triangle inequality and the Cauchy–Schwarz inequality yield that

$$(\alpha n_X n_Y)^{-1} \left\| \nabla \hat{\mathcal{L}}_\alpha(\pi_1) - \nabla \hat{\mathcal{L}}_\alpha(\pi_2) \right\|_F \leq \left( \|D_{\hat{\mathbb{P}}_X}\|_{\text{op}} + \|D_{\hat{\mathbb{P}}_Y}\|_{\text{op}} \right)^2 \|\pi_1 - \pi_2\|_F.$$

Thus, we get

$$\left\| \nabla \hat{\mathcal{L}}_\alpha(\pi_1) - \nabla \hat{\mathcal{L}}_\alpha(\pi_2) \right\|_F \leq \alpha \cdot n_{\min} n_{\max} \left( \|D_{\hat{\mathbb{P}}_X}\|_{\text{op}} + \|D_{\hat{\mathbb{P}}_Y}\|_{\text{op}} \right)^2 \|\pi_1 - \pi_2\|_F, \tag{16}$$

where $n_{\min} = \min\{n_X, n_Y\}$ and $n_{\max} = \max\{n_X, n_Y\}$.

What remains is to show the convergence of the FW algorithm. Define

$$\beta := \alpha \cdot n_{\min} n_{\max} \left( \|D_{\hat{\mathbb{P}}_X}\|_{\text{op}} + \|D_{\hat{\mathbb{P}}_Y}\|_{\text{op}} \right)^2.$$

Let $\pi_n^* \in \arg\min_\pi \hat{\mathcal{L}}_\alpha(\pi)$, whose existence is guaranteed by the convexity. Then,

$$
\begin{aligned}
\hat{\mathcal{L}}_\alpha(\pi^{(t+1)}) - \hat{\mathcal{L}}_\alpha(\pi^{(t)}) &\leq \text{Tr}\left( \nabla \hat{\mathcal{L}}_\alpha(\pi^{(t)})^\top (\pi^{(t+1)} - \pi^{(t)}) \right) + \frac{\beta}{2} \|\pi^{(t+1)} - \pi^{(t)}\|_F^2 \\
&= \gamma_t \text{Tr}\left( \nabla \hat{\mathcal{L}}_\alpha(\pi^{(t)})^\top (\mu^{(t)} - \pi^{(t)}) \right) + \frac{\beta}{2} \|\pi^{(t+1)} - \pi^{(t)}\|_F^2 \\
&\leq \gamma_t \text{Tr}\left( \nabla \hat{\mathcal{L}}_\alpha(\pi^{(t)})^\top (\pi_n^* - \pi^{(t)}) \right) + \frac{\beta}{2} \|\pi^{(t+1)} - \pi^{(t)}\|_F^2 \\
&\leq \gamma_t \left( \hat{\mathcal{L}}_\alpha(\pi_n^*) - \hat{\mathcal{L}}_\alpha(\pi^{(t)}) \right) + \frac{\beta}{2} \|\pi^{(t+1)} - \pi^{(t)}\|_F^2.
\end{aligned}
$$

The first inequality comes from (16); the third inequality is due to the definition of $\mu^{(t)}$; and the last inequality is from the convexity of $\hat{\mathcal{L}}_\alpha$.

Considering that

$$\|\pi\|_F^2 = \sum_{ij} \pi_{ij}^2 \leq \left( \max_{i,j} \pi_{ij} \right) \sum_{ij} \pi_{ij} = \max_{i,j} \pi_{ij} \leq \frac{1}{n_{\max}},$$

we obtain

$$\|\pi^{(t+1)} - \pi^{(t)}\|_F = \gamma_t \|\mu^{(t)} - \pi^{(t)}\|_F \leq 2\gamma_t \sqrt{\frac{1}{n_{\max}}}.$$

Then, substituting $\gamma_t = 2/(t+2)$, it follows that

$$
\begin{aligned}
\hat{\mathcal{L}}_\alpha(\pi^{(t+1)}) - \hat{\mathcal{L}}_\alpha(\pi_n^*) &\leq (1 - \gamma_t) \left( \hat{\mathcal{L}}_\alpha(\pi^{(t)}) - \hat{\mathcal{L}}_\alpha(\pi_n^*) \right) + 2\beta \gamma_t^2 \cdot \frac{1}{n_{\max}} \\
&= \frac{t}{t+2} \left( \hat{\mathcal{L}}_\alpha(\pi^{(t)}) - \hat{\mathcal{L}}_\alpha(\pi_n^*) \right) + \frac{8\beta}{(t+2)^2} \cdot \frac{1}{n_{\max}}.
\end{aligned}
$$

By this inequality, for $t = 0$ (where $\gamma_0 = 1$), we have

$$\hat{\mathcal{L}}_\alpha(\pi^{(1)}) - \hat{\mathcal{L}}_\alpha(\pi_n^*) \leq 2\beta \cdot \frac{1}{n_{\max}} \leq \frac{8\beta}{2} \cdot \frac{1}{n_{\max}}.$$

Therefore, using mathematical induction, we arrive at the convergence rate. Assuming $\hat{\mathcal{L}}_\alpha(\pi^{(t)}) - \hat{\mathcal{L}}_\alpha(\pi_n^*) \le 8\beta/((t+2)n_{\max})$, we obtain

$$
\begin{aligned}
\hat{\mathcal{L}}_\alpha(\pi^{(t+1)}) - \hat{\mathcal{L}}_\alpha(\pi_n^*) &\le \left( \frac{t}{t+2} \cdot \frac{8\beta}{t+2} + \frac{8\beta}{(t+2)^2} \right) \cdot \frac{1}{n_{\max}} \\
&= \frac{8\beta(t+1)}{(t+2)^2} \cdot \frac{1}{n_{\max}} \\
&\le \frac{8\beta}{t+3} \cdot \frac{1}{n_{\max}},
\end{aligned}
$$

where the last inequality holds since $(t+1)(t+3) = t^2 + 4t + 3 < t^2 + 4t + 4 = (t+2)^2$. $\hspace{1cm}\square$

# H. Experimental Details and Additional Results

This section provides supplementary details regarding the experimental setup and implementation specifics, the discussion of which is postponed in the main text. We also present the formal mathematical definitions of the baseline algorithms and evaluation metrics. Finally, we report comprehensive experimental results on both the synthetic 2D point clouds and the OASIS-3 dataset, including a full comparison of Entropic FGW (EFGW) across varying regularization parameters.

### H.1. Comparison Algorithms

EFGW introduces an entropic regularization term to the original FGW objective, promoting smoother transport plans and enabling the use of Sinkhorn-based algorithms. Formally, for a regularization parameter $\varepsilon > 0$, the EFGW objective is defined by:

$$
\mathcal{L}^\varepsilon_{\text{EFGW},\alpha}(\pi) := (1-\alpha)\mathbb{E}_\pi \left[ \|f_{\mathcal{X}}(X) - f_{\mathcal{Y}}(Y)\|_2^2 \right] + \alpha\mathbb{E}_{\pi\otimes\pi} \left[ |d_{\mathcal{X}}(X,X') - d_{\mathcal{Y}}(Y,Y')|^2 \right] + \varepsilon \cdot D_{\text{KL}}(\pi \mid \mathbb{P}_X \otimes \mathbb{P}_Y),
$$

where $D_{\text{KL}}$ denotes the Kullback–Leibler (KL) divergence.

Let $X_1, ..., X_{n_X}$ be distinct points in $\mathcal{X}$, and $Y_1, ..., Y_{n_Y}$ be distinct points in $\mathcal{Y}$. For each sample, we observe features $f_{\mathcal{X}}(X_i)$ and $f_{\mathcal{Y}}(Y_j)$. In addition, define the empirical probability measures as in the main text. Then, the empirical version of the EFGW objective corresponds to

$$
\hat{\mathcal{L}}^\varepsilon_{\text{EFGW},\alpha}(\pi) := (1-\alpha)\text{Tr}\left( C_f^\top \pi \right) + \alpha \sum_{i,k=1}^{n_X} \sum_{j,l=1}^{n_Y} \left| [D^X]_{ik} - [D^Y]_{jl} \right|^2 \pi_{ij}\pi_{kl} + \varepsilon \sum_{i=1}^{n_X} \sum_{j=1}^{n_Y} \pi_{ij} \log\left( \frac{\pi_{ij}}{(1/n_X)(1/n_Y)} \right),
$$

where $\pi \in \Pi(\hat{\mathbb{P}}_X, \hat{\mathbb{P}}_Y)$ is a $n_X \times n_Y$ empirical coupling matrix, $[C_f]_{ij} = \|f_{\mathcal{X}}(X_i) - f_{\mathcal{Y}}(Y_j)\|_2^2$, $[D^X]_{ik} = d_{\mathcal{X}}(X_i, X_k)$, and $[D^Y]_{jl} = d_{\mathcal{Y}}(Y_j, Y_l)$.

COPT (Dong & Sawin, 2020) is a graph matching method that produces a vertex coupling by coordinating vertex alignment with graph signal alignment. Let $\hat{\mathbb{P}}_X = \sum_{i=1}^{n_X} \delta_{X_i}/n_X$ and $\hat{\mathbb{P}}_Y = \sum_{j=1}^{n_Y} \delta_{Y_j}/n_Y$ be the uniform vertex measures. COPT further equips each graph with Gaussian graph-signal measures $\mu_X$ on $\mathbb{R}^{n_X}$ and $\mu_Y$ on $\mathbb{R}^{n_Y}$ constructed from Laplacian pseudoinverses. For a coupling $\pi \in \Pi(\hat{\mathbb{P}}_X, \hat{\mathbb{P}}_Y)$, define the COPT objective as

$$
\hat{\mathcal{L}}_{\text{COPT}}(\pi) := \inf_{T: T_\# \mu_X = \mu_Y} \int_{g \in \mathbb{R}^{n_X}} \sum_{i=1}^{n_X} \sum_{j=1}^{n_Y} \left( g_i - (Tg)_j \right)^2 \pi_{ij}\, \mu_X(dg),
$$

where $g \in \mathbb{R}^{n_X}$ is a random graph signal on the source graph (with coordinates $g_i$ indexed by $X_i$). The COPT problem is then

$$
\min_{\pi \in \Pi(\hat{\mathbb{P}}_X, \hat{\mathbb{P}}_Y)} \hat{\mathcal{L}}_{\text{COPT}}(\pi).
$$

The inner infimum selects a signal transport map $T$ that aligns the graph-induced signal distributions, while the outer minimization learns a vertex coupling $\pi$ consistent with this global structural alignment. Unlike CDOT or EFGW that incorporate node features through an explicit feature cost matrix $C_f$, COPT does not use node features in our setting and relies solely on structural information induced by the input graphs. We use the resulting minimizer $\pi$ as the matching output.

Spectral Matching (Spectral) (Leordeanu & Hebert, 2005) aligns graph structures by comparing the spectral embeddings of their similarity or distance matrices. Let $U_X$ and $U_Y$ be the matrices containing the leading $k$ eigenvectors of the source and target distance matrices, respectively. Spectral seeks a correspondence that minimizes the distance between these spectral embeddings:

$$\min_{P \in \mathcal{P}} \|U_X - P U_Y\|_F^2,$$

where $\mathcal{P}$ is the set of permutation matrices. This method relies solely on structural information and is non-iterative, making it computationally efficient but potentially less robust to geometric ambiguities compared to OT-based approaches.

IsoRank (Singh et al., 2008) is a global network alignment algorithm based on the intuition that two nodes are similar if their neighbors are similar. It is formulated as an eigenvalue problem similar to PageRank. Formally, let $A_1$ and $A_2$ be the adjacency matrices of the two graphs (normalized to be column-stochastic). The similarity matrix $R$ is computed iteratively via:

$$R = \alpha A_1 R A_2^\top + (1 - \alpha) H,$$

where $H$ is a prior similarity matrix (often uniform or based on node features) and $\alpha$ controls the fusion weight. (Here, $\alpha$ is the IsoRank damping parameter and is distinct from the OT fusion weight used in CDOT/FGW/EFGW.)

## H.2. Synthetic Data Experiments

We utilize the same synthetic dataset described in Section 6.1, consisting of 2D point clouds with four distinct clusters. In this subsection, we provide implementation details and report comprehensive results on the synthetic 2D point clouds, including a full comparison of Entropic FGW (EFGW) across varying regularization parameters.

### H.2.1. IMPLEMENTATION DETAILS

The source and target distributions are sampled from isomorphic mm spaces defined on $[0, 2]^2$, and the feature cost $C_f$ is given by the 0-1 penalty based on ground-truth cluster labels. We compare CDOT against FGW, EFGW, IsoRank, and Spectral Matching (Spectral).

- **FGW & EFGW:** We use the `POT` library (Flamary et al., 2021). For FGW, we adopt the `square_loss` (GW-2) and the default conditional gradient solver (`ot.gromov.fused_gromov_wasserstein`). For EFGW, we use the Sinkhorn-based solver (`ot.gromov.entropic_fused_gromov_wasserstein`) and vary the regularization parameter $\varepsilon \in \{10^{-4}, ..., 10^{-1}\}$.

- **IsoRank:** We construct the similarity matrices $A_{\mathcal{X}}$ and $A_{\mathcal{Y}}$ from the distance matrices using a Gaussian kernel with $\sigma = 1$, such that $[A]_{ij} = \exp(-d_{ij}^2/2)$. These matrices are row-normalized to be stochastic. The prior affinity matrix $H$ is derived from the feature cost matrix $C_f$ by setting $H_{ij} = 1 - [C_f]_{ij}$, followed by normalization. The final alignment is obtained by applying the Hungarian algorithm (LAP) to the converged similarity matrix $R$.

- **Spectral:** We implement Spectral by performing eigen-decomposition on the normalized distance matrices $D^X$ and $D^Y$. Instead of selecting top $k$ eigenvectors, we use the full eigenmatrices. To mitigate the sign ambiguity of eigenvectors (i.e., $v$ and $-v$ represent the same eigenspace), we compute the cost matrix using the squared Euclidean distance between the *absolute values* of the eigenvectors. The final permutation is obtained by solving the LAP on this cost matrix.

Consistent with the main experiment, we report the mean squared error (MSE) between the source $X$ and the transported target $\hat{Y} = N\hat{\pi}Y$ over 100 independent trials. All distance matrices are normalized by their maximum values.

### H.2.2. RESULTS

Table 7 presents the comparative results between CDOT and EFGW across varying regularization parameters. When $\alpha = 0.0$, the problem reduces to matching points based solely on cluster labels. Since points within the same cluster are indistinguishable, CDOT (pure OT) yields a constant MSE of approximately 0.33. In contrast, EFGW consistently achieves a lower MSE ($\approx 0.16$) regardless of the regularization parameter. This is due to the entropic smoothing effect: rather than

*Table 7.* Comparison of MSE (mean $\pm$ std) over 100 independent trials. CDOT is compared against Entropic FGW (EFGW) with varying regularization parameters. The best MSE performance for each configuration is highlighted in bold.

| $\alpha$ | $n$ | **CDOT** MSE | **EFGW** $(10^{-4})$ MSE | **EFGW** $(10^{-3})$ MSE | **EFGW** $(10^{-2})$ MSE | **EFGW** $(10^{-1})$ MSE |
|---|---|---|---|---|---|---|
| | 100 | $0.3327 \pm 0.01$ | $\mathbf{0.1686} \pm 0.01$ | $\mathbf{0.1686} \pm 0.01$ | $\mathbf{0.1686} \pm 0.01$ | $\mathbf{0.1686} \pm 0.01$ |
| | 200 | $0.3355 \pm 0.01$ | $\mathbf{0.1677} \pm 0.00$ | $\mathbf{0.1677} \pm 0.00$ | $\mathbf{0.1677} \pm 0.00$ | $\mathbf{0.1677} \pm 0.00$ |
| 0.0 | 300 | $0.3326 \pm 0.01$ | $\mathbf{0.1676} \pm 0.00$ | $\mathbf{0.1676} \pm 0.00$ | $\mathbf{0.1676} \pm 0.00$ | $\mathbf{0.1676} \pm 0.00$ |
| | 400 | $0.3345 \pm 0.01$ | $\mathbf{0.1674} \pm 0.00$ | $\mathbf{0.1674} \pm 0.00$ | $\mathbf{0.1674} \pm 0.00$ | $\mathbf{0.1674} \pm 0.00$ |
| | 500 | $0.3329 \pm 0.01$ | $\mathbf{0.1671} \pm 0.00$ | $\mathbf{0.1671} \pm 0.00$ | $\mathbf{0.1671} \pm 0.00$ | $\mathbf{0.1671} \pm 0.00$ |
| | 100 | $\mathbf{0.0077} \pm 0.00$ | $2.6562 \pm 0.05$ | $0.0098 \pm 0.00$ | $0.0347 \pm 0.00$ | $0.1438 \pm 0.01$ |
| | 200 | $\mathbf{0.0040} \pm 0.00$ | $2.6684 \pm 0.03$ | $0.0054 \pm 0.00$ | $0.0334 \pm 0.00$ | $0.1436 \pm 0.00$ |
| 0.5 | 300 | $\mathbf{0.0027} \pm 0.00$ | $2.6651 \pm 0.03$ | $0.0038 \pm 0.00$ | $0.0333 \pm 0.00$ | $0.1437 \pm 0.00$ |
| | 400 | $\mathbf{0.0020} \pm 0.00$ | $2.6670 \pm 0.02$ | $0.0031 \pm 0.00$ | $0.0332 \pm 0.00$ | $0.1436 \pm 0.00$ |
| | 500 | $\mathbf{0.0016} \pm 0.00$ | $2.6666 \pm 0.02$ | $2.6666 \pm 0.02$ | $0.0330 \pm 0.00$ | $0.1434 \pm 0.00$ |
| | 100 | $0.6743 \pm 0.04$ | $2.6562 \pm 0.05$ | $1.3286 \pm 0.65$ | $1.2439 \pm 0.57$ | $\mathbf{0.6665} \pm 0.02$ |
| | 200 | $0.6712 \pm 0.02$ | $2.6684 \pm 0.03$ | $1.4212 \pm 0.60$ | $1.3157 \pm 0.60$ | $\mathbf{0.6691} \pm 0.01$ |
| 1.0 | 300 | $0.6682 \pm 0.02$ | $2.6651 \pm 0.03$ | $1.3781 \pm 0.65$ | $1.2735 \pm 0.56$ | $\mathbf{0.6671} \pm 0.01$ |
| | 400 | $0.6688 \pm 0.01$ | $2.6670 \pm 0.02$ | $1.1405 \pm 0.63$ | $1.1384 \pm 0.62$ | $\mathbf{0.6682} \pm 0.01$ |
| | 500 | $0.6673 \pm 0.01$ | $2.6666 \pm 0.02$ | $1.4051 \pm 0.70$ | $1.2483 \pm 0.66$ | $\mathbf{0.6670} \pm 0.01$ |

selecting a single random point, EFGW distributes mass across all valid targets within the cluster. This maps each source point to the centroid (barycenter) of the corresponding target cluster, thereby minimizing the expected squared error.

In the fused setting where both feature and geometric constraints are active, CDOT consistently achieves the lowest MSE ($\approx 0.002$ for $n = 400$), demonstrating the effectiveness of its convex formulation. EFGW exhibits significant sensitivity to the regularization parameter $\varepsilon$. At a low regularization level ($\varepsilon = 10^{-4}$), EFGW yields a high MSE of 2.66, suggesting numerical instability or entrapment in local stationary points, which is typical for non-convex solvers in complex landscapes. While increasing $\varepsilon$ to $10^{-3}$ significantly improves performance, it still does not match the precision of CDOT. Further increasing $\varepsilon$ leads to over-smoothing, degrading the MSE again. This highlights the difficulty of tuning $\varepsilon$ in EFGW, whereas CDOT provides robust performance without such hyperparameters.

In the pure geometric setting, the symmetry of the four identical clusters renders the true alignment unidentifiable. Under this regime, CDOT maintains a stable solution with an MSE of approximately 0.67 and a remarkably low standard deviation. Interestingly, EFGW with high regularization ($\varepsilon = 10^{-1}$) yields nearly identical performance. This is because both methods essentially converge toward the independent coupling due to the non-identifiability. However, EFGW with lower $\varepsilon$ shows high variance and instability, often failing to find a consistent solution.

### H.3. Real Data Experiments on the OASIS-3 Dataset

In this subsection, we provide supplementary details regarding the experimental setup, mathematical definitions of the metrics, and comprehensive results on the OASIS-3 dataset that are omitted from the main text.

#### H.3.1. GEODESIC AND DIFFUSION DISTANCES

The geodesic distance between two nodes on a graph $G = (V, E)$ is defined as the length of the shortest path connecting them. In the context of weighted graphs (connectomes), edge weights typically represent connection strength (number of fibers). We convert these weights into costs by taking the reciprocal ($1/w_{ij}$) before computing the shortest paths using Dijkstra's algorithm. Since brain networks may contain disconnected components, the geodesic distance between disconnected nodes is mathematically infinite. To handle this numerically, we replace infinite values with a finite maximum

*Table 8.* Full comparison of matching accuracy on the OASIS-3 dataset (mean $\pm$ std). For EFGW, results across varying regularization parameters ($\varepsilon$) are provided. The best value for each metric is highlighted in bold.

| Method | Metric | | |
| --- | --- | --- | --- |
| | Diffusion | Geodesic | Topology |
| **CDOT (Ours)** | **0.6136** $\pm$ 0.11 | 0.4640 $\pm$ 0.14 | – |
| FGW | 0.1853 $\pm$ 0.03 | **0.5375** $\pm$ 0.16 | – |
| EFGW ($\varepsilon = 10^{-4}$) | 0.4097 $\pm$ 0.14 | 0.0000 $\pm$ 0.00 | – |
| EFGW ($\varepsilon = 10^{-3}$) | 0.3342 $\pm$ 0.06 | 0.4583 $\pm$ 0.12 | – |
| EFGW ($\varepsilon = 10^{-2}$) | 0.2889 $\pm$ 0.05 | 0.4178 $\pm$ 0.10 | – |
| EFGW ($\varepsilon = 10^{-1}$) | 0.2567 $\pm$ 0.05 | 0.3988 $\pm$ 0.09 | – |
| Spectral | – | – | 0.0737 $\pm$ 0.03 |
| IsoRank | – | – | **0.4055** $\pm$ 0.09 |
| COPT | – | – | 0.0253 $\pm$ 0.01 |

distance, defined as:

$$d_{ij} = \begin{cases} \text{shortest\_path}(i,j) & \text{if } i \text{ and } j \text{ are connected,} \\ \max_{(u,v) \in E} d_{uv} & \text{otherwise.} \end{cases}$$

Diffusion distance captures the connectivity between nodes based on the probability of transition in a random walk, rather than a single shortest path. It is defined using the heat kernel $H_t = \exp(-t\mathcal{L})$, where $\mathcal{L}$ is the graph Laplacian and $t$ is the diffusion scale parameter. The diffusion distance between nodes $i$ and $j$ at scale $t$ is the Euclidean distance between their corresponding rows in the heat kernel:

$$D_t(i,j) = \|H_t(i,\cdot) - H_t(j,\cdot)\|_2.$$

Unlike geodesic distance, diffusion distance integrates information from all possible paths of length proportional to $t$, resulting in a smoother metric that is robust to noise and topological perturbations. It evaluates whether two nodes share similar functional roles (i.e., similar diffusion profiles) within the global network structure. This metric is widely adopted in connectome analysis to capture global information flow and functional geometry (Coifman & Lafon, 2006; Abdelnour et al., 2014). In our experiments, we set the scale $t = 1$.

### H.3.2. IMPLEMENTATION DETAILS

For node features, we utilize 6 categorical labels derived from the OASIS-3 metadata: hemisphere (left, right, central) and cortical status (cortical, subcortical). These are compared using 0-1 penalty. For the optimization, we set the fusion weight $\alpha = 0.5$ for CDOT, FGW, EFGW, and IsoRank to ensure a balanced contribution. We use the `square_loss` for the GW term. All matching algorithms are initialized with an independent coupling (product of marginals) and run for $T = 200$ iterations. We compare CDOT against FGW, EFGW, Spectral, IsoRank, and COPT. For FGW and EFGW, we utilize the `POT` library. For EFGW, we perform a grid search for the regularization parameter $\varepsilon \in \{10^{-4}, ..., 10^{-1}\}$. SM is implemented by eigen-decomposing the graph Laplacian $L = D - A$ and computing node embeddings from the full set of eigenvectors. To mitigate the sign ambiguity of eigenvectors, we build the cost matrix using squared Euclidean distances between the elementwise absolute values of the eigenvectors. For IsoRank, we incorporate feature information by setting the prior similarity matrix $H = \mathbf{1} - C_f$, where $C_f$ is the feature cost matrix. COPT is implemented by constructing the combinatorial Laplacian $L = D - A$, computing its Moore–Penrose pseudoinverse $L^\dagger$ via eigen-decomposition with threshold $\varepsilon_{\text{pinv}} = 10^{-10}$, and optimizing a nonnegative coupling with projected gradient descent combined with Sinkhorn–Knopp marginal scaling.

### H.3.3. RESULTS

Table 8 presents the detailed matching accuracy (mean $\pm$ standard deviation) across all subject pairs. The results highlight a clear trade-off depending on the geometric representation. FGW performs best under geodesic distance because the shortest-path metric creates a rigid, isometric structure that aligns well with FGW's quadratic objective.

The stark performance gap under diffusion distance (CDOT: 0.6136 vs. FGW: 0.1853) provides critical insight into the behavior of the proposed loss function. Diffusion distance, by definition, averages over all possible paths to capture global information flow. While this yields a robust metric insensitive to topological noise, it essentially acts as a low-pass filter, smoothing out local variations and reducing the variance of pairwise distances. For FGW, this smoothing is detrimental. The GW objective minimizes a quadratic discrepancy, $|d_\mathcal{X}(x, x') - d_\mathcal{Y}(y, y')|^2$, which relies heavily on distinct pairwise contrasts (high variance) to lock the alignment. When diffusion smooths the metric, the distance landscape becomes flat, causing the non-convex GW objective to lose discriminative power and get trapped in local stationary points. In contrast, CDOT overcomes this via its *operator-based formulation* and *convexity*. First, the distance operator $D_{\mathbb{P}_X}$ transforms the diffusion metric into a global geometric profile (e.g., mapping a point to its average diffusion distance to the rest of the brain). Even if pairwise contrasts are diminished, these global profiles remain distinct and alignable. CDOT's structural penalty aligns these global functional geometries directly. Second, and most importantly, the CDOT objective is convex (Theorem 3.4). Unlike FGW, which wanders in the flattened landscape, CDOT effectively utilizes the linear feature term to anchor the solution and guarantees convergence to the global optimum even when the geometric signal is smooth.

Regarding topology-based methods, SM shows poor performance (0.0737), implying that purely structural alignment is insufficient. IsoRank, when enhanced with feature information ($H = 1 - C_f$), shows significantly improved performance (0.4055). However, it still falls short of OT-based methods (CDOT and FGW). This suggests that while feature information is critical, IsoRank's reliance on local neighborhood similarity is less effective for brain registration than OT frameworks, which align graphs based on detailed geometric structures (pairwise distance matrices).

