# OpenReview forum: "Convex Distance Operator Transport: A Convex and Geometry-Preserving Formulation"
_ICML.cc/2026/Conference — ICML 2026 regular_

### Official Review · Reviewer_y8Jm · 2026-03-10

**Soundness:** 4
**Presentation:** 4
**Significance:** 4
**Originality:** 4
**Overall Recommendation:** 5
**Confidence:** 5

**Summary:**

Roughly speaking, the main idea underlying this paper is to mimic the known decomposition of the variance of a RV as the sum of its 2nd order moment minus its squared expected value in order to replace the objective function in the 2-Gromov Wasserstein (GW) distance by an alternative quantity having the unquestionable advantage of being a convex functional of the transport plan. This is formally done via the introduction of two operators (Distance and Conditional Expectation) and extended to featured measure metric spaces via the Convex Distance Operator Transport (CDOT). The authors prove that CDOT is a pseudometric, establish relations with FGW, propose two optimisation algorithms (relying on Franck-Wolfe) in order to infer an optimal transport plan in the discrete case, obtain statistical guarantees in the finite sample framework and validate the whole thing with experiments on simulated and real data.

**Compliance With Llm Reviewing Policy:**

Affirmed.

**Final Justification:**

As stated in my review, this work provides a clear and incisive introduction to OT. The authors rebuttal addressed all my questions.

**Key Questions For Authors:**

1. Among the advantages of the proposed work (page 2, line 90) the authors state "...guaranteeing convergence to a global optimum without relying on additional relaxations such as entropic regularization". I find it slightly unfair: entropic regularization (ER) would allow the optimization problem to be *stricly* convex, which is not the case for CDOT. Indeed, for $α=0$ , CDOT reduces to 2-Wasserstein and ER is known to make it strongly convex. A remark to better state it would be appreciated.

2. I would be very curious to know what effects this formulation has in terms of *sparsity* (or density!) of the estimated transport plan. How it compares with FGW? Could it be used to obtained sparser solutions or not?

3. In the same vein of the previous point: are there some scenarios (apart form the one in Table 3) in which FGW should be preferred to CDOT? Or should this paper be read as an invitation for FGW to retire?

4. In the experiment in Section 6.2, starting from line 380 you state "Since all subjects share a common .... average accuracy". These few lines are absolutely not enogh to figure out what you do. Are you seeking to match graphs corresponding to the main individual? Why some brain connections should be matched? What do "true" labels correspond to?

5. This is a positive remark and possibly a suggestion: when discussing the complexity of your lazy gradient FW algorithm you state that computational complexity is reduced to quadratic **except** for the linear minimization problem, i.e. Wasserstein. I find it very interesting because it seems to suggest that is some scenario, using CDOT with $\alpha = 1$ might be more interesting that doing Wasserstein, even with Euclidean data!

**Limitations:**

Se above.

**Strengths And Weaknesses:**

This paper is monumental. The amount of work the authors did is impressive, and although optimal transport between attributed metric spaces (i.e., often, attributed graphs) is a *niche* topic in maching learning, well, this work has the potential to definitely be a change point in the story of that niche. Introducing a convex objective function replacing (F)GW and having such a nice properties (pseudometric)  is the main strength of the paper event apart from its statistical properties and the essentially reduced computational burden. The only reason why I do not assign the highest note (for the moment) is a list of minor remarks stated below.
Here, I just have a single complain. This paper is so intense and reach that it does not fit with the format of ICML. It would deserve a more in depth review that unfortunately is not compatible with having so many papers to review in such a short time.

---

> ### Author Rebuttal · Authors · 2026-03-30
>
> We appreciate the reviewer’s positive assessment of our work, as well as the detailed and thoughtful comments on wording, algorithmic interpretation, and experimental protocol. We address the main points below.
>
> > 1. “...guaranteeing convergence to a global optimum without relying on additional relaxations such as entropic regularization” is slightly unfair.
>
> Thank you for your feedback. Our wording should be more careful here. In particular, when `alpha=0`, CDOT reduces to the 2-Wasserstein problem, and entropic regularization indeed makes that OT objective strictly convex. More generally, entropic regularization can also improve the optimization landscape of GW-type problems. Our intended point is narrower: CDOT is already convex in its original, unregularized formulation, so global optimization does not depend on introducing an additional smoothing parameter. We will revise the sentence so that it does not read as dismissing entropic regularization.
>
> Reference: Rioux, G., Goldfeld, Z., and Kato, K. (2024). “Entropic Gromov--Wasserstein distances: Stability and algorithms.” Journal of Machine Learning Research 25(363):1-52.
>
> > 2. "Can CDOT be used to obtain sparser solutions?"
>
> Thank you for your insightful suggestion. The unregularized CDOT objective does not explicitly promote sparsity of the transport plan. Compared with FGW, it can naturally favor more diffuse couplings because CDOT removes the dispersion-driven part of the GW objective that tends to prefer concentrated, nearly deterministic matchings. For applications where a hard correspondence is required, our current approach is to recover a soft coupling first and then apply a linear assignment post-processing step. In fact, the dispersion-gap perspective suggests a natural way to control this behavior: one can introduce an additional coefficient on the dispersion-related component to interpolate between CDOT and GW. We have not studied this systematically in the present paper, so we prefer not to make a strong empirical claim here, but we agree that sparsity control is an interesting and well-motivated direction for future work.
>
> > 3. "When should FGW be preferred to CDOT?"
>
> FGW is preferable when the task  calls for a nearly one-to-one alignment between points or cells that preserves pairwise geometry, as in shape correspondence or GW/FGW-based single-cell alignment (Vayer et al., 2020; Klein et al., 2025). By contrast, CDOT is more suitable when the geometry is intentionally smoothed or diffusion-based, so that exact preservation of each pairwise distance is less important than capturing a global structural signal. This is closer in spirit to settings such as our diffusion-distance connectome experiment and diffusion-pseudotime analysis in single-cell data (Haghverdi et al., 2016).
>
> References: Vayer, T., Chapel, L., Flamary, R., Tavenard, R., and Courty, N. (2020). Fused Gromov-Wasserstein Distance for Structured Objects. Algorithms 13(9):212. Klein, D., Palla, G., Lange, M. et al. (2025). Mapping Cells Through Time and Space with Moscot. Nature 638:1065-1075. Haghverdi, L., Buettner, M., Wolf, F. A., Buettner, F., and Theis, F. J. (2016). Diffusion Pseudotime Robustly Reconstructs Lineage Branching. Nature Methods 13(10):845-848.
>
> > 4. “Regarding OASIS-3 brain connectome analysis, are you seeking to match graphs corresponding to the main individual? Why some brain connections should be matched? What do ‘true’ labels correspond to?”
>
> In the OASIS-3 experiment, we seek node correspondence across subjects, not edge correspondence. Because all graphs are built from a common atlas, node `i` in one subject and node `i` in another subject represent the same anatomical ROI, so the ground-truth matching is the identity permutation. The connectivity matrices are used only to define the graph geometry, while the six anatomical categories are auxiliary node features. Each method outputs a soft coupling or similarity matrix, which we convert to a hard one-to-one matching via a linear assignment problem; the reported accuracy is the fraction of nodes matched to their correct atlas index. We have revised the main text to make this evaluation protocol explicit.
>
> > 5. “Can CDOT be interpreted as computationally more advantageous than Wasserstein in certain scenarios?”
>
> Thank you for this remark. Our complexity discussion may indeed be read in that way, but this is not our intended claim. The quadratic improvement refers only to the gradient-update part of each Frank--Wolfe iteration. The linear minimization oracle remains a standard OT subproblem, so we do not intend to claim an asymptotic advantage over directly solving Wasserstein when `alpha=0`. The practical advantage of the lazy update appears when `alpha>0`, where the structural term would otherwise require repeated dense gradient recomputation. We will revise the manuscript to make this distinction explicit.

---

> > ### Author Rebuttal · Reviewer_y8Jm · 2026-04-02
> >
> > My concerns are well addressed.

---

### Official Review · Reviewer_gKrD · 2026-03-11

**Soundness:** 3
**Presentation:** 2
**Significance:** 2
**Originality:** 3
**Overall Recommendation:** 3
**Confidence:** 4

**Summary:**

This submission proposes an alternative to the Gromov-Wasserstein (GW) and fused GW distances to enable comparing probability distributions on different metric spaces with features. As opposed to the fused GW distance, the proposed framework, called the convex distance operator transport (CDOT) defines a convex optimization problem. It is further illustrated that the CDOT discrepancy is a pseudometric and it is shown that the structural component of its loss is related to the standard GW objective up to a term accounting for the spatial uncertainty of the coupling. To solve the underlying optimization problem, it is proposed to utilize the Frank-Wolfe algorithm. The final theoretical contribution of this submission regard quantifying the effects of the optimization and statistical errors on estimating the CDOT discrepancy based on empirical measures. An error bound is proved in Theorem 5.6 and consistency of the estimate is derived in Corollary 5.7.

The paper concludes with some synthetic and real life examples. It is shown that the proposed CDOT discrepancy compares well to other competing methods.

**Compliance With Llm Reviewing Policy:**

Affirmed.

**Final Justification:**

The authors' rebuttal has clarified some of the initial concerns I had with the submission and I believe the paper will be improved with the added discussion. As noted below, I believe that the proposed CDOT is interesting and novel, but some of the technical properties of this framework were not sufficiently fleshed out. It is also unclear how the convexity properties of this framework clash with the inherent nonconvexity of the GW-based approaches; these two approaches require a more careful comparison.

**Key Questions For Authors:**

My main questions in regards to this submission are contained in the strengths and weaknesses section above. They are compiled here for ease of reference.

1. Can the conditions under which the CDOT discrepancy is equal to 0 in Theorem 3.5 be more clearly discussed? Ideally, the result should state that $d_{\mathrm{CT}}^{(\alpha)}(\mathfrak X,\mathfrak Y)=0$ if and only if ...
2. Can primitive sufficient conditions for Assumption 5.5 to hold in the metric space setting be provided? Such conditions are reasonably well understood in Euclidean spaces, but to my knowledge are less clear in the metric space setting.
3. Can the the statistical error in Theorem 5.6 be discussed more carefully? It would be helpful to have a better picture of how this error decreases as a function of the number of samples (e.g., can concentration inequalities be provided or the expectation of this quantity be bounded).
4. Can the CDOT discrepancy and GW framework be more carefully contrasted? Going back to the first question, it is known that the GW distance vanishes if and only if the two metric measure spaces are isomorphic and so optimal couplings provide alignments of the two spaces in a very precise sense. Given that the CDOT discrepancy appears to be strictly convex as noted above (this follows by using the strong convexity of the squared Hilbert-Schmidt norm at the end of the proof of Theorem 3.4) there cannot be multiple optimal couplings in the CDOT framework which is quite different from the GW case.

**Limitations:**

yes

**Strengths And Weaknesses:**

The submission appears technically sound; the various results are supported by proofs in the appendix. I believe that the assumptions in Section 5 are too difficult to check in practice however. In effect, while Assumption 5.5 is justified by the fact that it holds when the optimal coupling is induced by a Lipschitz Monge map (see the first paragraph on p.6), I do not believe that this assumption is well-justified in the metric space setting considered herein. Indeed, the cited papers Deb et al., Hutter and Rigollet, and Manole et al. all study Euclidean spaces since, in that setting, the smoothness of Monge maps can be characterized by properties of the marginals. In the abstract metric space setting, it is not so clear if there are primitive conditions guaranteeing that the Monge map is Lipschitz.

The presentation of the paper is good altogether, but I find some of the discussion a bit unclear. For instance, it is stated on p.3 line 125 that "few works have fully resolved the non-convexity". It is my understanding that non-convexity should in fact be an unavoidable property in the context of comparing metric measure spaces modulo isomorphisms, which is the main purpose of the GW framework. I believe that a more clear comparison should be made in this regard. Another point is that Table 1 includes the header Asymp. Theory which is a bit confusing, since the caption states that this also includes non-asymptotic bounds. There are also a few errors in the references. For instance, proper names such as Gromov and Wasserstein should be capitalized, there are some references which include a spurious et al. (e.g. the paper of Frank and Wolfe, and Villani's book).

The contribution is, in my opinion, reasonable, as it provides a means to compare distinct metric measure spaces with features that has desirable optimization properties. The strength of the contribution could be improved by:

1. Characterizing under what conditions the CDOT discrepancy is equal to 0 in Theorem 3.5 i.e., $d_{\mathrm{CT}}^{(\alpha)}(\mathfrak X,\mathfrak Y)=0$ if and only if ...
2. Establishing primitive sufficient conditions for Assumption 5.5 to hold in the metric space setting.
3. Discussing the statistical error in Theorem 5.6 more carefully (e.g., what is the expected magnitude of this error, how does it decay as the number of samples increases, etc).

It also appears to me that the CDOT objective is strictly convex so that, in Theorem 3.4, a unique optimal coupling exists.

The proposed CDOT discrepancy appears to be novel and inspired by the fused GW framework.

---

> ### Author Rebuttal · Authors · 2026-03-30
>
> We appreciate the reviewer’s close reading of the theory and address below the requests for sharper statements about zero discrepancy, convexity, and statistical assumptions.
>
> > 1. “Can the conditions under which the CDOT discrepancy is equal to 0 in Theorem 3.5 be more clearly discussed?"
>
> Thank you for your suggestion. The zero-discrepancy condition in Theorem 3.5 should be stated more explicitly. In the revision, we will write it in an “if and only if” form: for $\\alpha\\in(0,1)$,
> $$
> d_{\\mathrm{CT}}^{(\\alpha)}(\\mathfrak X,\\mathfrak Y)=0
> $$
> if and only if there exists a coupling $\\pi\\in\\Pi(P_X,P_Y)$ such that the feature term vanishes $\\pi$-a.e. and
> $$
> D_{P_X}T_{\\pi}=T_{\\pi}D_{P_Y}.
> $$
> Thus, zero discrepancy is equivalent to the existence of a coupling preserving both the feature map and the operator-level geometry. Any feature-preserving measure-preserving isomorphism is therefore a sufficient condition for zero discrepancy, but the converse does not hold in general. In the discrete (graph) setting, this concept is closely related to fractional isomorphism, which is slightly weaker than strict isomorphism. We will revise Theorem 3.5 and the surrounding discussion accordingly.
>
> > 2. "Is CDOT strictly convex?"
>
> Theorem 3.4 establishes convexity of the CDOT objective, but not strict convexity; in general, multiple optimal couplings may exist. This is consistent with the pseudometric nature of the discrepancy: for example, if two spaces are isomorphic, all measure-preserving isometries are optimal solutions. We will clarify this point in the discussion of Theorem 3.4 in the revised version.
>
> > 3. “Non-convexity should in fact be an unavoidable property in the context of comparing metric measure spaces modulo isomorphisms …”
>
> This is a very interesting perspective, and we largely agree. It also helps clarify the contrast between CDOT and GW. If the goal is to distinguish spaces up to a rigid notion of isomorphism, then some form of non-convexity is indeed difficult to avoid. Our dispersion-gap result clarifies the difference: unlike CDOT, the GW objective contains the dispersion term that favors deterministic couplings, which makes GW more sensitive to exact pairwise structure but also gives rise to the non-convex optimization landscape. CDOT instead compares aggregated geometric profiles through the operator formulation. This relaxed notion of equivalence is what makes convex optimization attainable, and we will revise the paper to emphasize this trade-off more clearly.
>
> > 4. "Can more justification for Assumption 5.5 be provided?"
>
> Thank you for your insightful feedback. We acknowledge that Assumption 5.5 is difficult to verify in full generality and that our original discussion relied too heavily on Euclidean OT-map estimation papers. In the revised paper, we will present Assumption 5.5 as a kernel-regularity condition under which our theorem applies, and we will add a more concrete sufficient condition in the discrete setting. Specifically, if the optimal coupling has conditional laws that are Lipschitz in total variation in both variables, then Assumption 5.5 follows from the standard bound $W_1\\leq diam\\,\\|\\cdot\\|_{TV}$. In the discrete setting, this becomes directly checkable from the rows and columns of the optimal coupling matrix. We will retain the bi-Lipschitz Monge-map case only as a simple geometric example, rather than as the main justification.
>
> > 5. "Can the statistical error in Theorem 5.6 be discussed more carefully?"
>
> The statistical term in Theorem 5.6 can be made more explicit. Under i.i.d. sampling, it can be specialized using standard Wasserstein convergence rates for empirical measures. In particular, the statistical error term is $W_1(P_X,\\hat P_X)+W_1(P_Y,\\hat P_Y)$, whose expectation is of order $n^{-1/2}$ in dimension 1, $n^{-1/2}\\log(1+n)$ in dimension 2, and $n^{-1/d}$ in dimension $d\\geq 3$. We will add this interpretation in the revision.
>
> References:
> Weed, J. and Bach, F. (2019). “Sharp asymptotic and finite-sample rates of convergence of empirical measures in Wasserstein distance.” Bernoulli 25(4A):2620-2648.
> Fournier, N. and Guillin, A. (2015). “On the rate of convergence in Wasserstein distance of the empirical measure.” Probability Theory and Related Fields 162(3-4):707-738.
>
> > 6. "Terminologies in Table 1 should be clearer".
>
> We agree that the original header “Asymp. Theory” was too vague. In the revised paper, we will replace it by “Consistency” and “Finite-sample.” These refer to the asymptotic consistency of the population risk attained by the lifted empirical estimator and the deterministic finite-sample excess-risk bound in Theorem 5.6. The legend now states that conditional GW/FGW entries concern empirical-risk-to-population-risk guarantees, not explicit excess-risk guarantees for the returned plan.
>
> > 7. "There are some errors in the references."
>
> We will correct these reference issues and recheck the references.

---

> > ### Author Rebuttal · Reviewer_gKrD · 2026-04-01
> >
> > I thank the authors for their response. They have partially addressed some of my questions, but I believe the paper is still not transparent enough regarding its comparison to the GW distances. Indeed the current framing of the paper does not carefully discuss what the fundamental differences between the proposed object and GW differences are. As noted in the review, nonconvexity is inherent to these problems, as they generalize quadratic assignment problems.
> >
> > Furthermore, it seems strange that the current formulation includes all plans generated by measuring preserving isometric as solutions, as then any convex combination of solutions is also a solution and this may not correspond to any sensible alignment of the spaces.
> >
> > Finally, Theorem 5.6 is presented in arbitrary metric spaces whereas the Fournier and Guillin paper treats only Euclidean spaces. My concern is mainly with the case where the metric spaces are infinite dimensional. Can anything about the bound be said in that case?

---

> > > ### Author Response · Authors · 2026-04-08
> > >
> > > Thank you for the follow-up comment. We agree that the distinction between CDOT and GW should be stated more transparently.
> > >
> > > Our intention is not to present CDOT as a convex reformulation of the same alignment problem as GW. Rather, the two objects capture different notions of geometric comparison. GW is built on direct pairwise distance matching and is therefore tailored to rigid measure-preserving isometric alignment; this is also why non-convexity is intrinsic. By contrast, CDOT compares spaces through operator-level aggregated geometry, via the interaction between distance operators and conditional expectation operators induced by a coupling.
> > >
> > > We also agree that if several measure-preserving isometric plans achieve zero cost, then their convex combinations need not correspond to a meaningful one-to-one alignment in the GW sense. We do not view this as a defect of the formulation, but rather as reflecting that the zero-discrepancy class of CDOT is inherently weaker than rigid isomorphism. In this sense, CDOT is closer to a relaxed or fractional notion of equivalence. This perspective is also aligned with the literature on fractional isomorphism: in the graphon setting, relaxed equivalence can be characterized through operator objects such as Markov operators, which is conceptually close to our operator-based viewpoint. Accordingly, CDOT minimizers should be interpreted as soft correspondences rather than unique hard matchings. This is consistent with our estimator $\\hat{\\pi}$, which is generally a soft transport plan; when a deterministic one-to-one matching is needed, we apply the LAP projection in Section 4. We will revise the discussion around Theorem 3.5 to make clear that CDOT captures an operator-level / fractional notion of equivalence rather than strict GW-style alignment.
> > >
> > > Regarding Theorem 5.6, we agree that our previous discussion should distinguish more carefully between the general deterministic statement and Euclidean rate specializations. The theorem itself remains valid on arbitrary compact metric-measure spaces, including infinite-dimensional ones, because it only controls the excess risk through empirical Wasserstein terms. What requires additional assumptions is any explicit sample-size rate for those terms. The Fournier–Guillin and Weed–Bach results were intended only as Euclidean examples, not as general guarantees for arbitrary metric spaces. In particular, in infinite-dimensional settings one should not expect a dimension-free polynomial rate without further structural assumptions. We will revise the discussion to make this limitation precise.
> > >
> > > Thank you again for highlighting these points. We believe these clarifications will substantially improve the transparency of the paper.
> > >
> > > Reference:
> > > Grebík, J., & Rocha, I. (2022). Fractional isomorphism of graphons. Combinatorica, 42(3), 365–404.

---

### Official Review · Reviewer_LL17 · 2026-03-12

**Soundness:** 4
**Presentation:** 4
**Significance:** 4
**Originality:** 4
**Overall Recommendation:** 5
**Confidence:** 5

**Summary:**

The paper introduces Convex Distance Operator Transport (CDOT), a new OT-based discrepancy that compares two attributed compact metric-measure spaces by how a coupling aligns their distance operators, rather than by directly matching all pairwise distances as in GW/FGW. More precisely, the method matches geometry not by directly comparing pairwise distances edge-by-edge, but by comparing aggregated distance profiles, i.e., conditional expectations of distances induced by the coupling. This shift yields a convex optimization problem, which is the paper’s main technical advantage. The authors show that CDOT defines a pseudometric and establish existence and consistency results. Most importantly, the authors relate the method to quadratic GW through a decomposition that highlights an additional dispersion gap term responsible for GW’s nonconvexity. I think this dispersion gap is an important result, as it explains what CDOT is actually changing relative to (F)GW. Empirically, the method performs competitively on synthetic matching, brain alignment, and graph classification tasks, suggesting that this operator-based relaxation can preserve useful geometric structure while being significantly easier to optimize.

**Compliance With Llm Reviewing Policy:**

Affirmed.

**Final Justification:**

This is a strong paper, and the authors’ rebuttal further reinforces my positive assessment. I therefore maintain my very favorable view of this work.

**Key Questions For Authors:**

Please refer to the weaknesses for details. To summarize my main concerns into two concrete questions:

Q1. Could you please provide empirical convergence plots for the proposed CDOT solver, comparing lazy Frank–Wolfe and standard Frank–Wolfe across different attributed compact metric-measure spaces?

Q2. Could you please provide wall-clock runtime comparisons on these problems, including comparisons with the relevant baselines?

**Limitations:**

Unfortunately, limitations and impact were not discussed.

**Strengths And Weaknesses:**

**Strengths:**

* Gromov-Wasserstein (GW) and Fused GW (FGW) have attracted substantial interest across a wide range of fields, from computer graphics to computational biology. However, their optimization is both computationally demanding and inherently non-convex, which limits scalability and practical applicability. This paper proposes a principled relaxation of GW that captures local geometry through aggregated distance profiles—formalized as conditional expectations of distances under the coupling—rather than through direct pairwise distance matching. A key strength of the work is that this reformulation yields a convex objective, making the problem significantly more amenable to scalable optimization with well-established convex methods.

* The authors further decompose the GW objective into two terms: a convex structural term, given by the proposed CDOT, and a dispersion term. This decomposition is particularly insightful, as it clarifies the relationship between CDOT and GW, identifies the source of GW’s non-convexity, and provides a clear conceptual explanation for why the proposed relaxation is easier to optimize.

* The authors provide strong theoretical support for the proposed framework, including: 1) showing that CDOT defines a pseudometric on the space of attributed compact metric-measure spaces, 2) proving the existence of an optimal transport plan and the convexity of the objective, and 3) establishing finite-sample guarantees through non-asymptotic risk bounds and consistency under a globally convergent Frank–Wolfe algorithm.

* Finally, the numerical experiments lend credible support to the theory by showing that CDOT performs competitively across synthetic matching, brain connectome alignment, and graph classification tasks, while also exhibiting stable behavior in practice.

**Weaknesses:**

Overall, I find this to be a strong paper. The following are relatively minor weaknesses that, if addressed, could further strengthen the work and broaden its impact. They mostly relate to the practical aspects of the proposed algorithm.

* The paper already provides formal convergence guarantees for the Frank–Wolfe solver, including an $\mathcal{O}(\frac{1}{T})$ optimization error bound and a consistency result, which is a clear strength. However, I would still have liked to see the empirical convergence behavior of the algorithm in the numerical section. In particular, plots showing the CDOT objective, matching accuracy, or MSE as a function of the number of FW iterations would make the optimization story more concrete and help the reader assess the practical iteration budget. Such results would further strengthen the paper by complementing the theory with evidence on convergence in practice. This can be included in the Appendix if space is an issue.

* While the appendix provides a useful complexity analysis of the proposed Frank–Wolfe solvers, the paper does not report actual wall-clock runtimes. Since a major motivation of the work is improved optimization tractability through convexification, it would substantially strengthen the paper to include runtime comparisons against FGW/EFGW (and other baselines) and, if relevant, between standard FW and the proposed lazy-gradient FW implementation. Such results would help quantify the practical computational benefit of the method beyond the asymptotic complexity discussion.

---

> ### Author Rebuttal · Authors · 2026-03-30
>
> We appreciate the reviewer’s focused questions on solver behavior, convergence, and runtime, which helped us strengthen the empirical validation. We address the main points below.
>
> > 1. “Could you please provide empirical convergence plots for the proposed CDOT solver, comparing lazy Frank-Wolfe and standard Frank-Wolfe across different attributed compact metric-measure spaces?”
>
> We ran the requested empirical convergence study. Since plots cannot be attached directly in this response field, we summarize the main findings in the table below. In our response experiments, we compare lazy Frank--Wolfe (FW) and standard FW on three attributed compact metric-measure space families: quadrant-labeled point clouds, anisotropic clustered point clouds, and attributed stochastic block model graphs. In all three cases, the CDOT objective decreases monotonically and the two FW variants produce nearly indistinguishable trajectories. For the representative `N=400` instances, the maximum discrepancy is below `1.9e-7` in objective value and below `2.5e-6` in duality gap. We also checked larger quadrant instances (`N=800` and `N=1200`) and observed the same behavior. This is consistent with the theory: the lazy update implements the same FW iterates through the affine-gradient identity, so it preserves the convergence behavior of the standard method. Here, `Lazy (s)` and `Standard (s)` denote total wall-clock runtime, `Max obj. diff.` denotes the maximum absolute difference in objective value over the run, and `Max gap diff.` denotes the maximum absolute difference in Frank--Wolfe duality gap.
>
>
> | Space | N | Lazy (s) | Standard (s) | Max obj. diff. | Max gap diff. |
> | --- | ---: | ---: | ---: | ---: | ---: |
> | Quadrant point cloud | 400 | 2.692 | 2.699 | 1.84e-7 | 2.47e-6 |
> | Anisotropic point cloud | 400 | 2.764 | 2.865 | 1.33e-7 | 8.36e-7 |
> | SBM graph metric | 400 | 2.890 | 2.954 | 7.80e-8 | 1.51e-6 |
>
> > 2. “Could you please provide wall-clock runtime comparisons on these problems, including comparisons with the relevant baselines?”
>
> We measured wall-clock runtime on the same synthetic problems, reporting the median of three runs and the corresponding outer-iteration counts in parentheses. Under our benchmark setting, CDOT requires more total wall-clock time than FGW and EFGW, while CDOT and FGW have comparable per-iteration cost. This indicates that the main runtime gap is driven primarily by the stopping rule and resulting iteration count (CDOT: 200 iterations in all reported runs; FGW: 5--15 iterations), rather than by a substantially larger per-iteration computational burden.
>
> For CDOT, the reported runtimes use the stricter setting `tol=1e-7` with a maximum of 200 Frank--Wolfe iterations. In the representative `N=400` convergence runs, however, the duality gap is already on the order of `1e-5` to `1e-4` by iteration 200 and crosses `1e-4` earlier on all three tested families, so a looser rule such as `tol=1e-4` would terminate noticeably earlier.
>
> | Family | N | CDOT | FGW | EFGW | CDOT/iter (s) | FGW/iter (s) |
> | --- | ---: | ---: | ---: | ---: | ---: | ---: |
> | Quadrant point cloud | 200 | 0.644s (200) | 0.015s (5) | 0.019s (2) | 3.22e-3 | 3.03e-3 |
> | Quadrant point cloud | 400 | 2.768s (200) | 0.065s (5) | 0.047s (2) | 1.38e-2 | 1.30e-2 |
> | Anisotropic point cloud | 200 | 0.696s (200) | 0.013s (5) | 0.010s (2) | 3.48e-3 | 2.61e-3 |
> | Anisotropic point cloud | 400 | 2.697s (200) | 0.061s (5) | 0.048s (2) | 1.35e-2 | 1.21e-2 |
> | SBM graph metric | 200 | 0.721s (200) | 0.026s (9) | 0.011s (2) | 3.61e-3 | 2.84e-3 |
> | SBM graph metric | 400 | 3.063s (200) | 0.220s (15) | 0.040s (2) | 1.53e-2 | 1.47e-2 |
>
> > 3. "Discussions regarding limitation and impact are not discussed."
>
> Thank you for pointing this out. We will add an explicit limitations and impact discussion in the revision. CDOT is designed for operator-level alignment rather than nearly one-to-one matching. In particular, zero discrepancy does not necessarily imply exact isomorphism, so GW/FGW may be more appropriate when rigid matching is the main goal. At the same time, this relaxation is what makes CDOT attractive in settings where global optimization and robustness to noisy or smoothed geometry are more important. In addition, although CDOT is convex, the current solver is still dominated by an OT subproblem, so our current implementation is better suited to moderate problem sizes than to very large-scale settings.

---

> > ### Author Rebuttal · Reviewer_LL17 · 2026-04-02
> >
> > Thank you for the rebuttal. Including these additional discussions on solver behavior, convergence, and runtime would further strengthen an already strong paper.

---

### Official Review · Reviewer_55AN · 2026-03-12

**Soundness:** 4
**Presentation:** 4
**Significance:** 4
**Originality:** 3
**Overall Recommendation:** 5
**Confidence:** 4

**Summary:**

The paper introduces Convex Distance Operator Transport - a convex relaxation to the Gromov-Wasserstein (GW) problem where the idea is to compare the the expected difference-squared between the distance operators, that in turn are expressed via the Conditional Expectation Operators under the coupling. This allows them to compare the geometries that are similar, but with different numbers of points. The main upshot is also that this problem becomes convex in the plan, unlike the GW (Theorem 3.4). The authors then establish other properties such as pseudometricity and compare the gap with GW distance (Theorems 3.5 and 3.7). Section 4 presented the approach to compute the CDOT for distributions supported on samples. Synthetic and real data studies are conducted

**Compliance With Llm Reviewing Policy:**

Affirmed.

**Key Questions For Authors:**

Questions
1. Lemma F.1 - should also use compactness of the transport polytope along with l.s.c to establish existence?
2. Can one simplify proofs in the discrete setting?
3. What about entropy regularization of CDOT?
4. Is there a dynamical formulation?
5. Can one provide “overviews of the proofs” - make a flow-chart of different Lemmas, show how they connect in a flow-chart style to yeild the proofs (especially that of Theorem 3.7) and highlight what is the novel in the proof technique. This is essential for the readers to learn and appreciate the theory contributions. ICML is a mixed venue afterall.

**Limitations:**

No.

1. Please discuss potential pitfalls that can happen due to pseudo-metricity.
2. Include issues of scalability and applicability.

**Strengths And Weaknesses:**

Strengths
1. A new distance between measures supported on different metric measure spaces is presented along with metric and statistical properties.
2. The paper is written very well and is quite comprehensive on the theoretical front.
3. The numerical results provided are very comprehensive, making the contribution stand out in terms of both theory and its utility in applications.


Weakness
1. The effect of pseudometricity on performance  is not that clear.
2. The appendices are way too long to check (I do appreciate the details very much) for correctness. I believe the ideas can be distilled easily just with the discrete case.
3. This is not a weakness but perhaps an omission (do correct me if I am wrong here) - The idea resembles what is called as weak-optimal transport. I would like the authors to look into that and see if there are any connections.

---

> ### Author Rebuttal · Authors · 2026-03-30
>
> We appreciate the reviewer’s constructive feedbacks for a clearer practical interpretation, potential directions, and a more clear presentation. We address the main points below.
>
> > 1. “The effect of pseudometricity on performance is not that clear.”
>
> We address the theorem-level characterization at Comment 1 for Reviewer gKrD. Practically, pseudometricity is crucial in graph classification, where CDOT is used as a pairwise distance between attributed graphs. It ensures that similar graphs are mapped to small distances in a consistent manner, which is essential for downstream classifiers. In contrast to FGW, CDOT avoids non-convex optimization and yields more reliable distance estimates, which translates into strong empirical performance across all five benchmark datasets.
>
> > 2. “What about entropy regularization of CDOT?”
>
> Entropic regularization is a natural extension of CDOT, but it is optional here because the original CDOT objective is already convex. More generally, entropic regularization can also improve GW-type optimization landscapes. An entropic CDOT variant would define a different regularized discrepancy and introduce an additional hyperparameter. For this reason, we focus in this work on the original unregularized formulation, but we agree that an entropic CDOT variant is a promising direction for future work.
>
> * Rioux, G., Goldfeld, Z., and Kato, K. (2024). “Entropic Gromov--Wasserstein distances: Stability and algorithms.” Journal of Machine Learning Research 25(363):1-52.
>
> > 3. "Is there any connection to weak optimal transport?"
>
> Thank you for pointing this out. CDOT is related to weak optimal transport because its structural term depends on conditional laws and conditional expectations rather than directly on pointwise pairs. In standard weak OT, the objective typically has the form
> $$
> \\inf_{\\pi\\in\\Pi(\\mu,\\nu)} \\int_X C(x,\\pi_x)\\,\\mu(dx),
> $$
> where the cost depends on a point and a conditional law rather than directly on $(x,y)$. CDOT is similar in spirit, but classical weak OT is typically one-sided, whereas CDOT uses both conditional laws symmetrically. We will add this comparison in the revision.
>
> * Backhoff-Veraguas, J. and Pammer, G. (2022). “Stability of martingale optimal transport and weak optimal transport.” The Annals of Applied Probability 32(1):721-752.
>
> > 4. “Is there a dynamical formulation?”
>
> This is a very insightful suggestion. If by a dynamical formulation one means a Benamou-Brenier-type characterization with time-dependent measures and a continuity equation, then such a formulation is not immediate for CDOT. The main obstacle is that CDOT is a static coupling problem between two possibly different metric-measure spaces with an operator-based structural term, rather than transport within a common ambient space. We therefore view a genuine dynamical characterization as an interesting open direction.
>
> > 5. “Lemma F.1 -- should also use compactness of the transport polytope along with l.s.c. to establish existence?”
>
> You are absolutely right that compactness is required for Lemma F1. While the compactness statement was already present in the manuscript (line 1012), we agree that its role in the proof was not sufficiently explicit. We will therefore revise the proof so that the existence argument more clearly invokes compactness.
>
> > 6. “Can the ideas in the appendices distilled easily with discrete cases?"
>
> Yes, and we agree that this is a helpful way to present the main idea. In this setting, the operators reduce to matrices and the feasible set becomes the transport polytope, so the CDOT objective is a quadratic function over a compact convex set. As a result, existence follows from continuity and compactness, without requiring lower-semicontinuity arguments. Convexity is also immediate: the feature term is linear in the coupling, while the structural term is a squared Frobenius norm of an affine transformation.
>
> > 7. "Can one provide the overviews of the proofs?"
>
> Thank you for your suggestion. We agree that the appendix would benefit from a clearer roadmap. For instance, Theorem 3.7 builds on the kernel representation of CDOT in Appendix F.2, yielding the decomposition $E(Z^2) = E(Z)^2 + Var(Z)$, which respectively correspond to GW, CDOT, and the dispersion term. We will add a concise overview to clarify the flow of the main proof components.
>
> > 8. "What is novel in the proof technique?"
>
> Thank you for your question. The novelty lies in handling conditional measures via the operator formulation, in particular linking weak convergence of couplings to weak-operator convergence to establish existence. For risk consistency, we introduce a lifting construction that bridges empirical and population couplings, overcoming the discrete–continuous mismatch.
>
> > 9. "Limitations should be discussed."
>
> Thank you for your feedback. Due to space constraints, we provide discussions on limitations and potential impacts in our response to Reviewer LL17.

---

> > ### Author Rebuttal · Reviewer_55AN · 2026-04-01
> >
> > The authors promise to address presentation and technical concerns in the final version. I look forward to those.

---

### Decision · Program_Chairs · 2026-04-30

**Decision:**

Accept (regular)

**Comment:**

This work introduces a novel optimal transport distance for comparing metric measure spaces, in the spirit of the Gromov-Wasserstein distance. The key idea is to rely on aligning operators rather than point clouds directly. The resulting distance has the important property of being convex, and is evaluated on a broad range of applications including graph classification, point clouds, and brain connectomes.

Overall, the paper was very well received by all reviewers, who appreciated the proposed approach, the quality of the writing, and the comprehensive numerical results covering a wide scope of applications. The theoretical contributions were also found compelling, notably the convexity of the distance, finite sample guarantees, existence of OT plans, and convergence of the Frank-Wolfe algorithm. The rebuttal further strengthened the paper by providing additional runtime experiments, demonstrating that the proposed metric is comparable to standard GW in terms of computational cost (seconds per iteration).

One point of discussion that received more mixed reactions concerns the relationship between CDOT and GW. Some reviewers found the current characterization of when CDOT vanishes compared to GW unsatisfactory, and argued that the positioning with respect to GW should be clarified: CDOT should be presented as an alternative rather than a replacement, as it does not share the same invariance properties as GW. Table 1 was flagged as potentially misleading in this regard, and a broader discussion of the respective properties of CDOT and GW would strengthen the paper.

It seems to me that the graph matching literature might offer some relevant insights on this point. Specifically, in the case of permutation matrices, CDOT and GW appear to be equivalent: GW minimizes over X a loss of the form $−\text{tr}(B X^\top A X)$ (see e.g. Proposition 1 Gromov-Wasserstein Averaging of Kernel and Distance Matrices, Peyré, Cuturi, Solomon), while CDOT minimizes over X the quantity $\|AX − XB\|^{2}_F$ as described in Section 4).
When X is constrained to permutation matrices — and is therefore orthogonal — both losses are equivalent, as described for instance at the beginning of "Probably Concave Graph Matching" (Maron & Lipman).

Overall, this is a strong contribution, and I recommend acceptance. I am confident that the authors will take into account the different remarks on GW vs CDOT for improving again the paper in its final version.